# Error Broadcast and Decorrelation as a Potential Artificial and Natural Learning Mechanism

**Mete Erdogan**[1,2]    **Cengiz Pehlevan**[3,4,5]    **Alper T. Erdogan**[1,2]

[1]KUIS AI Center, Koc University, Turkey    [2]EEE Department, Koc University, Turkey
[3]John A. Paulson School of Engineering & Applied Sciences, Harvard University, USA
[4]Kempner Institute for the Study of Natural and Artificial Intelligence, Harvard University, USA
[5]Center for Brain Science, Harvard University, USA

{merdogan18, alperdogan}@ku.edu.tr,  cpehlevan@seas.harvard.edu

## Abstract

We introduce *Error Broadcast and Decorrelation* (EBD), a novel learning framework for neural networks that addresses credit assignment by directly broadcasting output errors to individual layers, circumventing weight transport of backpropagation. EBD is rigorously grounded in the stochastic orthogonality property of Minimum Mean Square Error estimators. This fundamental principle states that the error of an optimal estimator is orthogonal to functions of the input. Guided by this insight, EBD defines layerwise loss functions that directly penalize correlations between layer activations and output errors, thereby establishing a principled foundation for error broadcasting. This theoretically sound mechanism naturally leads to the experimentally observed three-factor learning rule and integrates with biologically plausible frameworks to enhance performance and plausibility. Numerical experiments demonstrate EBD's competitive or better performance against other error-broadcast methods on benchmark datasets. Our findings establish EBD as an efficient, biologically plausible, and principled alternative for neural network training. The implementation is available at: https://github.com/meterdogan07/error-broadcast-decorrelation.

## 1 Introduction

Neural networks are dominant mathematical models for biological and artificial intelligence. A major challenge in these networks is determining how to adjust individual synaptic weights to optimize a global learning objective, known as the *credit assignment problem*. In Artificial Neural Networks (ANNs), the most common solution is the *backpropagation* (BP) algorithm [1].

In contrast to ANNs, the mechanisms for credit assignment in biological neural networks remain poorly understood. While backpropagation is highly effective for training ANNs, it is not directly applicable to biological systems because it relies on biologically implausible assumptions. In its standard form, backpropagation propagates output errors backward through a separate pathway, reusing the same synaptic weights as in the forward pass (Figure 1a). This requirement for weight symmetry is not supported by biological evidence [2]. Although many experimentally motivated models of local synaptic plasticity have been proposed [3], a biologically feasible theory of credit assignment that integrates these mechanisms remains unresolved.

To address the credit assignment problem in biological networks, researchers have proposed methods known as *error broadcasting* [4–9]. These methods involve broadcasting the global output error directly to all layers, often through random projections or fixed pathways, without relying on precise backward paths or symmetric weights (as summarized in Section 1.1). This eliminates the weight

39th Conference on Neural Information Processing Systems (NeurIPS 2025).

symmetry issue inherent in backpropagation. Error broadcasting offers practical benefits for hardware implementation; recent work [10] demonstrates potential for efficient neural network execution. However, despite encouraging progress in both theory and application [11, 12], error broadcasting still needs stronger theoretical foundations to fully validate and enhance its training effectiveness.

In this context, we introduce a novel learning framework termed the *Error Broadcast and Decorrelation* (EBD), which builds on basic error broadcasting by introducing layer-specific objectives grounded in estimation theory. The fundamental principle of EBD is to adjust network weights to minimize the correlation between broadcast output errors and the activations of each layer. This approach is rigorously grounded in Minimum Mean Square Error (MMSE) estimation, where an optimal estimator's error is orthogonal to any measurable function of its input. We leverage this orthogonality principle for EBD, defining layer-specific training losses to drive layer activations (functions of the network input) towards orthogonality with the broadcast error. This enables a more distributed mechanism for credit assignment, alternative to approaches relying solely on an output-defined loss and end-to-end error propagation.

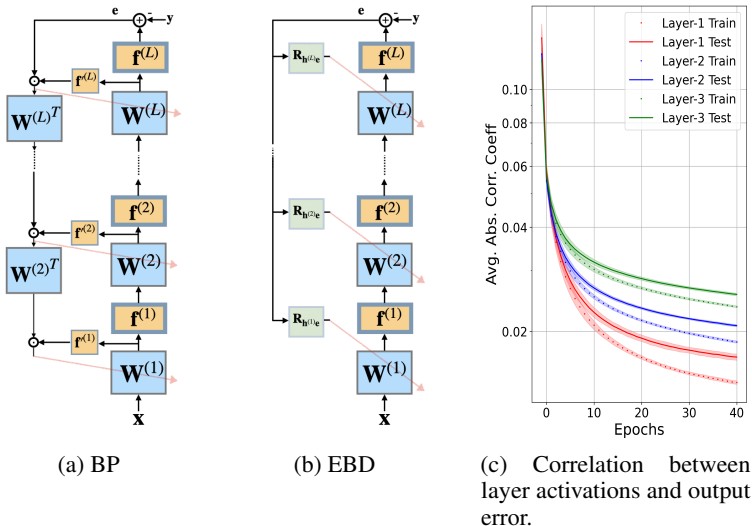

(a) BP      (b) EBD      (c) Correlation between layer activations and output error.

Figure 1: Comparison of error feedback mechanisms and correlation dynamics in multilayer perceptrons. (a) Backpropagation (BP) transmits errors sequentially through symmetric backward paths. (b) Error Broadcast and Decorrelation (EBD) broadcasts output errors to all layers using error–activation cross-correlations. (c) Average absolute correlation between layer activations and the output error during BP training on CIFAR-10 with MSE loss, illustrating its decline over epochs (see Appendix J).

EBD directly broadcasts output errors to layers, simplifying credit assignment and enabling parallel synaptic updates. It offers two key advantages for biologically realistic networks. First, optimizing EBD's loss naturally leads to experimentally observed three-factor learning rules [13, 14], which extend Hebbian plasticity by incorporating a neuromodulatory signal (the third factor) modulating synaptic updates based on pre- and postsynaptic activity. Second, by broadcasting errors directly to layers as shown in Figure 1b, it overcomes the weight transport problem inherent in backpropagation and some more biologically plausible credit assignment approaches [15, 16].

We demonstrate EBD's utility by applying it to both artificial and biologically realistic neural networks. Benchmark results show EBD matching/exceeding state-of-the-art error-broadcast techniques. Its successful application to a 10-layer biologically plausible network (CorInfoMax-EBD) provides initial evidence of depth scalability for more complex tasks.

## 1.1 Related work and contributions

Several frameworks have been proposed as alternatives to the backpropagation algorithm for modeling credit assignment in biological networks [8]. These include predictive coding [15, 17, 18], similarity matching [16, 19], time-contrastive approaches [20–22], forward-only methods [23–25], target

propagation [26–28], random feedback alignment [29], and learned feedback weights [30, 31]. Alternative strategies also seek to establish local learning rules by optimizing statistical objectives, such as the Hilbert-Schmidt Independence Criterion (HSIC) bottleneck [32].

Another significant alternative is error-broadcast methods, where output errors are directly transmitted to network layers without relying on precise backward pathways or symmetric weights. Two important examples of this approach are weight and node perturbation algorithms [4, 33–35], in which global error signals are broadcast to all network units. These signals reflect the change in overall error caused by individual perturbations in the network's weights or units. A more recent and prominent example of error broadcast is Direct Feedback Alignment (DFA) [6]. In DFA, the output errors are projected onto the hidden layers through fixed random weights, effectively replacing the symmetric backward weights required in traditional backpropagation. The core challenge of weight transport has been tackled by several other methods, many of which also rely on fixed random signals or avoid feedback entirely [36, 37]. Encouragingly, a number of these biologically-plausible frameworks have demonstrated the ability to scale effectively to large datasets, underscoring their potential as viable training mechanisms [38]. This approach first emerged as a modification to the feedback alignment approach (which replaced the symmetric weights of the backpropagation algorithm with random ones). DFA has been extended and analyzed in several studies [11, 12, 39–41], demonstrating its potential in training neural networks with less biologically implausible mechanisms. Clark et al. [9] introduced another broadcast approach for a network with vector units and nonnegative weights for which three factor learning based update rule is applied.

Our framework for error broadcasting differentiates itself through

- a **principled method** based on the orthogonality property of nonlinear MMSE estimators,
- error projection weights determined by the **cross-correlation** between the output errors and the layer activations as opposed to random weights of DFA,
- dynamic Hebbian updating of projection weights as opposed to fixed weights of DFA,
- updates involving **arbitrary nonlinear functions** of layer activities, encompassing a family of three-factor learning rules,
- the option to project layer activities forward to the output layer.

In summary, our approach provides a theoretical grounding for the error broadcasting mechanism and suggests ways to enhance its effectiveness in training networks.

## 2    Error Broadcast and Decorrelation method

### 2.1    Problem statement

To illustrate our approach, we first assume a multi-layer perceptron (MLP) network with $L$ layers. We label the input $\mathbf{x} = \mathbf{h}^{(0)} \in \mathbb{R}^{N^{(0)}}$ and layer activations $\mathbf{h}^{(k)} \in \mathbb{R}^{N^{(k)}}$ for $k = 1, \ldots, L$, where $N^{(k)}$ is layer size. The layer activations are:

$$\mathbf{h}^{(k)} = f^{(k)}(\mathbf{u}^{(k)}), \qquad \mathbf{u}^{(k)} = \mathbf{W}^{(k)}\mathbf{h}^{(k-1)} + \mathbf{b}^{(k)}, \tag{1}$$

where $k \in \{1, \ldots L\}$ is the layer index, $f^{(k)}$ are activation functions, $\mathbf{W}^{(k)}$ weights, $\mathbf{u}^{(k)}$ preactivations and $\mathbf{b}^{(k)}$ biases. We consider input-output pairs $(\mathbf{x}, \mathbf{y})$ sampled from a joint distribution $P(\mathbf{x}, \mathbf{y})$. The performance criterion is the mean square of the output error $\boldsymbol{\epsilon} = \mathbf{h}^{(L)} - \mathbf{y}$, i.e., $\mathbb{E}_{P(\mathbf{x}, \mathbf{y})}[\|\boldsymbol{\epsilon}\|_2^2]$.

### 2.2    Error Broadcast and Decorrelation loss functions

To guide the training of our neural network (which aims to minimize this MSE), we draw inspiration from the fundamental principles of Minimum Mean Square Error (MMSE) estimation theory [42]. This theory defines an ideal estimator, denoted $\hat{\mathbf{y}}_*(\mathbf{x})$, which achieves the absolute minimum possible MSE for a given joint data distribution $P(\mathbf{x}, \mathbf{y})$. A crucial characteristic of this optimal estimator is its stochastic orthogonality property, which forms the theoretical cornerstone of our EBD approach.

Formally, considering input-output pairs $(\mathbf{x}, \mathbf{y})$ drawn from $P(\mathbf{x}, \mathbf{y})$, this optimal nonlinear MMSE estimator is given by $\hat{\mathbf{y}}_*(\mathbf{x}) = \mathbb{E}[\mathbf{y}|\mathbf{x}]$ (its derivation and properties as the optimal MSE-minimizing

function are detailed in Appendix A, Lemma A.1). Its estimation error $\boldsymbol{\epsilon}_* = \mathbf{y} - \hat{\mathbf{y}}_*(\mathbf{x})$ satisfies:

$$\mathbb{E}[\mathbf{g}(\mathbf{x})\boldsymbol{\epsilon}_*^T] = \mathbf{0}, \tag{2}$$

*for any properly measurable function* $\mathbf{g}(\mathbf{x})$ *of the input* $\mathbf{x}$ *(see Appendix A, Lemma A.2).* This means $\boldsymbol{\epsilon}_*$ is orthogonal to $\mathbf{g}(\mathbf{x})$ (i.e., their expected outer product is zero). While this orthogonality property, stated in Eq. (2), is foundational, its application in constructing estimators has predominantly been in *linear* MMSE estimation. In that context, the estimator $\hat{\mathbf{y}}(\mathbf{x})$ is constrained to be a linear function of $\mathbf{x}$, and under the linearity constraint on the estimator, Eq. (2) is restricted to a form where $\mathbf{g}(\mathbf{x}) = \mathbf{x}$. This restricted orthogonality condition has long been used to derive parameters for linear estimators, such as Wiener-Kolmogorov and Kalman filters [43].

A key aspect of our work is to leverage the *full generality* of the orthogonality condition Eq. (2) for obtaining *nonlinear* estimators. For such estimators, this condition holds for *any* measurable function $\mathbf{g}(\mathbf{x})$ and, crucially, is not only necessary but also *sufficient* for MMSE optimality (as established in Appendix B, Theorem B.1). EBD distinctively employs this sufficiency as a constructive principle to train nonlinear estimators, specifically the neural network parameters.

We model the neural network (Eq. (1)) as a parameterized nonlinear estimator $f_\Theta(\mathbf{x}) = \mathbf{h}^{(L)}(\mathbf{x}; \Theta)$ and aim to satisfy Eq. (2) for its output error $\boldsymbol{\epsilon} = f_\Theta(\mathbf{x}) - \mathbf{y}$. We choose $\mathbf{g}(\mathbf{x})$ as the network's hidden layer activations $\mathbf{h}^{(k)}(\mathbf{x}; \Theta^{(k)})$, where $\Theta^{(k)} = (\mathbf{W}^{(k)}, \mathbf{b}^{(k)})$. This choice is motivated because:

  (i). Since each hidden-layer activation is a nonlinear function of input $\mathbf{x}$, the output error of an optimal estimator should be stochastically orthogonal to those activations. Figure 1c illustrates this phenomenon by showing the evolution of the average absolute correlation between layer activations and the error signal during backpropagation training of an MLP with three hidden layers on the CIFAR-10 dataset, based on the MSE criterion. Similar correlation declining trends are also observed across different datasets and architectures (see Appendix J). The declining correlation during MSE training reflects the MMSE estimator's stochastic orthogonality of layer activations and output errors.,

  (ii). $\mathbf{h}^{(k)}$ depends on layer parameters $\Theta^{(k)}$, enabling their direct updates via differentiation,

  (iii). if hidden-layer activations form a "rich enough" set of functions of $\mathbf{x}$ (as elaborated below), then enforcing error orthogonality to them implies orthogonality to "every" function of $\mathbf{x}$.

Indeed, in Theorem B.2 of Appendix B, we show that when the hidden-layer activations—say, those in the first layer—form a sufficiently rich basis (for example, becoming dense in $L_2(P_\mathbf{x})$ as network width tends to infinity), enforcing that the output error $\boldsymbol{\epsilon}$ be orthogonal to these activations naturally drives the estimator toward the true MMSE solution. Accordingly, we aim to enforce zero correlation between the output error $\boldsymbol{\epsilon}$ and hidden layer activations, or more generally their nonlinear functions:

$$\mathbf{R}_{\mathbf{g}\boldsymbol{\epsilon}}^{(k)} = \mathbb{E}[\mathbf{g}^{(k)}(\mathbf{h}^{(k)})\boldsymbol{\epsilon}^T] = \mathbf{0}, \quad k = 0, \ldots, L, \text{ with the typical choice } \mathbf{g}^{(k)}(\mathbf{h}^{(k)}) = \mathbf{h}^{(k)}. \tag{3}$$

Building on the orthogonality property and its established sufficiency for optimality (Appendix B, Theorem B.1), we define layer-specific surrogate loss functions that enforce orthogonality conditions with respect to the hidden layer activations. As demonstrated in Section 2.3, these losses yield an alternative to backpropagation by broadcasting output errors directly to network nodes (Figure 1b).

Specifically, based on the stochastic orthogonality condition in Eq. (3), we propose minimizing the Frobenius norm of the cross-correlation matrices $\mathbf{R}_{\mathbf{g}\boldsymbol{\epsilon}}^{(k)}$ as a replacement for the standard MSE loss. To this end, we define the estimated cross-correlation matrix between a function $g^{(k)}$ of layer activations and the output error for batch $m$ and layer $k$ as

$$\hat{\mathbf{R}}_{\mathbf{g}\boldsymbol{\epsilon}}^{(k)}[m] = \lambda\hat{\mathbf{R}}_{\mathbf{g}\boldsymbol{\epsilon}}^{(k)}[m-1] + \frac{1-\lambda}{B}\mathbf{G}^{(k)}[m]\mathbf{E}[m]^T,$$

where $m$ is the batch index, $\lambda \in [0, 1]$ is the forgetting factor used in the autoregressive estimation, $B$ is the batch size, $\hat{\mathbf{R}}_{\mathbf{g}\boldsymbol{\epsilon}}^{(k)}[0]$ is the initial value hyperparameter for the correlation matrix, and

$$\mathbf{G}^{(k)}[m] = \left[ \begin{array}{ccc} \mathbf{g}^{(k)}(\mathbf{h}^{(k)}[mB+1]) & \ldots & \mathbf{g}^{(k)}(\mathbf{h}^{(k)}[mB+B]) \end{array} \right], \tag{4}$$

is the matrix of nonlinearly transformed activations of layer $k$ for batch $m$. In the above equation, $mB + l$ refers to absolute (sequence) index for the $l^{th}$ member of batch-$m$. Furthermore,

$$\mathbf{E}[m] = \left[ \begin{array}{ccc} \boldsymbol{\epsilon}[mB+1] & \ldots & \boldsymbol{\epsilon}[mB+B] \end{array} \right], \tag{5}$$

is the output error matrix for batch $m$. We then define the layer-specific loss function based on the stochastic orthogonality condition for layer $k$ as

$$J^{(k)}(\mathbf{h}^{(k)}, \boldsymbol{\epsilon})[m] = \frac{1}{2} \left\| \hat{\mathbf{R}}_{\mathbf{g}\boldsymbol{\epsilon}}^{(k)}[m] \right\|_F^2, \tag{6}$$

where $\| \cdot \|_F$ denotes the Frobenius norm. This loss function captures the sum of the squared magnitudes of all cross-correlations between the components of the output error and the (potentially transformed) activations of layer $k$. Thus, we refer to the minimization of this loss as *decorrelation*.

## 2.3 Error Broadcast and Decorrelation algorithm

The functions in Eq. (6) defines individual loss functions for each hidden layer, which are used to adjust the layer parameters. These loss functions can be minimized using a gradient based algorithm.

To minimize the loss for layer $k$, we compute the gradient of the loss function $J^{(k)}(\mathbf{h}^{(k)}, \boldsymbol{\epsilon})$ with respect to the weight $W_{ij}^{(k)}$. The derivative can be decomposed into two terms:

$$\frac{\partial J^{(k)}(\mathbf{h}^{(k)}, \boldsymbol{\epsilon})}{\partial W_{ij}^{(k)}}[m] = \underbrace{\zeta Tr\left( \hat{\mathbf{R}}_{\mathbf{g}\boldsymbol{\epsilon}}^{(k)}[m]\mathbf{E}[m]\frac{\partial \mathbf{G}^{(k)}[m]^T}{\partial W_{ij}^{(k)}} \right)}_{[\Delta \mathbf{W}_1^{(k)}[m]]_{ij}} + \underbrace{\zeta Tr\left( \hat{\mathbf{R}}_{\mathbf{g}\boldsymbol{\epsilon}}^{(k)}[m]\frac{\partial \mathbf{E}[m]}{\partial W_{ij}^{(k)}}\mathbf{G}^{(k)}[m]^T \right)}_{[\Delta \mathbf{W}_2^{(k)}[m]]_{ij}},$$

where $\zeta := (1 - \lambda)/B$. Similarly, the derivative with respect to the bias $b_i^{(k)}$ is given by:

$$\frac{\partial J^{(k)}(\mathbf{h}^{(k)}, \boldsymbol{\epsilon})}{\partial b_i^{(k)}}[m] = \underbrace{\zeta Tr\left( \hat{\mathbf{R}}_{\mathbf{g}\boldsymbol{\epsilon}}^{(k)}[m]\mathbf{E}[m]\frac{\partial \mathbf{G}^{(k)}[m]^T}{\partial b_i^{(k)}} \right)}_{[\Delta \mathbf{b}_1^{(k)}[m]]_i} + \underbrace{\zeta Tr\left( \hat{\mathbf{R}}_{\mathbf{g}\boldsymbol{\epsilon}}^{(k)}[m]\frac{\partial \mathbf{E}[m]}{\partial b_i^{(k)}}\mathbf{G}^{(k)}[m]^T \right)}_{[\Delta \mathbf{b}_2^{(k)}[m]]_i}.$$

Here $\Delta \mathbf{W}_1^{(k)}, \Delta \mathbf{b}_1^{(k)}[m]$ $(\Delta \mathbf{W}_2^{(k)}, \Delta \mathbf{b}_2^{(k)}[m])$ represent the components of the gradients containing derivatives of activations (output errors) with respect to the layer parameters. As derived in Appendix C.1, we obtain the closed-form expressions for $\Delta \mathbf{W}_1^{(k)}[m]$ and $\Delta \mathbf{b}_1^{(k)}[m]$:

$$[\Delta \boldsymbol{W}_1^{(k)}[m]]_{ij} = \zeta \sum_{n=mB+1}^{(m+1)B} \vartheta^{(k)}[n]h_j^{(k-1)}[n], \qquad [\Delta \mathbf{b}_1^{(k)}[m]]_i = \zeta \sum_{n=mB+1}^{(m+1)B} \vartheta^{(k)}[n], \tag{7}$$

where $\vartheta^{(k)}[n] = g'^{(k)}_i(h_i^{(k)}[n])f'^{(k)}(u_i^{(k)}[n])q_i^{(k)}[n]$, $g'^{(k)}_i$ and $f'^{(k)}$ denote the derivatives of the nonlinearity $g^{(k)}$ and the activation function $f^{(k)}$, respectively. The term $\mathbf{q}^{(k)}[m]$ is defined as:

$$\mathbf{q}^{(k)}[m] = \hat{\mathbf{R}}_{\mathbf{g}\boldsymbol{\epsilon}}^{(k)}[m]\, \boldsymbol{\epsilon}[m],$$

representing the projection of the output error onto the layer activations, with the cross-correlation matrix $\hat{\mathbf{R}}_{\mathbf{g}\boldsymbol{\epsilon}}^{(k)}[m]$ as the transformation matrix. These projections are shown in Figure 1b.

For the special case of batchsize, $B = 1$, the weight update in (7) simplifies to

$$[\Delta \boldsymbol{W}_1^{(k)}[m]]_{ij} = \zeta g'^{(k)}_i(h_i^{(k)}[m])f'^{(k)}(u_i^{(k)}[m])q_i^{(k)}[m]h_j^{(k-1)}[m]. \tag{8}$$

The update terms $\Delta \boldsymbol{W}_1^{(k)}[m]$ and $\Delta \mathbf{b}_1^{(k)}[m]$ aim to adjust the activations to gradually become orthogonal to $\boldsymbol{\epsilon}$, as they are based on the derivatives of activations with respect to layer parameters. In contrast, $\Delta \boldsymbol{W}_2^{(k)}[m]$ and $\Delta \mathbf{b}_2^{(k)}[m]$, derived from the derivatives of the output error, work to push the output errors into a configuration more orthogonal to the activations. While both update types strive for decorrelation, a critical distinction exists: $\Delta \boldsymbol{W}_1^{(k)}[m]$ and $\Delta \mathbf{b}_1^{(k)}[m]$ depend only on layer activations and broadcast error signals, whereas $\Delta \boldsymbol{W}_2^{(k)}[m]$ and $\Delta \mathbf{b}_2^{(k)}[m]$ rely on signals propagated backward from the output layer, resembling backpropagation (see Appendix C.2).

By focusing solely on $\Delta \boldsymbol{W}_1^{(k)}[m]$ and $\Delta \mathbf{b}_1^{(k)}[m]$, we eliminate the need for propagation terms, resulting in a completely localized update mechanism for training the neural network. This simplification to

localized updates is supported by their positive alignment with backpropagation and full (untruncated) EBD gradient directions, as demonstrated in Appendix E.1 and E.2, respectively. Therefore, we prescribe the Error Broadcast and Decorrelation (EBD) update expressions as:

$$\mathbf{W}^{(k)}[m+1] = \mathbf{W}^{(k)}[m] - \mu^{(k)}[m]\Delta\mathbf{W}_1^{(k)}[m], \quad \mathbf{b}^{(k)}[m+1] = \mathbf{b}^{(k)}[m] - \mu^{(k)}[m]\Delta\mathbf{b}_1^{(k)}[m],$$

for $k = 1, \ldots, L-1$, where $\mu^{(k)}[m]$ is the learning rate for layer $k$ at batch $m$. Although these updates resemble backpropagation, a key difference lies in the error signals: the backpropagated error is replaced by the broadcasted error. Furthermore, the algorithm introduces flexibility by allowing the choice of nonlinearity functions $g^{(k)}$, which influence the gradient terms $\Delta\mathbf{W}_1^{(k)}[m]$ and $\Delta\mathbf{b}_1^{(k)}[m]$.

For the final layer ($k = L$), we utilize the standard MMSE gradient update:

$$\mathbf{W}^{(L)}[m+1] = \mathbf{W}^{(L)}[m] - \frac{\mu^{(L)}[m]}{B} \sum_{n=mB+1}^{(m+1)B} \left( f'^{(k)}(\mathbf{u}^{(L)}[n]) \odot \boldsymbol{\epsilon}[n] \right) \mathbf{h}^{(L-1)}[n]^T,$$

$$\mathbf{b}^{(L)}[m+1] = \mathbf{b}^{(L)}[m] - \frac{\mu^{(L)}[m]}{B} \sum_{n=mB+1}^{(m+1)B} f'^{(k)}(\mathbf{u}^{(L)}[n]) \odot \boldsymbol{\epsilon}[n],$$

where $f'^{(L)}$ is the derivative of the activation function of the output layer.

## 2.4  Further EBD algorithm extensions

We propose further extensions to the EBD framework to address potential activation collapse, which can arise when minimizing correlations is the sole objective. To prevent unit-level collapse, we introduce power regularization, while entropy regularization is employed to prevent dimensional collapse. Both regularizations can be implemented in ANNs as well as biologically plausible networks. The biological plausibility of employing these regularizers in MLP-based EBD is discussed in Appendix H.1. Although CorInfoMax-EBD inherently includes entropy regularization, it can also benefit from the addition of power regularization for enhanced stability. Additionally, we introduce forward layer activation projections to improve the algorithm's versatility. We also extend the EBD formulations to more complex architectures, including Convolutional Neural Networks (CNNs) and Locally Connected (LC) networks. For further details on these extensions, please refer to Appendix D.

### 2.4.1  Avoiding collapse

A critical challenge with EBD is potential activation collapse, where decorrelation losses (Eq. (6)) are minimized by driving activations $\mathbf{h}^{(k)} \to 0$, even with non-zero output errors, undermining learning. To counteract this, we introduce two complementary regularizers:

**Power normalization:** To prevent total activation collapse, power normalization (Eq. (9)) regulates layer activation power around a target level $P^{(k)}$.

$$J_P^{(k)}(\mathbf{h}^{(k)})[m] = \sum_{l=1}^{N^{(k)}} (B^{-1} \sum_{n=mB+1}^{(m+1)B} h_l^{(k)}[n]^2 - P^{(k)})^2, \tag{9}$$

which simplifies to

$$J_P^{(k)}(\mathbf{h}^{(k)})[m] = \sum_{l=1}^{N^{(k)}} (h_l^{(k)}[n]^2 - P^{(k)})^2, \tag{10}$$

for $B = 1$.

**Layer entropy:** To mitigate collapse into low-dimensional subspaces, which restricts expressiveness, we incorporate layer entropy (Eq. (11)), building on prior work [44, 45].

$$J_E^{(k)}(\mathbf{h}^{(k)})[m] = \frac{1}{2} \log \det(\mathbf{R_h}^{(k)}[m] + \varepsilon^{(k)}\mathbf{I}). \tag{11}$$

Here, $\mathbf{R}_{\mathbf{h}}^{(k)}[m]$ is the layer autocorrelation matrix, updated autoregressively with forgetting factor $\lambda_E$:

$$\mathbf{R}_{\mathbf{h}}^{(k)}[m] = \lambda_E \mathbf{R}_{\mathbf{h}}^{(k)}[m-1] + (1 - \lambda_E)\frac{1}{B}\mathbf{H}^{(k)}[m]\mathbf{H}^{(k)}[m]^T, \qquad (12)$$

where $\qquad \mathbf{H}^{(k)}[m] = [\ \mathbf{h}^{(k)}[mB+1]\ \ \ldots\ \ \mathbf{h}^{(k)}[(m+1)B]\ ], \qquad (13)$

is the activation matrix. Gradient derivations for these objectives are in Appendices C.3 and C.4.

### 2.4.2 Forward broadcast

In the EBD algorithm (Section 2.3), output errors are broadcast to layers to adjust weights and reduce correlations with activations. To complement this, we introduce forward broadcasting, projecting hidden layer activations onto the output layer to optimize the decorrelation loss by adjusting the final layer's parameters. Details are provided in Appendix C.5.

### 2.4.3 Extensions to other network architectures

The EBD approach is independent of network topology. We extend EBD to convolutional neural networks (CNNs) in Appendix D.1 and to locally connected (LC) networks in Appendix D.2.

## 3 EBD for biologically realistic networks

In the previous section, we introduced the EBD algorithm within the context of MLP networks. While MLPs can resemble biologically plausible networks depending on the credit assignment mechanism, in this section, we extend the application of the EBD approach to neural networks that exhibit more biologically realistic dynamics and architectures, motivated by its inherent solutions to key neuroscientific challenges: 1) EBD's direct error broadcast naturally resolves the problematic weight symmetry requirement of BP, and 2) its update rules intrinsically manifest as modulated, extended Hebbian mechanisms (three-factor learning), aligning with current understanding of synaptic plasticity. In the following subsections, we explore how EBD relates to the biologically plausible three-factor learning rule and demonstrate its integration with the biologically more realistic CorInfoMax networks [45].

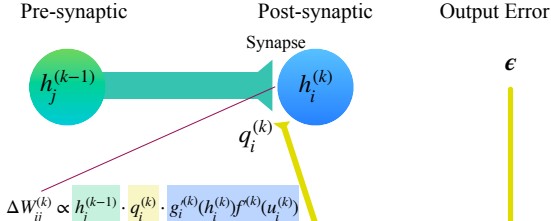

Figure 2: **Error–broadcast learning as a three-factor synaptic update.** The presynaptic firing rate $h_j^{(k-1)}$ (green, left) projects onto the postsynaptic neuron $h_i^{(k)}$ (blue, centre) through the synapse $W_{ij}^{(k)}$. A layer-specific broadcast of the output error $e \to q_i^{(k)}$ (yellow, right) provides the modulatory third factor that gates plasticity. Together, presynaptic activity, postsynaptic non-linear derivatives $g_i^{\prime(k)} f^{\prime(k)}$ (blue rectangle), and the modulatory signal form the product that drives the EBD weight change $\Delta W_{ij}^{(k)}$ displayed underneath the circuit.

### 3.1 Three factor learning rule and EBD

The three-factor learning rule for biological neural networks extends the traditional two-factor Hebbian rule by incorporating a modulatory signal into synaptic updates based on presynaptic and postsynaptic activity [13, 46]. While backpropagation can be expressed similarly, it is not typically considered a three-factor rule in neuroscience, as its 'third factor' is a locally tailored signal specific to each neuron requiring a biologically implausible dual network with symmetric weights, unlike global neuromodulatory signals [13]. In contrast, EBD update for batchsize $B = 1$ in (8) naturally matches the three-factor structure:

$$\Delta W_{ij}^{(k)} \propto \underbrace{g_i^{\prime(k)}(h_i^{(k)}) f^{\prime(k)}(u_i^{(k)})}_{\text{Postsynaptic}}\ \underbrace{q_i^{(k)}}_{\text{Modulatory}}\ \underbrace{h_j^{(k-1)}}_{\text{Presynaptic}},$$

where $q_i^{(k)}$ is the projected global error. Thus, EBD supports various three-factor rules depending on nonlinearity $g^{(k)}$. For example, $g_i^{(k)}(h_i^{(k)}) = h_i^{(k)2}$ yields the error-modulated Hebbian update

[11, 47]: $\Delta W_{ij}^{(k)} \propto h_i^{(k)} f'^{(k)}(u_i^{(k)}) q_i^{(k)} h_j^{(k-1)}$. By enabling diverse three-factor updates via different nonlinear functions, EBD holds potential for modeling biologically consistent neural learning processes.

Figure 2 breaks the EBD weight update (8) into its three interacting factors. The presynaptic activity $h_j^{(k-1)}$ from the sending unit; the postsynaptic term is $g_i'^{(k)} f'^{(k)}$ computed from the receiving unit's own activation; and the modulatory broadcast error $q_i^{(k)}$, derived from the network's output error $\epsilon$. Multiplying these three quantities produces the weight change $\Delta W_{ij}^{(k)}$ shown beneath the diagram, revealing that EBD naturally realises the classical three-factor learning rule in neural networks.

### 3.2 CorInfoMax-EBD: CorInfoMax with three factor learning rule

One of the significant advantages of the EBD framework is its flexibility to broadcast output errors into network nodes, which can be leveraged to transform time-contrastive, biologically plausible approaches into non-contrastive forms. To illustrate this property, we propose a modification of the recently introduced CorInfoMax framework [45] (see Appendix F for a summary). The CorInfoMax approach uses correlative information flow between layers as its objective function:

$$\mathcal{J}_{CI}[m] = \sum_{k=1}^{L-1}(\hat{I}_r^{(\varepsilon_k)}(\mathbf{h}^{(k-1)}, \mathbf{h}^{(k)})[m] + \hat{I}_l^{(\varepsilon_k)}(\mathbf{h}^{(k)}, \mathbf{h}^{(k+1)})[m]), \quad \text{where,}$$

$$\hat{I}_r^{(\varepsilon_k)}(\mathbf{h}^{(k)}, \mathbf{h}^{(k+1)})[m] = 0.5\log\det(\hat{\mathbf{R}}_{\mathbf{h}^{(k+1)}}[m] + \varepsilon_k\boldsymbol{I}) - 0.5\log\det(\hat{\boldsymbol{R}}_{\overrightarrow{\mathbf{e}}^{(k+1)}}[m] + \varepsilon_k\boldsymbol{I}),$$

$$\hat{I}_l^{(\varepsilon_k)}(\mathbf{h}^{(k)}, \mathbf{h}^{(k+1)})[m] = 0.5\log\det(\hat{\mathbf{R}}_{\mathbf{h}^{(k)}}[m] + \varepsilon_k\boldsymbol{I}) - 0.5\log\det(\hat{\boldsymbol{R}}_{\overleftarrow{\mathbf{e}}^{(k)}}[m] + \varepsilon_k\boldsymbol{I}),$$

are alternative forms of correlative mutual information between nodes, defined in terms of the correlation matrices of layer activations, i.e., $\hat{\mathbf{R}}_{\mathbf{h}^{(k)}}$ and forward and backward prediction errors ($\hat{\boldsymbol{R}}_{\overrightarrow{\mathbf{e}}^{(k+1)}}$ and $\hat{\boldsymbol{R}}_{\overleftarrow{\mathbf{e}}^{(k)}}$). Here, forward/backward prediction errors are defined by

$$\overrightarrow{\mathbf{e}}^{(k+1)}[n] = \mathbf{h}^{(k+1)}[n] - \mathbf{W}^{(f,k)}[m]\mathbf{h}^{(k)}[n], \qquad \overleftarrow{\mathbf{e}}^{(k)}[n] = \mathbf{h}^{(k)}[n] - \mathbf{W}^{(b,k)}[m]\mathbf{h}^{(k+1)}[n],$$

respectively. Here, $\mathbf{W}^{(f,k)}[m]$ ($\mathbf{W}^{(b,k)}[m]$) is the forward (backward) prediction matrix for layer $k$.

This objective leads to network dynamics corresponding to a structure with feedforward and feedback prediction weights, and lateral connections $\mathbf{B}^{(k)}$ that maximize layer entropy. In the original work [45], the two-phase EP approach [22] is proposed to train the network weights. As an alternative, we propose employing the EBD update rule to replace the two-phase EP adaptation. The proposed CorInfoMax-EBD algorithm is described by the following update equations defined in Algorithm 1.

---

**Algorithm 1** CorInfoMax-EBD Algorithm for Updating Weights in Layer $k$

---

**Input:** Batch size $B$, layer index $k$, iteration step $m$, learning rates $\mu^{(f,k)}$, $\mu^{(b,k)}$, $\mu^{(d_f,k)}$, $\mu^{(d_b,k)}$, $\mu^{(d_l,k)}$, factors $\lambda_d$, $\lambda_E$, $\gamma_E$, activations $\mathbf{H}^{(k)}$ in Eq. (13), the nonlinear function of layer activations $\mathbf{G}^{(k)}$ in Eq. (4) and their derivatives $\mathbf{G}_d^{(k)}$ in Eq. (28), the derivative of activations $\mathbf{F}_d^{(k)}$ in Eq. (29), output error $\mathbf{E}$ in Eq. (5), prediction errors $\overleftarrow{\mathbf{E}}^{(k)}$ and $\overrightarrow{\mathbf{E}}^{(k)}$ in Eq. (54)-(55), lateral outputs $\mathbf{Z}^{(k)}$ in Eq. (56).
**Output:** Updated weights $\mathbf{W}^{(f,k)}$, $\mathbf{W}^{(b,k)}$, $\mathbf{B}^{(k)}$.
**Step 1: Update error projection weights:** $\hat{\mathbf{R}}_{\mathbf{g}\epsilon}^{(k)}[m] = \lambda_d\hat{\mathbf{R}}_{\mathbf{g}\epsilon}^{(k)}[m-1] + \frac{1-\lambda_d}{B}\mathbf{G}^{(k)}[m]\mathbf{E}[m]^T$
**Step 2: Project errors to layer $k$:** $\mathbf{Q}^{(k)}[m] = \hat{\mathbf{R}}_{\mathbf{g}\epsilon}^{(k)}[m]\mathbf{E}[m]$
**Step 3: Find the gradient of the nonlinear function of activations for layer $k$:**

$$\boldsymbol{\Phi}^{(k)}[m] = \mathbf{F}_d^{(k)}[m] \odot \mathbf{Q}^{(k)}[m] \odot \mathbf{G}_d^{(k)}[m]$$

**Step 4: Update forward, backward and lateral weights for layer $k$:**

$$\mathbf{W}^{(f,k)}[m] = \mathbf{W}^{(f,k)}[m-1] + \left(B^{-1}\mu^{(f,k)}[m]\overrightarrow{\mathbf{E}}^{(k)}[m] - B^{-1}\mu^{(d_f,k)}[m]\boldsymbol{\Phi}^{(k)}[m]\right)\mathbf{H}^{(k-1)}[m]^T$$

$$\mathbf{W}^{(b,k)}[m] = \mathbf{W}^{(b,k)}[m-1] + \left(B^{-1}\mu^{(b,k)}[m]\overleftarrow{\mathbf{E}}^{(k)}[m] - B^{-1}\mu^{(d_b,k)}[m]\boldsymbol{\Phi}^{(k)}[m]\right)\mathbf{H}^{(k+1)}[m]^T$$

$$\mathbf{B}^{(k)}[m] = \lambda_E^{-1}\mathbf{B}^{(k-1)}[m] - B^{-1}\gamma_E\mathbf{Z}^{(k)}[m]\mathbf{Z}^{(k)}[m]^T - B^{-1}\mu^{(d_l,k)}[m]\boldsymbol{\Phi}^{(k)}[m]\mathbf{H}^{(k)}[m]^T$$

---

Table 1: Accuracy (%) results for MLP, CNN, and LC networks on MNIST and CIFAR-10. Best and second-best results are bold and underlined. GEVB results are from Clark et al. [9].

| Dataset | Model | DFA | DFA+E (ours) | NN-GEVB | MS-GEVB | BP (MSE) | EBD (ours) |
|---------|-------|-----|--------------|---------|---------|----------|------------|
| MNIST | MLP | 98.1±0.21 | 98.2±0.03 | 98.1 | 97.7 | **98.7±0.05** | 98.2±0.08 |
| | CNN | 99.1±0.05 | 99.1±0.07 | 97.7 | 98.2 | **99.5±0.04** | 99.1±0.07 |
| | LC | 98.9±0.03 | 98.9±0.04 | 98.2 | 98.2 | **99.1±0.04** | 98.9±0.04 |
| CIFAR-10 | MLP | 52.1±0.33 | 52.2±0.49 | 52.4 | 51.1 | **56.4±0.33** | 55.5±0.19 |
| | CNN | 58.4±1.59 | 58.6±0.66 | 66.3 | 61.6 | **75.2±0.28** | 66.4±0.43 |
| | LC | 62.2±0.21 | 62.1±0.19 | 58.9 | 59.9 | **67.8±0.27** | 64.3±0.26 |

Table 2: Accuracy (%) results for the CNN on CIFAR-100.

| Dataset | Model | DFA | BP (CE) | EBD (ours) |
|---------|-------|-----|---------|------------|
| CIFAR-100 | CNN | 41.9±0.32 | **60.5±0.17** | 45.9±0.69 |

Here, we assume layer activations $\mathbf{H}^{(k)}$, forward (backward) prediction errors $\overrightarrow{\mathbf{E}}$ ($\overleftarrow{\mathbf{E}}$), output error $\mathbf{E}$, and lateral weight outputs $\mathbf{Z}$ are computed via CorInfoMax network dynamics in [45] (see Appendix F). Integrating EBD enables single-phase updates per input, eliminating the less biologically plausible two-phase mechanism required by CorInfoMax-EP. EP's two-phase approach—separate label-free and label-connected phases—is implausible, as biological neurons unlikely alternate between distinct global phases for learning. Our method simplifies the process, aligns more closely with biological learning, and achieves comparable or superior performance to CorInfoMax-EP (see Section 4).

The CorInfoMax-EBD scheme introduced in this section is more biologically plausible than the earlier MLP-based EBD formulation, as its learning rules can be implemented through local mechanisms such as lateral and three-factor Hebbian/anti-Hebbian updates, realistic neuron models with apical and basal dendrites, and feedback via backward predictors. Additionally, both the entropy and power normalization terms in CorInfoMax are realizable using biologically plausible operations, particularly in the online setting with single-sample updates. See Appendix H.2 for further discussion.

## 4 Numerical experiments

In this section, we evaluate the performance of the proposed Error Broadcast and Decorrelation (EBD) approach on benchmark datasets: MNIST [48] and CIFAR-10/100 [49]. For experiments with MNIST and CIFAR-10 involving MLP, CNN and LC, we use the same architectures used in [9]; while for CIFAR-100 we adopt a CNN architecture closely following that of [41]. We also tested the proposed CorInfoMax-EBD model against the CorInfoMax-EP model of [45]. More details about architectures, implementations, hyperparameter selections, and experimental outputs are provided in the Appendix I.

EBD test accuracy results compared to BP (with MSE criterion) and three error-broadcast methods: DFA without and with entropy regularization (DFA-E) [6], global error vector broadcasting (nonnegative-(NN-GEVB) and mixed-sign-(MS-GEVB))[9] are in Table 1 for both MNIST and CIFAR-10. Under our training setup, BP yielded comparable test accuracies for these datasets with both MSE and Cross-Entropy losses, though we report only the MSE results. In addition, CIFAR-100 results of EBD compared to BP (with Cross-Entropy criterion) and DFA is given in Table 2. Lastly, the test accuracies for biological CorInfoMax networks trained with EP and EBD methods are in Table 3.

These results show that EBD-trained networks achieve equivalent performance on the MNIST dataset and significantly better performance on the CIFAR-10 and CIFAR-100 datasets compared to other error broadcasting methods. These improvements of EBD in Table 1 over DFA can be attributed to the adaptability of error projection weights in EBD. The improvement of CorInfoMax-EBD over CorInfoMax-EP in Table 3 can be attributed to CorInfoMax-EBD incorporating error decorrelation in updating lateral weights, whereas CorInfoMax-EP relies only on (anti-)Hebbian updates. Particularly noteworthy is the performance of CorInfoMax-EBD, which not only substantially improves upon the original CorInfoMax-EP on CIFAR-10 (e.g., $55.79\%$ vs. $50.97\%$ for 3-layers with batch size 20) but also demonstrates encouraging scalability with depth, with a 10-layer CorInfoMax-EBD achieving $96.38\%$ on MNIST and $54.89\%$ on CIFAR-10 using online learning (batch size 1). This highlights EBD's potential in deeper, more complex biological networks.

Table 3: Accuracy (%) results for EP and EBD CorInfoMax (CIM) algorithms on MNIST and CIFAR-10. Best and second-best are bold and underlined. Column marked with [*] is from Bozkurt et al. [45].

| Dataset | CIM-EP [*] 3-Layers (batch size : 20) | CIM-EBD (Ours) 3-Layers (batch size : 20) | CIM-EBD (Ours) 3-Layers (batch size : 1) | CIM-EBD (Ours) 10-Layers (batch size : 1) |
|---------|------|------|------|------|
| MNIST | **97.6** | 97.5±0.12 | 94.9±0.16 | 96.4±0.11 |
| CIFAR-10 | 51.0 | **55.7**±**0.17** | 53.4±0.33 | 54.9±0.58 |

## 5    Conclusions, extensions and limitations

**Conclusions.**    We introduced the Error Broadcast and Decorrelation framework, a biologically plausible alternative to backpropagation. EBD addresses the credit assignment problem by minimizing correlations between layer activations and output errors, offering fresh insights into biologically realistic learning. This approach provides a theoretical foundation for existing error broadcast mechanisms and three-factor learning rules in biological neural networks and facilitates flexible implementations in neuromorphic and artificial neural systems. EBD's error-broadcasting mechanism aligns with biological processes using local updates, and notably, has proven effective for training deep recurrent biologically-plausible networks (e.g., the 10-layer CorInfoMax-EBD), thereby addressing a key challenge in effectively scaling deep, biologically plausible learning with local rules. Moreover, EBD's simplicity and parallelism suit efficient hardware, like neuromorphic systems.

**Extensions.**    The MMSE orthogonality property underlying EBD offers significant promise for new algorithms, deeper theoretical understanding, and neural network analysis in both artificial and biological contexts. Further theoretical extensions, drawing from the groundwork laid in Appendix B.2, could focus on deriving tighter convergence guarantees for EBD in practical (finite-width) settings and on investigating the impact of more adaptive choices for the decorrelation functions $g^{(k)}$. In addition, EBD provides theoretical underpinnings for error-broadcast mechanisms with three-factor learning rules, enabling the conversion of two-phase contrastive methods into a single-phase approach. We are currently unaware of similar theoretical properties for alternative loss functions. Finally, our numerical experiments in Appendix J.2 reveal that similar decorrelation behavior occurs for networks trained with backpropagation and categorical cross entropy loss, suggesting that decorrelation may be a general feature of the learning process and an intriguing avenue for further investigation.

**Impact and limitations.**    This paper seeks to advance the fields of Machine Learning and Computational Neuroscience by proposing a novel learning mechanism. As a foundational learning algorithm, we do not identify specific negative societal impacts arising directly from the EBD mechanisms beyond general considerations common to advancements in machine learning. While EBD offers a theoretically-grounded framework for error broadcast based and three factor learning that has yielded competitive (and in some cases, superior) performance against other error-broadcast methods on the presented benchmarks, several aspects warrant future investigation:

Scalability: The current work evaluates EBD on MLP, CNN, LC and recurrent biological networks for image classification tasks like MNIST and CIFAR-10. Results on the 10-layer CorInfoMax-EBD demonstrate the potential to scale EBD to deeper, biologically realistic recurrent architectures using online learning. However, assessing EBD's performance on significantly larger datasets, or its applicability to diverse large-scale architectures in other domains, remains an important open direction. While related methods like DFA have been explored in such contexts [41], comprehensive empirical validation of EBD itself under those conditions is needed.

Computational Cost and Hyperparameters: The dynamic updating of error projection matrices $\hat{\mathbf{R}}_{g\epsilon}^{(k)}$ and the optional inclusion of regularization terms like layer entropy (discussed in Appendix G and Appendix I.7) contribute to computational and memory overhead compared to simpler schemes like DFA with fixed projectors, or standard backpropagation. EBD also introduces several hyperparameters (e.g., learning rates for decorrelation and regularization, forgetting factors) that require careful tuning, although this offers flexibility. Future work could explore more efficient update mechanisms or automated tuning strategies.

## Acknowledgements

This work was supported by KUIS AI Center Research Award. C.P. was supported by an NSF CAREER Award (IIS-2239780) and a Sloan Research Fellowship. This work has been made possible in part by a gift from the Chan Zuckerberg Initiative Foundation to establish the Kempner Institute for the Study of Natural and Artificial Intelligence.

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

# Appendix

**Table of contents**

# A Preliminaries on nonlinear Minimum Mean Square Error (MMSE) estimation

Let $\mathbf{y} \in \mathbb{R}^p$ and $\mathbf{x} \in \mathbb{R}^n$ represent two non-degenerate random vectors with a joint probability density function $f_{\mathbf{yx}}(\mathbf{y}, \mathbf{x})$ and conditional density $f_{\mathbf{y}|\mathbf{x}}(\mathbf{y}|\mathbf{x})$. The goal of nonlinear minimum mean square error (MMSE) estimation is to find an estimator function $\mathbf{b} : \mathbb{R}^n \to \mathbb{R}^p$ that minimizes the mean squared error (MSE), which is defined as:

$$MSE(\mathbf{b}) = \mathbb{E}[\|\mathbf{y} - \mathbf{b}(\mathbf{x})\|_2^2].$$

**Lemma A.1.** *The best nonlinear MMSE estimate of $\mathbf{y}$ given $\mathbf{x}$ is:*

$$\mathbf{b}_*(\mathbf{x}) = \mathbb{E}_{\mathbf{y}|\mathbf{x}}(\mathbf{y}|\mathbf{x}).$$

The proof of Lemma A.1 relies on the following fundamental result (see, for example, Papoulis and Pillai [42]), which is central to the development of the entire EBD framework in the current article:

**Lemma A.2.** *The estimation error for $\mathbf{b}_*(\mathbf{x}) = \mathbb{E}_{\mathbf{y}|\mathbf{x}}[\mathbf{y}|\mathbf{x}]$, denoted as $\mathbf{e}_* = \mathbf{y} - \mathbf{b}_*(\mathbf{x})$, is orthogonal to any vector-valued function $\mathbf{g} : \mathbb{R}^n \to \mathbb{R}^k$ of $\mathbf{x}$. Formally, we have:*

$$\mathbb{E}[\mathbf{e}_* \mathbf{g}(\mathbf{x})^T] = \mathbf{0}.$$

*Proof.* (Lemma A.2) The proof follows simple steps:

$$
\begin{aligned}
\mathbb{E}[\mathbf{e}_* \mathbf{g}(\mathbf{x})^T] &= \mathbb{E}_{\mathbf{x}}\left[\mathbb{E}_{\mathbf{y}|\mathbf{x}}\left[(\mathbf{y} - \mathbb{E}_{\mathbf{y}|\mathbf{x}}[\mathbf{y}|\mathbf{x}])\mathbf{g}(\mathbf{x})^T|\mathbf{x}\right]\right] \\
&= \mathbb{E}_{\mathbf{x}}\left[\left(\mathbb{E}_{\mathbf{y}|\mathbf{x}}[\mathbf{y}|\mathbf{x}] - \mathbb{E}_{\mathbf{y}|\mathbf{x}}[\mathbf{y}|\mathbf{x}]\right)\mathbf{g}(\mathbf{x})^T\right] = \mathbf{0}.
\end{aligned}
$$

$\square$

Using Lemma A.2, we can now prove Lemma A.1:

*Proof.* (Lemma A.1) Let $\mathbf{b} : \mathbb{R}^n \to \mathbb{R}^p$ be any arbitrary function. The corresponding MSE can be written as:

$$MSE(\mathbf{b}) = \mathbb{E}[\|\mathbf{y} - \mathbf{b}(\mathbf{x})\|_2^2].$$

By adding and subtracting $\mathbb{E}_{\mathbf{y}|\mathbf{x}}[\mathbf{y}|\mathbf{x}]$, we can decompose the error as:

$$
\begin{aligned}
MSE(\mathbf{b}) &= \mathbb{E}\left[\|\mathbf{y} - \mathbb{E}_{\mathbf{y}|\mathbf{x}}[\mathbf{y}|\mathbf{x}] + \mathbb{E}_{\mathbf{y}|\mathbf{x}}[\mathbf{y}|\mathbf{x}] - \mathbf{b}(\mathbf{x})\|_2^2\right] \\
&= \mathbb{E}[\|\mathbf{y} - \mathbb{E}_{\mathbf{y}|\mathbf{x}}[\mathbf{y}|\mathbf{x}]\|_2^2] + \mathbb{E}(\|\mathbb{E}_{\mathbf{y}|\mathbf{x}}[\mathbf{y}|\mathbf{x}] - \mathbf{b}(\mathbf{x})\|_2^2) \\
&\quad + 2\mathbb{E}[(\mathbf{y} - \mathbb{E}_{\mathbf{y}|\mathbf{x}}[\mathbf{y}|\mathbf{x}])^T(\mathbb{E}_{\mathbf{y}|\mathbf{x}}[\mathbf{y}|\mathbf{x}] - \mathbf{b}(\mathbf{x}))] \\
&= \mathbb{E}[\|\mathbf{y} - \mathbb{E}_{\mathbf{y}|\mathbf{x}}[\mathbf{y}|\mathbf{x}]\|_2^2] + \mathbb{E}[\|\mathbb{E}_{\mathbf{y}|\mathbf{x}}(\mathbf{y}|\mathbf{x}) - \mathbf{b}(\mathbf{x})\|_2^2] \\
&\quad + 2\mathbb{E}\left[Tr\left((\mathbf{y} - \mathbb{E}_{\mathbf{y}|\mathbf{x}}[\mathbf{y}|\mathbf{x}])(\mathbb{E}_{\mathbf{y}|\mathbf{x}}[\mathbf{y}|\mathbf{x}] - \mathbf{b}(\mathbf{x}))^T\right)\right] \\
&= \mathbb{E}[\|\mathbf{y} - \mathbb{E}_{\mathbf{y}|\mathbf{x}}[\mathbf{y}|\mathbf{x}]\|_2^2] + \mathbb{E}[\|\mathbb{E}_{\mathbf{y}|\mathbf{x}}[\mathbf{y}|\mathbf{x}] - \mathbf{b}(\mathbf{x})\|_2^2] \\
&\quad + 2Tr(\mathbb{E}[\mathbf{e}_*(\mathbb{E}_{\mathbf{y}|\mathbf{x}}[\mathbf{y}|\mathbf{x}] - \mathbf{b}(\mathbf{x}))^T]).
\end{aligned}
$$

The third term, representing the cross product, vanishes by Lemma A.2, leaving us with:

$$MSE(\mathbf{b}) = \mathbb{E}[\|\mathbf{y} - \mathbf{b}_*(\mathbf{x})\|_2^2] + \mathbb{E}[\|\mathbf{b}_*(\mathbf{x}) - \mathbf{b}(\mathbf{x})\|_2^2].$$

Since the second term is always non-negative, the MSE is minimized when $\mathbf{b}(\mathbf{x}) = \mathbf{b}_*(\mathbf{x})$. $\square$

# B   On the stochastic orthogonality condition based network training

This appendix provides further theoretical grounding for the Error Broadcast and Decorrelation (EBD) framework, addressing the use of the nonlinear Minimum Mean Square Error (MMSE) orthogonality condition for deriving estimators and the implications of orthogonality in high-dimensional spaces. We begin by re-establishing the geometric interpretation of stochastic orthogonality in the linear MMSE setting. We then demonstrate the sufficiency of the nonlinear MMSE orthogonality condition for an estimator to be optimal. Crucially, we address, under reasonable assumptions, how the EBD algorithm, by enforcing orthogonality to hidden unit activations, can approximate orthogonality to arbitrary measurable functions of the input, particularly in the infinite width limit. Finally, we briefly touch upon the scaling of these orthogonality conditions.

## B.1   The stochastic orthogonality condition and linear MMSE estimation

Within the context of this article, *stochastic orthogonality* refers to statistical uncorrelatedness. The term *orthogonality* is defined within a Hilbert space of random variables, where the inner product between two random variables $a$ and $b$ is given by:

$$\langle a, b \rangle \triangleq \mathbb{E}[ab]. \tag{14}$$

Here, the inner product is defined as the expected value of their product, which corresponds to their correlation. Two random variables $a$ and $b$ are said to be orthogonal if their inner product is zero:

$$\langle a, b \rangle = \mathbb{E}[ab] = 0. \tag{15}$$

Thus, two random variables are orthogonal if and only if their correlation is zero (see, for example, Kailath et al. [43], Chapter 3).

In the context of Minimum Mean Square Error (MMSE) estimation, where the goal is to estimate a vector $\mathbf{y} \in \mathbb{R}^p$ from observations $\mathbf{x} \in \mathbb{R}^n$, it is known (e.g., Lemma A.2 in Appendix A) that the error $\mathbf{e}_*$ of the optimal MMSE estimator $\hat{\mathbf{y}}_*(\mathbf{x})$ satisfies:

$$\mathbb{E}\left[\mathbf{e}_* \mathbf{g}(\mathbf{x})^\top\right] = \mathbf{0}_{p \times k}, \tag{16}$$

where $\mathbf{g} : \mathbb{R}^n \to \mathbb{R}^k$ is an arbitrary vector-valued measurable function of $\mathbf{x}$. This equation states that the cross-correlation matrix between the nonlinear function of the input $\mathbf{g}(\mathbf{x})$ and the output error $\mathbf{e}_*$ is a $p \times k$ zero matrix.

The matrix equation in Eq. (16) can be expressed more explicitly as $p \cdot k$ zero-correlation conditions:

$$\mathbb{E}\left[e_{*,i} \cdot g_j(\mathbf{x})\right] = 0, \quad i = 1, \ldots, p, \quad j = 1, \ldots, k. \tag{17}$$

Using the inner product definition in Eq. (14), we can rewrite Eq. (17) as $p \cdot k$ stochastic orthogonality conditions:

$$\langle e_{*,i}, g_j(\mathbf{x}) \rangle = 0, \quad i = 1, \ldots, p, \quad j = 1, \ldots, k. \tag{18}$$

This geometric interpretation, where correlation is viewed as an inner product, is foundational. In the linear MMSE estimation setting, the orthogonality condition in (18) is restricted by choosing $\mathbf{g}(\mathbf{x})$ to be linear functions of $\mathbf{x}$, most commonly the identity mapping, i.e., $\mathbf{g}(\mathbf{x}) = \mathbf{x}$. This leads to:

$$\langle e_{*,i}, x_j \rangle = 0, \quad i = 1, \ldots, p, \quad j = 1, \ldots, n. \tag{19}$$

These linear orthogonality conditions are fundamental and are used in reverse to derive the parameters of linear MMSE estimators, such as the Wiener and Kalman filters [43]. The estimator's structure is assumed (linear), and its parameters are found by enforcing these orthogonality conditions.

## B.2   The use of stochastic orthogonality conditions for nonlinear MMSE estimation

The principle of using orthogonality conditions to define estimators extends to the nonlinear domain. A key question is whether the nonlinear MMSE orthogonality condition can be similarly used "in reverse" to identify optimal nonlinear estimators.

### B.2.1 The sufficiency of the nonlinear MMSE orthogonality condition

The nonlinear orthogonality principle states that for an optimal nonlinear MMSE estimator, the estimation error is orthogonal to any measurable function of the input. The following theorem establishes that this is not only a necessary condition but also a sufficient one, providing a strong theoretical basis for using these conditions to define and seek optimal estimators [50].

**Theorem B.1** ( Nonlinear MMSE Estimation and Orthogonality Condition)**.** *Let $y \in \mathbb{R}^p$ and $x \in \mathbb{R}^n$ be random vectors with a joint probability distribution. An estimator $\hat{y}(x)$ is the optimal nonlinear MMSE estimator, $\hat{y}_{MMSE}(x)$, if and only if the error $e = y - \hat{y}(x)$ satisfies:*

$$\mathbb{E}[e \cdot g(x)^\top] = \mathbf{0}_{p \times k}$$

*for all measurable functions $g : \mathbb{R}^n \to \mathbb{R}^k$ for any $k \geq 1$.*

*Proof.* The necessity part ($\Rightarrow$), i.e., if $\hat{y}(x) = \hat{y}_{MMSE}(x)$, then the orthogonality condition holds, is a standard result (established, for instance, in Lemma A.2 Appendix A).

Here we prove the sufficiency part ($\Leftarrow$). Let $f_\Theta(x)$ be an estimator such that its error $e_\Theta = y - f_\Theta(x)$ satisfies the orthogonality condition:

$$\mathbb{E}[(y - f_\Theta(x)) \cdot g(x)^\top] = \mathbf{0}_{p \times k_g} \tag{20}$$

for all measurable functions $g : \mathbb{R}^n \to \mathbb{R}^{k_g}$ (where $k_g$ is the dimension of $g(x)$). Consider the difference between $f_\Theta(x)$ and the true MMSE estimator $\hat{y}_{MMSE}(x)$. We can write:

$$y - f_\Theta(x) = (y - \hat{y}_{MMSE}(x)) + (\hat{y}_{MMSE}(x) - f_\Theta(x)).$$

Substituting this into the assumed orthogonality condition for $f_\Theta(x)$:

$$\mathbb{E}[((y - \hat{y}_{MMSE}(x)) + (\hat{y}_{MMSE}(x) - f_\Theta(x))) \cdot g(x)^\top] = \mathbf{0}_{p \times k_g}.$$

By linearity of expectation:

$$\mathbb{E}[(y - \hat{y}_{MMSE}(x)) \cdot g(x)^\top] + \mathbb{E}[(\hat{y}_{MMSE}(x) - f_\Theta(x)) \cdot g(x)^\top] = \mathbf{0}.$$

The first term is zero due to the (necessary) orthogonality property of the MMSE estimator $\hat{y}_{MMSE}(x)$. Therefore, we are left with:

$$\mathbb{E}[(\hat{y}_{MMSE}(x) - f_\Theta(x)) \cdot g(x)^\top] = \mathbf{0}, \tag{21}$$

for any measurable function $g(x)$. Choosing $g(x) = \hat{y}_{MMSE}(x) - f_\Theta(x)$, and taking the trace of Eq. (21), and applying the cyclic property of the trace operator, we obtain

$$\mathbb{E}[\|\hat{y}_{MMSE}(x) - f_\Theta(x)\|_2^2] = 0.$$

Since $\|\hat{y}_{MMSE}(x) - f_\Theta(x)\|_2^2$ is a non-negative random variable, its expectation being zero implies that $\|\hat{y}_{MMSE}(x) - f_\Theta(x)\|_2^2 = 0$ almost surely. This means $\hat{y}_{MMSE}(x) - f_\Theta(x) = \mathbf{0}$ almost surely, or $f_\Theta(x) = \hat{y}_{MMSE}(x)$ almost surely. Thus, $f_\Theta(x)$ is the optimal MMSE estimator. $\square$

This theorem provides a strong justification: if we can find an estimator $\hat{f}_\Theta(x)$ whose error $y - f_\Theta(x)$ is orthogonal to *all* measurable functions of $x$, then $f_\Theta(x)$ is indeed the optimal MMSE estimator. This underpins the EBD framework's objective.

### B.2.2 From orthogonality to hidden units to arbitrary functions: an infinite width perspective

A critical point is that Theorem B.1 requires the estimation error to be orthogonal to *any* measurable function $g(x)$ of the input. However, the EBD algorithm, as practically implemented, enforces orthogonality of the output error $e_\Theta(x) = y - f_\Theta(x)$ to the activations of the network's hidden units, $h_j^{(k)}(x)$. The question is whether satisfying these more limited orthogonality conditions is sufficient to approach the true MMSE estimator. We argue that in the limit of infinite network width, this can indeed be the case.

The argument relies on the universal approximation capabilities of wide neural networks:

- Rahimi and Recht showed that a hidden layer with random i.i.d. Gaussian weights and appropriately chosen biases (e.g., uniform for Fourier features) can linearly approximate functions within a corresponding Reproducing Kernel Hilbert Space (RKHS) $\mathcal{H}_k$ associated with a shift-invariant kernel $k$ [51, 52]. Specifically, for a function in $\mathcal{H}_k$, $N$ such hidden units (random features) can achieve an expected $L_2(P_X)$ approximation error of order $1/\sqrt{N}$, assuming the input distribution $P_X$ has compact support.
- Building on this, Sun et.al. analyzed random ReLU networks, where hidden units are of the form $\text{ReLU}(w^\top x)$ with $w \sim \mathcal{N}(0, I)$ [53]. They demonstrated that the RKHS induced by the corresponding random feature kernel is dense in $L_2(P_X)$ under mild conditions on $P_X$. This establishes the universality of random ReLU features for approximating any square-integrable function with the linear combination of that hidden layer units.

Our initial setting for the EBD framework—specifically, a first hidden layer with ReLU activations initialized with random i.i.d. Gaussian weights—is essentially the same as the random ReLU feature setting analyzed in [53]. Although weights change during training, one might hypothesize that in the infinite width limit and with sufficiently controlled learning rates (to limit the deviation of weights from their initial distribution), the set of first hidden layer activations $\{h_i^{(1)}(x)\}_{i=1}^{N^{(1)}}$ could retain its universal approximation capability. This is an area for future rigorous analysis.

Under this crucial assumption that the (potentially trained) first hidden layer activations $\{h_i^{(1)}(x)\}_{i=1}^{N^{(1)}}$ can form a basis that is dense in $L_2(P_X)$ as $N^{(1)} \to \infty$, we can state the following result concerning an estimator that achieves perfect orthogonality with these activations.

**Theorem B.2** (Convergence to MMSE for Estimators Orthogonal to a Dense Basis of First-Layer Activations). *Let $f_\Theta(x)$ be an estimator for $y$ given $x$. Assume its error $e_\Theta(x) = y - f_\Theta(x)$ satisfies the orthogonality condition with respect to a set of $N^{(1)}$ first-layer hidden unit activations $\{h_i^{(1)}(x)\}_{i=1}^{N^{(1)}}$:*

$$\langle y - f_\Theta(x), h_i^{(1)}(x)\rangle = 0, \quad for \ i = 1, \ldots, N^{(1)}. \tag{22}$$

*Further assume that the linear span of these activations $\{h_i^{(1)}(x)\}_{i=1}^{N^{(1)}}$ is dense in $L_2(P_X)$ as $N^{(1)} \to \infty$. That is, for any $g(x) \in L_2(P_X)$ and any $\varepsilon > 0$, there exists a sufficiently large $N^{(1)}$ and a linear combination $\hat{g}^{(1)}(x) = \sum_{i=1}^{N^{(1)}} c_i h_i^{(1)}(x)$ such that:*

$$\left\| g(x) - \hat{g}^{(1)}(x) \right\|_{L_2(P_X)} \le \varepsilon. \tag{23}$$

*Then, as $\varepsilon \to 0$ (corresponding to the infinite width limit where $N^{(1)} \to \infty$), $f_\Theta(x)$ converges to the optimal MMSE estimator $f_{MMSE}(x)$ in the $L_2(P_X)$ sense:*

$$\|f_{MMSE}(x) - f_\Theta(x)\|_{L_2(P_X)} \to 0.$$

*Proof.* Consider the inner product between the error of the estimator $f_\Theta(x)$, $e_\Theta(x) = y - f_\Theta(x)$, and an arbitrary function $g(x) \in L_2(P_X)$. We can write:

$$\langle y - f_\Theta(x), g(x)\rangle = \langle y - f_\Theta(x), g(x) - \hat{g}^{(1)}(x) + \hat{g}^{(1)}(x)\rangle$$
$$= \langle y - f_\Theta(x), g(x) - \hat{g}^{(1)}(x)\rangle + \langle y - f_\Theta(x), \hat{g}^{(1)}(x)\rangle.$$

The second term, $\langle y - f_\Theta(x), \hat{g}^{(1)}(x)\rangle = \left\langle y - f_\Theta(x), \sum_{j=1}^{N^{(1)}} c_j h_j^{(1)}(x)\right\rangle = \sum_{j=1}^{N^{(1)}} c_j \langle y - f_\Theta(x), h_j^{(1)}(x)\rangle$, is zero due to the assumed orthogonality condition (22). Thus, we are left with the first term. Applying the Cauchy-Schwarz inequality and using the denseness assumption Eq. (23):

$$|\langle y - f_\Theta(x), g(x)\rangle| = |\langle y - f_\Theta(x), g(x) - \hat{g}^{(1)}(x)\rangle|$$
$$\le \|y - f_\Theta(x)\|_{L_2(P_X)} \|g(x) - \hat{g}^{(1)}(x)\|_{L_2(P_X)}$$
$$\le \|e_\Theta(x)\|_{L_2(P_X)} \cdot \varepsilon. \tag{24}$$

This shows that as $\varepsilon \to 0$, the error $e_\Theta(x)$ becomes orthogonal to any $g(x) \in L_2(P_X)$, i.e., $\langle y - f_\Theta(x), g(x)\rangle \to 0$.

Now, let $f_{MMSE}(x)$ be the true MMSE estimator. By definition, its error $e_{MMSE}(x) = y - f_{MMSE}(x)$ is orthogonal to any $g(x) \in L_2(P_X)$, so $\langle y - f_{MMSE}(x), g(x) \rangle = 0$. We can rewrite the left side of Eq. (24) as:

$$\begin{aligned} \langle y - f_\Theta(x), g(x) \rangle &= \langle (y - f_{MMSE}(x)) + (f_{MMSE}(x) - f_\Theta(x)), g(x) \rangle \\ &= \langle y - f_{MMSE}(x), g(x) \rangle + \langle f_{MMSE}(x) - f_\Theta(x), g(x) \rangle \\ &= 0 + \langle f_{MMSE}(x) - f_\Theta(x), g(x) \rangle \\ &= \langle f_{MMSE}(x) - f_\Theta(x), g(x) \rangle. \end{aligned}$$

Substituting this back into the inequality Eq. (24):

$$|\langle f_{MMSE}(x) - f_\Theta(x), g(x) \rangle| \leq \|e_\Theta(x)\|_{L_2(P_X)} \cdot \varepsilon. \tag{25}$$

This inequality holds for any $g(x) \in L_2(P_X)$. We can choose $g(x) = f_{MMSE}(x) - f_\Theta(x)$ (assuming this difference is in $L_2(P_X)$). This yields:

$$\|f_{MMSE}(x) - f_\Theta(x)\|^2_{L_2(P_X)} \leq \|e_\Theta(x)\|_{L_2(P_X)} \cdot \varepsilon.$$

As $\varepsilon \to 0$ (corresponding to $N^{(1)} \to \infty$ under the denseness assumption), and assuming the error of the estimator $f_\Theta(x)$, $\|e_\Theta(x)\|_{L_2(P_X)}$, remains bounded, the right-hand side approaches zero. Therefore:

$$\|f_{MMSE}(x) - f_\Theta(x)\|^2_{L_2(P_X)} \to 0,$$

which implies $f_\Theta(x) \to f_{MMSE}(x)$ in the $L_2(P_X)$ sense. $\qquad\square$

**Discussion of Theorem B.2:** This theorem is significant because it elucidates the properties of an estimator that achieves the specific orthogonality targeted by the EBD algorithm with respect to its first-layer hidden units. Theorem B.1 established that orthogonality to *all* measurable functions is sufficient for MMSE optimality. Theorem B.2 demonstrates that, *if* an estimator's error is perfectly orthogonal to its first-layer hidden unit activations, and *if* these activations form a dense basis (a condition motivated by infinite-width random feature networks), then such an estimator indeed converges to the true MMSE solution. This provides a theoretical rationale for the EBD algorithm's objective: by striving to decorrelate output errors with hidden unit activations, EBD aims to satisfy the premise of this theorem. The result offers a theoretical justification for how achieving orthogonality with respect to a practical, finite set of internal network features can, under idealized conditions of network width and feature richness, lead to overall MMSE optimality. The "meaningfulness" of these orthogonality constraints, even in high-dimensional spaces, is thus linked to the expressive power of the network's learned features. The theorem hinges on the denseness of the feature space generated by the first hidden layer and the perfect satisfaction of the orthogonality conditions. Formalizing the conditions under which EBD effectively approximates these conditions and the extent to which universality of features is preserved during training remain important directions for future research.

### B.2.3 EBD framework objective

The EBD framework designs loss functions that aim to satisfy the orthogonality conditions discussed. Specifically, for a network $f_\Theta(x)$, the target is to achieve:

$$\langle y - f_\Theta(x), h_j^{(k)}(x) \rangle = 0, \quad \text{for all output components, layers } k, \text{ and units } j. \tag{26}$$

By minimizing decorrelation losses based on Eq. (26), particularly for a universal first layer as analyzed in Theorem B.2, the network is guided towards the MMSE optimal solution. The practical EBD algorithm uses empirical estimates of these correlations and updates network weights to minimize their magnitudes.

### B.2.4 Scaling of orthogonality conditions for nonlinear MMSE

The generality of the stochastic orthogonality condition in Eq. (18) for nonlinear estimators allows, in principle, for an even greater expansion of the number of constraints. If the error $e_*(x)$ of the optimal MMSE estimator is orthogonal to any $g(x)$, it is also orthogonal to any function of its own hidden unit activations, $g_m(h_j^{(k)}(x))$, since $h_j^{(k)}(x)$ is itself a function of $x$. Thus, one could enforce:

$$\langle e_{EBD,i}(x), g_m(h_j^{(k)}(x)) \rangle = \mathbb{E}\left[ e_{EBD,i}(x) \cdot g_m(h_j^{(k)}(x)) \right] = 0, \tag{27}$$

for $i = 1, \ldots, p$, $j = 1, \ldots, N^{(k)}$, and for a set of $M$ different nonlinear functions $g_m(\cdot)$. This extension could potentially introduce a greater diversity of updates and more strongly enforce the conditions for MMSE optimality. While this approach theoretically increases the number of constraints and might offer benefits, it also increases computational complexity and has not been pursued in our current numerical experiments. The practical benefit versus the added cost of enforcing orthogonality against more complex functions of hidden units remains an open question. The primary EBD algorithm focuses on $g_m(h_j^{(k)}) = h_j^{(k)}$.

## C  The derivation of update terms

In this section, we present the detailed derivations for the EBD algorithm and its variations, as introduced in Section 2.3.

### C.1  $\Delta \mathbf{W}_1$ and $\Delta \mathbf{b}_1$ calculation

In Section 2.3, we defined the weight update elemet $[\Delta \mathbf{W}_1]_{ij}$ as follows:

$$\zeta Tr\left(\hat{\mathbf{R}}_{\mathbf{g}\boldsymbol{\epsilon}}^{(k)}[m]\mathbf{E}[m]\frac{\partial \mathbf{G}^{(k)}[m]^T}{\partial W_{ij}^{(k)}}\right).$$

The derivative term in this expression can be expanded as

$$\frac{\partial \mathbf{G}^{(k)}[m]}{\partial W_{ij}^{(k)}} = \mathbf{e}_i \begin{bmatrix} g'^{(k)}_i(h_i^{(k)}[mB+1])f'^{(k)}(u_i^{(k)}[mB+1])h_j^{(k-1)}[mB+1] \\ g'^{(k)}_i(h_i^{(k)}[mB+2])f'^{(k)}(u_i^{(k)}[mB+2])h_j^{(k-1)}[mB+2] \\ \vdots \\ g'^{(k)}_i(h_i^{(k)}[(m+1)B])f'^{(k)}(u_i^{(k)}[(m+1)B])h_j^{(k-1)}[(m+1)B] \end{bmatrix}^T,$$

where $\mathbf{e}_i$ represents the standard basis vector with all elements set to zero, except for the element at index $i$, which is equal to 1.

By defining the matrix

$$\boldsymbol{Q}^{(k)}[m] = \hat{\mathbf{R}}_{\mathbf{g}\boldsymbol{\epsilon}}^{(k)}[m]\mathbf{E}[m] = \begin{bmatrix} \mathbf{q}^{(k)}[mB+1] & \cdots & \mathbf{q}^{(k)}[(m+1)B] \end{bmatrix},$$

which represents the projection of the output error onto layer $k$, we can express the weight update as:

$$[\Delta \boldsymbol{W}_1^{(k)}[m]]_{ij} = \zeta \sum_{n=mB+1}^{(m+1)B} g'^{(k)}_i(h_i^{(k)}[n])f'^{(k)}(u_i^{(k)}[n])q_i^{(k)}[n]h_j^{(k-1)}[n].$$

To further simplify this expression, we introduce the matrices:

$$\mathbf{G}_d^{(k)}[m] = \begin{bmatrix} \mathbf{g}^{(k)\prime}(\mathbf{h}^{(k)}[mB+1]) & \mathbf{g}^{(k)\prime}(\mathbf{h}^{(k)}[mB+2]) & \cdots & \mathbf{g}^{(k)\prime}(\mathbf{h}^{(k)}[(m+1)B]) \end{bmatrix}, \quad (28)$$

$$\mathbf{F}_d^{(k)}[m] = \begin{bmatrix} \mathbf{f}^{(k)\prime}(\mathbf{u}^{(k)}[mB+1]) & \mathbf{f}^{(k)\prime}(\mathbf{u}^{(k)}[mB+2]) & \cdots & \mathbf{f}^{(k)\prime}(\mathbf{u}^{(k)}[(m+1)B]) \end{bmatrix}, \quad (29)$$

and $\boldsymbol{Z}^{(k)}[m] = \mathbf{G}_d^{(k)}[m] \odot \mathbf{F}_d^{(k)}[m] \odot \mathbf{Q}^{(k)}[m]$, which allows us to express the weight update in a more compact form:

$$\Delta \boldsymbol{W}_1^{(k)}[m] = \zeta \boldsymbol{Z}^{(k)}[m]\boldsymbol{H}^{(k-1)}[m]^T.$$

Following a similar procedure, the bias update is given by:

$$\Delta \mathbf{b}_1^{(k)}[m] = \zeta \boldsymbol{Z}^{(k)}[m]\mathbf{1}_{B\times 1}.$$

## C.2   $\Delta\mathbf{W}_2$ and $\Delta\mathbf{b}_2$ calculation

In Section 2.3, we defined the weight update element $[\Delta\mathbf{W}_2]_{ij}$ involving the derivative of the output error as

$$\zeta Tr\left(\hat{\mathbf{R}}_{\mathbf{g}\boldsymbol{\epsilon}}^{(k)}[m]\frac{\partial\mathbf{E}[m]}{\partial W_{ij}^{(k)}}\mathbf{G}^{(k)}[m]^T\right).$$

To begin, we consider the derivative term:

$$\frac{\partial\boldsymbol{\epsilon}}{\partial W_{ij}^{(k)}},$$

which can be expanded as

$$\frac{\partial\boldsymbol{\epsilon}}{\partial W_{ij}^{(k)}}=\underbrace{\frac{\partial\boldsymbol{\epsilon}}{\partial\mathbf{h}^{(L)}}}_{\mathbf{I}}\underbrace{\frac{\partial\mathbf{h}^{(L)}}{\partial\mathbf{u}^{(L)}}}_{\mathrm{diag}(f'^{(k)}(\mathbf{u}^{(L)}))}\underbrace{\frac{\partial\mathbf{u}^{(L)}}{\partial\mathbf{h}^{(L-1)}}}_{\mathbf{W}^{(L)}}\underbrace{\frac{\partial\mathbf{h}^{(L-1)}}{\partial\mathbf{u}^{(L-1)}}}_{\mathrm{diag}(f'^{(k)}(\mathbf{u}^{(L-1)}))}\cdots$$

$$\cdots\underbrace{\frac{\partial\mathbf{h}^{(k+1)}}{\partial\mathbf{u}^{(k+1)}}}_{\mathrm{diag}(f'^{(k)}(\mathbf{u}^{(k+1)}))}\underbrace{\frac{\partial\mathbf{u}^{(k+1)}}{\partial\mathbf{h}^{(k)}}}_{\mathbf{W}^{(k+1)}}\underbrace{\frac{\partial\mathbf{h}^{(k)}}{\partial\mathbf{u}^{(k)}}}_{\mathrm{diag}(f'^{(k)}(\mathbf{u}^{(k)}))}\underbrace{\frac{\partial\mathbf{u}^{(k)}}{\partial W_{ij}^{(k)}}}_{\mathbf{e}_i h_j^{(k-1)}}$$

This expression reflects propagation terms, from the output back to the layer $k$. Defining $\Phi^{(L)}[n]=\mathrm{diag}(f^{(L)'}(\mathbf{u}^{(L)}[n]))$, and

$$\Phi^{(k)}[n]=\Phi^{(k+1)}[n]\mathbf{W}^{(k+1)}[m]\mathrm{diag}(f'^{(k)}(\mathbf{u}^{(k)}[n])),$$

we obtain

$$\frac{\partial\boldsymbol{\epsilon}[n]}{\partial W_{ij}^{(k)}}=\Phi^{(k)}[n]h_j^{(k-1)}[n]\mathbf{e}_i.$$

Thus, the derivative of the error at time step $n$ with respect to $W_{ij}^{(k)}$ can be written as:

$$\zeta Tr\left(\mathbf{R}_{\mathbf{g}\boldsymbol{\epsilon}}^{(k)}[m]\frac{\partial\mathbf{E}[m]}{\partial W_{ij}^{(k)}}\mathbf{G}^{(k)}[m]^T\right)=$$

$$\zeta Tr\left(\mathbf{R}_{\mathbf{g}\boldsymbol{\epsilon}}^{(k)}[m]\sum_{n=mB+1}^{(m+1)B}\frac{\partial\boldsymbol{\epsilon}[n]}{\partial W_{ij}^{(k)}}\mathbf{g}^{(k)}(h^{(k)}[n])^T\right).$$

Substituting the definition $\tilde{\mathbf{g}}^{(k)}[n]=\mathbf{R}_{\mathbf{g}\boldsymbol{\epsilon}}^{(k)}[m]^T\mathbf{g}^{(k)}(h^{(k)}[n])$, we obtain:

$$\zeta Tr\left(\mathbf{R}_{\mathbf{g}(\mathbf{h}^{(k)})\boldsymbol{\epsilon}}[m]\frac{\partial\mathbf{E}[m]}{\partial W_{ij}^{(k)}}\mathbf{G}^{(k)}[m]^T\right),$$

$$=\zeta Tr\left(\sum_{n=mB+1}^{(m+1)B}h_j^{(k-1)}[n]\Phi^{(k)}[n]\mathbf{e}_i\tilde{\mathbf{g}}^{(k)}[n]^T\right),$$

$$=\zeta\sum_{n=mB+1}^{(m+1)B}\mathbf{e}_j^T\mathbf{h}^{(k-1)}[n]\tilde{\mathbf{g}}^{(k)}[n]^T\Phi^{(k)}[n]\mathbf{e}_i,$$

$$=\mathbf{e}_i^T\left(\zeta\sum_{n=mB+1}^{(m+1)B}\Phi^{(k)}[n]^T\tilde{\mathbf{g}}^{(k)}[n]\mathbf{h}^{(k-1)}[n]^T\right)\mathbf{e}_j.$$

Now, defining:

$$\psi^{(k)}[n]=\Phi^{(k)}[n]^T\tilde{\mathbf{g}}^{(k)}[n],$$

and assembling these into the matrix:

$$\mathbf{\Psi}^{(k)}[m] = \begin{bmatrix} \psi^{(k)}[mB+1] & \psi^{(k)}[mB+2] & \ldots & \psi^{(k)}[(m+1)B] \end{bmatrix},$$

we can compactly express the weight and bias updates as:

$$\begin{aligned}
\Delta\mathbf{W}_2^{(k)}[m] &= \zeta\mathbf{\Psi}^{(k)}[m]\mathbf{H}^{(k-1)}[m]^T, \\
\Delta\mathbf{b}_2^{(k)}[m] &= \zeta\mathbf{\Psi}^{(k)}[m]\mathbf{1}_{B\times 1}.
\end{aligned}$$

## C.3 Update corresponding to the layer entropy regularization

In Section 2.4.1, we introduced the layer entropy objective as

$$J_E^{(k)}(\mathbf{h}^{(k)})[m] = \frac{1}{2}\log\det(\mathbf{R}_{\mathbf{h}}^{(k)}[m] + \varepsilon^{(k)}\mathbf{I}) \tag{30}$$

where,

$$\begin{aligned}
\mathbf{R}_{\mathbf{h}}^{(k)}[m] &= \lambda_E\mathbf{R}_{\mathbf{h}}^{(k)}[m-1] + (1-\lambda_E)\frac{1}{B}\mathbf{H}^{(k)}[m]\mathbf{H}^{(k)}[m]^T, \quad \text{and,} \\
\mathbf{H}^{(k)}[m] &= \begin{bmatrix} \mathbf{h}^{(k)}[mB+1] & .. & \mathbf{h}^{(k)}[(m+1)B] \end{bmatrix}.
\end{aligned} \tag{31}$$

The derivative of the entropy objective in Eq. (30) with respect to $W_{ij}$ is given by

$$\begin{aligned}
\frac{\partial J_E^{(k)}[m]}{\partial W_{ij}^{(k)}} &= Tr\left(\nabla_{\mathbf{R}_{\mathbf{h}}^{(k)}[m]+\varepsilon^{(k)}\mathbf{I}}J_E[m] \cdot \frac{\partial(\mathbf{R}_{\mathbf{h}}^{(k)}[m]+\varepsilon^{(k)}\mathbf{I})}{\partial W_{ij}^{(k)}}\right) \\
&= \frac{2(1-\lambda_E)}{B}Tr\left((\mathbf{R}_{\mathbf{h}}^{(k)}[m]+\varepsilon^{(k)}\mathbf{I})^{-1}\mathbf{H}^{(k)}[m]\frac{\partial\mathbf{H}^{(k)}[m]}{\partial W_{ij}^{(k)}}^T\right)
\end{aligned}$$

In this expression, the derivative term can be explicitly written as

$$\begin{aligned}
\frac{\partial\mathbf{H}^{(k)}[m]}{\partial W_{ij}^{(k)}} &= \mathbf{F}_d^{(k)}[m] \odot \frac{\partial(\mathbf{W}^{(k)}\mathbf{H}^{(k-1)}[m]+\mathbf{b}^{(k)})}{\partial W_{ij}^{(k)}} \\
&= \mathbf{F}_d^{(k)}[m] \odot (\mathbf{e}_i\mathbf{e}_j^T\mathbf{H}^{(k-1)}[m]) \\
&= \mathbf{e}_i(\mathbf{F}_{d_{i,:}}^{(k)}[m] \odot \mathbf{H}_{j,:}^{(k-1)}[m])
\end{aligned}$$

$$\begin{aligned}
\frac{\partial J_E^{(k)}[m]}{\partial W_{ij}^{(k)}} &= \frac{2(1-\lambda_E)}{B}Tr\left((\mathbf{R}_{\mathbf{h}}^{(k)}[m]+\varepsilon^{(k)}\mathbf{I})^{-1}\mathbf{H}^{(k)}[m](\mathbf{F}_{d_{i,:}}^{(k)}[m] \odot \mathbf{H}_{j,:}^{(k-1)}[m])^T\mathbf{e}_i^T\right) \\
&= \frac{2(1-\lambda_E)}{B}Tr\left((((\mathbf{R}_{\mathbf{h}}^{(k)}[m]+\varepsilon^{(k)}\mathbf{I})^{-1}\mathbf{H}^{(k)}[m]) \odot \mathbf{F}_d^{(k)}[m])\mathbf{H}^{(k-1)}[m]^T\mathbf{e}_j\mathbf{e}_i^T\right) \\
&= \frac{2(1-\lambda_E)}{B}\mathbf{e}_i^T(((\mathbf{R}_{\mathbf{h}}^{(k)}[m]+\varepsilon^{(k)}\mathbf{I})^{-1}\mathbf{H}^{(k)}[m]) \odot \mathbf{F}_d^{(k)}[m])\mathbf{H}^{(k-1)}[m]^T\mathbf{e}_j \\
&= \frac{2(1-\lambda_E)}{B}\left[(((\mathbf{R}_{\mathbf{h}}^{(k)}[m]+\varepsilon^{(k)}\mathbf{I})^{-1}\mathbf{H}^{(k)}[m]) \odot \mathbf{F}_d^{(k)}[m])\mathbf{H}^{(k-1)}[m]^T\right]_{ij}
\end{aligned}$$

Consequently, we obtain

$$\nabla_{\mathbf{W}^{(k)}}J_E^{(k)}[m] = \frac{2(1-\lambda_E)}{B}(((\mathbf{R}_{\mathbf{h}}^{(k)}[m]+\varepsilon^{(k)}\mathbf{I})^{-1}\mathbf{H}^{(k)}[m]) \odot \mathbf{F}_d^{(k)}[m])\mathbf{H}^{(k-1)}[m]^T. \tag{32}$$

Through a similar derivation, we also obtain

$$\nabla_{\mathbf{b}^{(k)}}J_E^{(k)}[m] = \frac{2(1-\lambda_E)}{B}(((\mathbf{R}_{\mathbf{h}}^{(k)}[m]+\varepsilon^{(k)}\mathbf{I})^{-1}\mathbf{H}^{(k)}[m]) \odot \mathbf{F}_d^{(k)}[m])\mathbf{1}_{B\times 1}. \tag{33}$$

## C.4 Update corresponding to the power normalization regularization

In Section 2.4.1, we introduced power normalization objective as,

$$J_P^{(k)}(\mathbf{h}^{(k)})[m] = \sum_{l=1}^{N^{(k)}} \left( \frac{1}{B} \sum_{n=mB+1}^{(m+1)B} h_l^{(k)}[n]^2 - P^{(k)} \right)^2.$$

The derivative of this objective with respect to $W_{ij}^{(k)}$ can be written as

$$\frac{\partial J_P^{(k)}[m]}{W_{ij}^{(k)}} = 4 \underbrace{\left( \frac{1}{B} \sum_{n=mB+1}^{(m+1)B} h_i^{(k)}[n]^2 - P^{(k)} \right)}_{d_i[m]} \left( \frac{1}{B} \sum_{n=mB+1}^{(m+1)B} h_i^{(k)}[n] f^{(k)'}(u^{(k)}[n]) \frac{\partial h_i^{(k)}[n]}{\partial W_{ij}^{(k)}} \right)$$

$$= 4 d_i[m] \left( \frac{1}{B} \sum_{n=mB+1}^{(m+1)B} h_i^{(k)}[n] f^{(k)'}(u^{(k)}[n]) h_j^{(k-1)}[n] \right).$$

Therefore, the gradient of the power-normalization objective with respect to $\mathbf{W}^{(k)}$ can be written as

$$\nabla_{\mathbf{W}^{(k)}} J_P^{(k)}[m] = \frac{4}{B} \mathbf{D}[m](\mathbf{H}^{(k)}[m] \odot \mathbf{F}_d^{(k)}) \mathbf{H}^{(k-1)}[m]^T, \tag{34}$$

where $\mathbf{D}[m] = \mathrm{diag}(d_1[m], d_2[m], \dots, d_{N^{(k)}}[m])$. The gradient with respect to $\mathbf{b}^{(k)}$ can be obtained in a similar way as

$$\nabla_{\mathbf{b}^{(k)}} J_P^{(k)}[m] = \frac{4}{B} \mathbf{D}[m](\mathbf{H}^{(k)}[m] \odot \mathbf{F}_d^{(k)}) \mathbf{1}_{B \times 1}. \tag{35}$$

## C.5 On the EBD with forward projections

In the EBD algorithm introduced in Section 2.3 , output errors are broadcast to individual layers to modify their weights, thereby reducing the correlation between hidden layer activations and output errors. To enhance this mechanism, we introduce forward broadcasting, where hidden layer activations are projected onto the output layer. This projection facilitates the optimization of the decorrelation loss by adjusting the parameters of the final layer more effectively.

The purpose of forward broadcasting is to enhance the network's ability to minimize the decorrelation loss by directly influencing the final layer's weights using the activations from the hidden layers. By projecting the hidden layer activations forward onto the output layer, we establish a direct pathway for these activations to impact the adjustments of the final layer's weights. This mechanism allows the final layer to update its parameters in a way that reduces the correlation between the output errors and the hidden layer activations. Consequently, the errors at the output layer are steered toward being orthogonal to the hidden layer activations.

This mechanism could potentially be effective because the final layer is responsible for mapping the network's internal representations to the output space. By incorporating information from earlier layers, we enable the final layer to align its parameters more closely with the features that are most relevant for reducing the overall error.

While the proposed forward broadcasting mechanism is primarily motivated by performance optimization, it can conceptually be related to the long-range [54] and bottom-up [55] synaptic connections in the brain, which allow certain neurons to influence distant targets. These long-range bottom-up connections are actively being researched, and incorporating similar mechanisms into computational models could enhance their alignment with biological neural processes. By integrating mechanisms that mirror these neural pathways, forward broadcasting may be useful for modeling how information is transmitted across different neural circuits.

### C.5.1 Gradient derivation for the EBD with forward projections

We derive the gradients of the layer decorrelation losses with respect to the parameters of the final layer. The partial derivative of the objective function $J^{(k)}(\mathbf{h}^{(k)}, \boldsymbol{\epsilon})$ with respect to the final layer

weights can be written as:

$$
\begin{aligned}
\frac{\partial J^{(k)}(\mathbf{h}^{(k)}, \boldsymbol{\epsilon})}{\partial W_{ij}^{(L)}}[m] &= \zeta Tr\left(\hat{\mathbf{R}}_{\mathbf{g}(\mathbf{h}^{(k)})\boldsymbol{\epsilon}}[m]\frac{\partial(\mathbf{E}[m]\mathbf{G}^{(k)}[m]^T)}{\partial W_{ij}^{(L)}}\right) \\
&= \underbrace{\zeta Tr\left(\hat{\mathbf{R}}_{\mathbf{g}(\mathbf{h}^{(k)})\boldsymbol{\epsilon}}[m]\frac{\partial \mathbf{E}[m]}{\partial W_{ij}^{(L)}}\mathbf{G}^{(k)}[m]^T\right)}_{[\Delta\mathbf{W}^{(L,k),f}[m]]_{ij}}, \\
&= \zeta Tr\left(\mathbf{R}_{\mathbf{g}(\mathbf{h}^{(k)})\boldsymbol{\epsilon}}[m]\sum_{n=mB+1}^{(m+1)B}\frac{\partial\boldsymbol{\epsilon}[n]}{\partial W_{ij}^{(L)}}\mathbf{g}(h^{(k)}[n])^T\right).
\end{aligned}
$$

Substituting the definition $\tilde{\mathbf{g}}^{(k)}[n] = \mathbf{R}_{\mathbf{g}(\mathbf{h}^{(k)})\boldsymbol{\epsilon}}[m]^T\mathbf{g}(h^{(k)}[n])$, we can further express the partial derivative as:

$$
\begin{aligned}
\frac{\partial J^{(k)}(\mathbf{h}^{(k)}, \boldsymbol{\epsilon})}{\partial W_{ij}^{(L)}}[m] &= \zeta Tr\left(\sum_{n=mB+1}^{(m+1)B} h_j^{(L-1)}[n]\Phi^{(L)}[n]\mathbf{e}_i\tilde{\mathbf{g}}^{(k)}[n]^T\right), \\
&= \zeta\sum_{n=mB+1}^{(m+1)B}\mathbf{e}_j^T\mathbf{h}^{(L-1)}[n]\tilde{\mathbf{g}}^{(k)}[n]^T\Phi^{(L)}[n]\mathbf{e}_i, \\
&= \mathbf{e}_i^T\left(\zeta\sum_{n=mB+1}^{(m+1)B}\Phi^{(L)}[n]^T\tilde{\mathbf{g}}^{(k)}[n]\mathbf{h}^{(L-1)^T}\right)\mathbf{e}_j, \\
&= \mathbf{e}_i^T\left(\zeta\sum_{n=mB+1}^{(m+1)B}(f'(\mathbf{u}^{(L)}[n])\odot\tilde{\mathbf{g}}^{(k)}[n])\mathbf{h}^{(L-1)^T}\right)\mathbf{e}_j.
\end{aligned}
$$

Next, defining the following terms:

$$
\psi^{(k,L)}[n] = f'(\mathbf{u}^{(L)}[n])\odot\tilde{\mathbf{g}}^{(k)}[n],
$$

and assembling them into the matrix:

$$
\mathbf{\Psi}^{(k,L)}[m] = \left[\begin{array}{cccc} \psi^{(k,L)}[mB+1] & \psi^{(k,L)}[mB+2] & \dots & \psi[(m+1)B]\end{array}\right],
$$

we can write the weight update as:

$$
\Delta\mathbf{W}^{(L,k),f}[m] = \zeta\mathbf{\Psi}^{(k,L)}[m]\mathbf{H}^{(k-1)}[m]^T.
$$

Following a similar procedure, the bias update can be written as:

$$
\Delta\mathbf{b}^{(L,k),f}[m] = \zeta\mathbf{\Psi}^{(k,L)}[m]\mathbf{1}_{B\times 1}.
$$

Based on these expressions, we can write

$$
[\Delta\mathbf{W}^{(L,k),f}[m]]_{ij} = \zeta\sum_{n=mB+1}^{(m+1)B}f^{(L)'}(u_i^{(L)}[n])\tilde{g}_i^{(k)}[n]\mathbf{h}_j^{(L-1)}
$$

$$
[\Delta\mathbf{b}^{(L,k),f}[m]]_i = \zeta\sum_{n=mB+1}^{(m+1)B}f^{(L)'}(u_i^{(L)}[n])\tilde{g}_i^{(k)}[n].
$$

# D  Additional extensions of EBD

## D.1  Extensions to Convolutional Neural Networks (CNNs)

Let $\mathbf{H}^{(k)} \in \mathbb{R}^{P^{(k)} \times M^{(k)} \times N^{(k)}}$ represent the output of the $k^{th}$ layer of a Convolutional Neural Network (CNN), where $P^{(k)}$ is the number of channels and the layer's output is $M^{(k)} \times N^{(k)}$ dimensional. Furthermore, we use $\mathbf{W}^{(k,p)} \in \mathbb{R}^{P^{(k-1)} \times \Omega^{(k)} \times \Omega^{(k)}}$ and $\mathbf{b}^{(k,p)} \in \mathbb{R}$ to represent the filter tensor weights and bias coefficient respectively for the channel-$p$ of the $k^{th}$ layer, and $\Omega^{(k)}$ is the symmetric convolution kernel size. Then a convolutional layer can be defined as

$$\mathbf{H}^{(k,p)} = f(\mathcal{U}^{(k,p)}), \tag{36}$$

$$\mathcal{U}^{(k,p)} = (\mathbf{H}^{(k-1)} * \mathbf{W}^{(k,p)}) + \mathbf{b}^{(k,p)}, \tag{37}$$

where the symbol "$*$" represents the convolution [1] operation that acts upon both the spatial and channel dimensions to generate the $p^{th}$ channel of $k^{th}$ layer output $\mathbf{H}^{(k,p)}$, and $f$ is the nonlinearity acted on the convolution output.

### D.1.1  Error Broadcast and Decorrelation formulation

Similar to Eq. (3), we have the cross-correlation between output errors $\epsilon$ and the arbitrary function of the $k^{th}$ layer activation of the $p^{th}$ channel denoted as $\mathbf{g}^{(k)}(\mathbf{H}^{(k,p)})$, for each layer and spatial indexes $r \in \mathbb{Z} : 1 \leq r \leq M^{(k)}$ and $s \in \mathbb{Z} : 1 \leq s \leq N^{(k)}$ as

$$\mathbf{R}_{\mathbf{g}^{(k)}(\mathbf{H}^{(k,p)})\epsilon}[q, r, s] = \mathbb{E}[\mathbf{g}^{(k)}(\mathbf{H}^{(k,p)}[r, s])\epsilon_q] = \mathbf{0}. \tag{38}$$

Then this cross-correlation must ideally be zero due to the stochastic orthogonality condition. We can then write the loss for layer-$k$ at batch-$m$ as:

$$J^{(k)}(\mathbf{H}^{(k,p)}, \boldsymbol{\epsilon})[m] = \frac{1}{2} \sum_{q=1}^{n_c} \left\| \hat{\mathbf{R}}_{\mathbf{g}^{(k)}(\mathbf{H}^{(k,p)})\epsilon}[m, q, :, :] \right\|_F^2, \tag{39}$$

where $\hat{\mathbf{R}}_{\mathbf{g}(\mathbf{H}^{(k),p})\epsilon}$ is the recurrently estimated cross-correlation using the training batches. Then we can optimize the network by taking the derivative of the loss function with respect to the weight $\mathbf{W}_{hij}^{(k,p)}$ corresponding to input channel $h$ and weight spatial indexes $i, j \in \mathbb{Z} : 1 \leq i, j \leq \Omega^{(k)}$ as

$$
\begin{aligned}
&\frac{\partial J^{(k)}(\mathbf{H}^{(k,p)}, \boldsymbol{\epsilon})[m]}{\partial \mathbf{W}_{hij}^{(k,p)}} \\
&= \zeta \sum_{q=1}^{n_c} \sum_{n=mB+1}^{(m+1)B} \epsilon_q[n] \cdot Tr\left( (\hat{\mathbf{R}}_{\mathbf{g}^{(k)}(\mathbf{H}^{(k,p)})\epsilon}[m, q, :, :]^T \frac{\partial \mathbf{g}^{(k)}(\mathbf{H}^{(k,p)}[n, :, :])}{\partial \mathbf{W}_{hij}^{(k,p)}} \right) \\
&= \zeta \sum_{q=1}^{n_c} \sum_{n=mB+1}^{(m+1)B} \sum_{r,s} \epsilon_q[n] \left[ (\hat{\mathbf{R}}_{\mathbf{g}^{(k)}(\mathbf{H}^{(k,p)})\epsilon}[m, q, :, :] \odot \frac{\partial \mathbf{g}^{(k)}(\mathbf{H}^{(k,p)}[n, :, :])}{\partial \mathbf{W}_{hij}^{(k,p)}} \right]_{[r,s]},
\end{aligned}
\tag{40}
$$

in which $n_c$ is the error dimension, $N^{(k)}$ and $M^{(k)}$ are the width and height of the $k^{th}$ layer, and the derivative with respect to the $\epsilon$ term is neglected. The inner partial derivative term can be written as

$$\frac{\partial \mathbf{g}^{(k)}(\mathbf{H}^{(k,p)}[n, :, :])}{\partial \mathbf{W}_{hij}^{(k,p)}} = \mathbf{g}'^{(k)}(\mathbf{H}^{(k,p)}[n, :, :]) \odot \frac{\partial \mathbf{H}^{(k,p)}[n, :, :]}{\partial \mathbf{W}_{hij}^{(k,p)}}, \tag{41}$$

and using the Eq. (36) and Eq. (37),

$$\frac{\partial \mathbf{H}^{(k,p)}[n, :, :]}{\partial \mathbf{W}_{hij}^{(k,p)}} = f'(\mathcal{U}^{(k,p)}[n, :, :]) \odot (\mathcal{E}_{hij}^{(k)} * \mathbf{H}^{(k-1)}[n, :, :]). \tag{42}$$

---

[1] Although we call it as convolution, in CNNs, the actual operation used is the correlation operation where the kernel is unflipped.

where $\mathcal{E}_{hij}^{(k)} \in \mathbb{R}^{P^{(k-1)} \times \Omega^{(k)} \times \Omega^{(k)}}$ is a Kronecker delta tensor that occurs as the gradient of $\mathbf{W}^{(k,p)}$ with respect to $\mathbf{W}_{hij}^{(k,p)}$. Combining the expressions, we have

$$\phi[n,p,:,:] = \sum_{q=1}^{n_c} \epsilon_q[n] \cdot \left( \hat{\mathbf{R}}_{\mathbf{g}^{(k)}(\mathbf{H}^{(k,p)})\boldsymbol{\epsilon}}[n,q,:,:] \odot \mathbf{g}^{(k)}(\mathbf{H}^{(k,p)}[n,:,:]) \odot f'(\mathcal{U}^{(k,p)}[n,:,:]) \right). \quad (43)$$

Then, combining the Equations (40), (41), (42), and then writing the convolution explicitly, we have

$$\frac{\partial J^{(k)}(\mathbf{H}^{(k,p)}, \boldsymbol{\epsilon})[m]}{\partial \mathbf{W}_{hij}^{(k,p)}} = \zeta \sum_{n=mB+1}^{(m+1)B} \sum_{r,s} \left[ \phi[n,p,:,:] \odot (\mathcal{E}_{hij}^{(k)} * \mathbf{H}^{(k-1)}[n,:,:]) \right]_{[r,s]}$$

$$= \zeta \sum_{n=mB+1}^{(m+1)B} \sum_{r,s} \phi[n,p,r,s] \cdot \left( \sum_{h',i',j'} \mathcal{E}_{hij}^{(k)}[h',i',j'] \cdot \mathbf{H}^{(k-1,h')}[n, r+i', s+j'] \right).$$

By the definition of the delta function $\mathcal{E}_{hij}^{(k)}$ and writing the resulting expression as a 2D convolution between $\mathbf{H}^{(k-1)}$ and $\phi$ respectively, we have

$$= \zeta \sum_{n=mB+1}^{(m+1)B} \sum_{r,s} \phi[n,p,r,s] \cdot \mathbf{H}^{(k-1,h)}[n, r+i, s+j]$$

$$= \zeta \sum_{n=mB+1}^{(m+1)B} \left[ \phi[n,p,:,:] * \mathbf{H}^{(k-1,h)}[n,:,:] \right]_{[i,j]}.$$

The resulting expression for the weight update is:

$$\frac{\partial J^{(k)}(\mathbf{H}^{(k,p)}, \boldsymbol{\epsilon})[m]}{\partial \mathbf{W}_h^{(k,p)}} = \zeta \sum_{n=mB+1}^{(m+1)B} (\phi[n,p,:,:] * \mathbf{H}^{(k-1,h)}[n,:,:]). \quad (44)$$

Similarly, it can be shown that the bias update:

$$\frac{\partial J^{(k)}(\mathbf{H}^{(k,p)}, \boldsymbol{\epsilon})[m]}{\partial \mathbf{b}^{(k,p)}} = \zeta \sum_{n=mB+1}^{(m+1)B} \sum_{r=1}^{N^{(k)}} \sum_{s=1}^{M^{(k)}} \phi[n,p,r,s].$$

The convolutional layer parameters can be trained using these gradient formulas for each layer separately, and can be calculated by utilizing the batched convolution operation.

### D.1.2 Weight entropy objective

The layer entropy objective is computationally cumbersome for a convolutional layer that has multiple dimensions. Therefore, we propose the weight-entropy objective to avoid dimensional collapse

$$J_E^{(k)}(\mathbf{W}^{(k)}) = \frac{1}{2} \log \det(\mathbf{R}_{\overline{\mathbf{W}}^{(k)}} + \varepsilon^{(k)} \mathbf{I}),$$

where we define $\overline{\mathbf{W}}^{(k)} \in \mathbb{R}^{P^{(k)} \times P^{(k-1)} \cdot \Omega^{(k)} \cdot \Omega^{(k)}}$ as the unraveled version of the full size weight tensor $\mathbf{W}^{(k)}$, and the covariance matrix $\mathbf{R}_{\overline{\mathbf{W}}^{(k)}}$ is conditionally defined as:

$$\mathbf{R}_{\overline{\mathbf{W}}^{(k)}} = \begin{cases} \overline{\mathbf{W}}^{(k)T} \overline{\mathbf{W}}^{(k)}, & \text{if } P^{(k)} \geq P^{(k-1)} \cdot \Omega^{(k)} \cdot \Omega^{(k)}, \\ \overline{\mathbf{W}}^{(k)} \overline{\mathbf{W}}^{(k)T}, & \text{otherwise,} \end{cases}$$

to decrease its dimensions and reduce the computational costs for further steps. Then, the derivative of this objective can be written as:

$$\Delta J_E^{(k)}(\mathbf{W}^{(k)}) = \begin{cases} \overline{\mathbf{W}}^{(k)} \mathbf{R}_{\overline{\mathbf{W}}^{(k)}}^{-1}, & \text{if } P^{(k)} \geq P^{(k-1)} \cdot \Omega^{(k)} \cdot \Omega^{(k)}, \\ \mathbf{R}_{\overline{\mathbf{W}}^{(k)}}^{-1} \overline{\mathbf{W}}^{(k)}, & \text{otherwise.} \end{cases}$$

Therefore, $\frac{\partial J_E(\mathbf{W}^{(k)})}{\partial W_{hij}^{(k,p)}}$ can be obtained by reshaping $\Delta J_E^{(k)}(\mathbf{W}^{(k)})$ as the weight tensor $\mathbf{W}^{(k)}$.

### D.1.3 Activation sparsity regularization

To further regularize the model, we enforce the layer activation sparsity loss that is given as

$$J_{\ell_1}^{(k)}(\mathbf{H}^{(k,p)}) = \frac{\|\mathbf{H}^{(k,p)}\|_1}{|\mathbf{H}^{(k,p)}\|_2}. \tag{45}$$

The gradient of the sparsity loss with respect to the hidden layer can be written as:

$$\Delta J_{\ell_1}^{(k)}(\mathbf{H}^{(k,p)}) = \frac{1}{\|\mathbf{H}^{(k,p)}\|_2}\text{sign}(\mathbf{H}^{(k,p)}) - \frac{\|\mathbf{H}^{(k,p)}\|}{\|\mathbf{H}^{(k,p)}\|_2^3}\mathbf{H}^{(k,p)}. \tag{46}$$

Then, the gradient of the loss with respect to the model weights can be calculated in a similar manner as the Eq. (44):

$$\frac{\partial J_{\ell_1}^{(k)}(\mathbf{H}^{(k,p)})[m]}{\partial \mathbf{W}_h^{(k,p)}} = \frac{1}{B}\sum_{n=mB+1}^{(m+1)B}\left(\Delta J_{\ell_1}^{(k)}(\mathbf{H}^{(k,p)})[n,p,:,:] * \mathbf{H}^{(k-1,h)}[n,:,:]\right).$$

### D.2 Extensions to Locally Connected (LC) Networks

Let $\mathbf{H}^{(k)} \in \mathbb{R}^{P^{(k)} \times M^{(k)} \times N^{(k)}}$ represent the output of the $k^{th}$ layer of a Locally Connected Network (LC), where $P^{(k)}$ is the number of channels and the layer's output is $M^{(k)} \times N^{(k)}$ dimensional. We use $\mathbf{W}^{(k,p,r,s)} \in \mathbb{R}^{P^{(k-1)} \times \Omega^{(k)} \times \Omega^{(k)}}$ and $\mathbf{b}^{(k,p,r,s)} \in \mathbb{R}$ to represent the filter tensor weights and bias coefficient at spatial locations $r \in \mathbb{Z}: 1 \leq r \leq M^{(k)}$ and $s \in \mathbb{Z}: 1 \leq s \leq N^{(k)}$, for channel-$p$ of the $k^{th}$ layer, where $\Omega^{(k)}$ is the local receptive field size. Then a locally connected layer can be defined as

$$\mathbf{H}^{(k,p)} = f(\mathcal{U}^{(k,p)}), \tag{47}$$
$$\mathcal{U}^{(k,p)} = (\mathbf{H}^{(k-1)} \circledast \mathbf{W}^{(k,p)}) + \mathbf{b}^{(k,p)}, \tag{48}$$

where the symbol "$\circledast$" represents the locally connected operation which acts upon both the spatial and channel dimensions, but without weight sharing across spatial locations, generating the $p^{th}$ channel of the $k^{th}$ layer output $\mathbf{H}^{(k,p)}$, and $f$ is the nonlinearity applied to the result.

#### D.2.1 Error Broadcast and Decorrelation formulation

For the LC network, the stochastic orthogonality condition and the corresponding loss $J^{(k)}(\mathbf{H}^{(k,p)}, \boldsymbol{\epsilon})[m]$ for layer-$k$ at batch-$m$ can be written equivalently as Eq. (38) and Eq. (39) respectively. Then the optimization can be performed by taking the derivative of the loss function with respect to $\mathbf{W}_{hij}^{(k,p,r,s)}$ corresponding to input channel $h$, weight spatial indexes $i, j \in \mathbb{Z}: 1 \leq i, j \leq \Omega^{(k)}$ as

$$\frac{\partial J^{(k)}(\mathbf{H}^{(k,p)}, \boldsymbol{\epsilon})[m]}{\partial \mathbf{W}_{hij}^{(k,p,r,s)}}$$
$$= \zeta \sum_{q=1}^{n_c} \sum_{n=mB+1}^{(m+1)B} \epsilon_q[n] \cdot Tr\left((\hat{\mathbf{R}}_{\mathbf{g}^{(k)}(\mathbf{H}^{(k,p)})\boldsymbol{\epsilon}}[m,q,:,:]^T \frac{\partial \mathbf{g}^{(k)}(\mathbf{H}^{(k,p)}[n,:,:])}{\partial \mathbf{W}_{hij}^{(k,p,r,s)}}\right) \tag{49}$$
$$= \zeta \sum_{q=1}^{n_c} \sum_{n=mB+1}^{(m+1)B} \sum_{r,s} \epsilon_q[n] \left[(\hat{\mathbf{R}}_{\mathbf{g}^{(k)}(\mathbf{H}^{(k,p)})\boldsymbol{\epsilon}}[m,q,:,:] \odot \frac{\partial \mathbf{g}^{(k)}(\mathbf{H}^{(k,p)}[n,:,:])}{\partial \mathbf{W}_{hij}^{(k,p,r,s)}}\right]_{[r,s]}.$$

The inner partial derivative term can be written as

$$\frac{\partial \mathbf{g}^{(k)}(\mathbf{H}^{(k,p)}[n,:,:])}{\partial \mathbf{W}_{hij}^{(k,p,r,s)}} = \mathbf{g}'^{(k)}(\mathbf{H}^{(k,p)}[n,:,:]) \odot \frac{\partial \mathbf{H}^{(k,p)}[n,:,:]}{\partial \mathbf{W}_{hij}^{(k,p,r,s)}}, \tag{50}$$

and using Eq. (47) and Eq. (48), we obtain:

$$\frac{\partial \mathbf{H}^{(k,p)}[n,:,:]}{\partial \mathbf{W}_{hij}^{(k,p,r,s)}} = f'(\mathcal{U}^{(k,p)}[n,:,:]) \odot (\mathcal{E}_{hij}^{(k)} \circledast \mathbf{H}^{(k-1)}[n,:,:]). \tag{51}$$

Here, $\mathcal{E}_{hij}^{(k,r,s)} \in \mathbb{R}^{P^{(k-1)} \times \Omega^{(k)} \times \Omega^{(k)}} \times M^{(k)} \times N^{(k)}$ is a Kronecker delta tensor that occurs as the gradient of $\mathbf{W}^{(k,p)}$ with respect to $\mathbf{W}_{hij}^{(k,p,r,s)}$. Combining the expressions in Eq. (49), Eq. (50), Eq. (51), and the expression for $\phi$ as in Eq. (43) which is equivalent for both CNNs and LCs, and then writing the locally connected operation explicitly, we have

$$\frac{\partial J^{(k)}(\mathbf{H}^{(k,p)}, \boldsymbol{\epsilon})[m]}{\partial \mathbf{W}_{hij}^{(k,p,r,s)}} = \zeta \sum_{n=mB+1}^{(m+1)B} \sum_{r,s} \left[ \phi[n,p,:,:] \odot (\mathcal{E}_{hij}^{(k,r,s)} \circledast \mathbf{H}^{(k-1)}[n,:,:]) \right]_{[r,s]}$$

$$= \zeta \sum_{n=mB+1}^{(m+1)B} \phi[n,p,r,s] \cdot \left( \sum_{\substack{h',i',j' \\ r',s'}} \mathcal{E}_{hij}^{(k,r,s)}[h',i',j',r',s'] \cdot \mathbf{H}^{(k-1,h')}[n,r'+i',s'+j'] \right).$$

Then, by the definition of the Kronecker delta, the resulting expression for the weight update is:

$$\frac{\partial J^{(k)}(\mathbf{H}^{(k,p)}, \boldsymbol{\epsilon})[m]}{\partial \mathbf{W}_{hij}^{(k,p,r,s)}} = \zeta \sum_{n=mB+1}^{(m+1)B} \left( \phi[n,p,r,s] \cdot \mathbf{H}^{(k-1,h)}[n,r+i,s+j] \right). \tag{52}$$

Similarly, it can be shown that the bias update is:

$$\frac{\partial J^{(k)}(\mathbf{H}^{(k,p)}, \boldsymbol{\epsilon})[m]}{\partial \mathbf{b}^{(k,p,r,s)}} = \zeta \sum_{n=mB+1}^{(m+1)B} \phi[n,p,r,s].$$

### D.2.2 Weight entropy objective

Similar to CNNs, we propose the weight-entropy objective to avoid dimensional collapse in LCs

$$J_E^{(k)}(\mathbf{W}^{(k)}) = \frac{1}{2} \log \det(\mathbf{R}_{\overline{\mathbf{W}}^{(k)}} + \varepsilon^{(k)} \mathbf{I}),$$

where we define $\overline{\mathbf{W}}^{(k)} \in \mathbb{R}^{P^{(k)} \times P^{(k-1)} \cdot M^{(k)} \cdot N^{(k)} \cdot \Omega^{(k)} \cdot \Omega^{(k)}}$ as the unraveled version of the full size weight tensor $\mathbf{W}^{(k)}$, then the covariance matrix $\mathbf{R}_{\overline{\mathbf{W}}^{(k)}}$ is defined as:

$$\mathbf{R}_{\overline{\mathbf{W}}^{(k)}} = \overline{\mathbf{W}}^{(k)T} \overline{\mathbf{W}}^{(k)}.$$

Then, the derivative of this objective can be written as:

$$\Delta J_E^{(k)}(\mathbf{W}^{(k)}) = \overline{\mathbf{W}}^{(k)} \mathbf{R}_{\overline{\mathbf{W}}^{(k)}}^{-1}$$

$\frac{\partial J_E(\mathbf{W}^{(k)})}{\partial W_{hij}^{(k,p,r,s)}}$ can be obtained by reshaping $\Delta J_E^{(k)}(\mathbf{W}^{(k)})$ as the weight tensor $\mathbf{W}^{(k)}$.

### D.2.3 Activation sparsity regularization

The layer activation sparsity loss for the LC is the same as the one given for the CNN in Eq. (45), with its gradient with respect to the activations as in Eq. (46). Then, the gradient of the loss with respect to the model weights can be calculated in a similar manner as the expression in Eq. (52):

$$\frac{\partial J_{\ell_1}^{(k)}(\mathbf{H}^{(k)})[m]}{\partial \mathbf{W}_{hij}^{(k,p,r,s)}} = \frac{1}{B} \sum_{n=mB+1}^{(m+1)B} \left( \Delta J_{\ell_1}^{(k)}(\mathbf{H}^{(k)})[n,p,r,s] \circledast \mathbf{H}^{(k-1,h)}[n,r+i,s+j] \right).$$

# E    Gradient alignment in EBD

## E.1    Alignment between EBD updates and backpropagation gradients

To investigate the relationship between the EBD update directions and the gradients produced by backpropagation (BP), we analyze the cosine similarity between the EBD update vectors and the corresponding BP gradients throughout training. This analysis quantifies how well the EBD learning dynamics align with traditional gradient-based optimization methods.

We conduct experiments on two architectures: a 3-layer multilayer perceptron (MLP) and a locally connected (LC) network, both trained on CIFAR-10. Figure 3 and Figure 4 illustrate the cosine similarity between EBD updates and BP gradients at each training epoch.

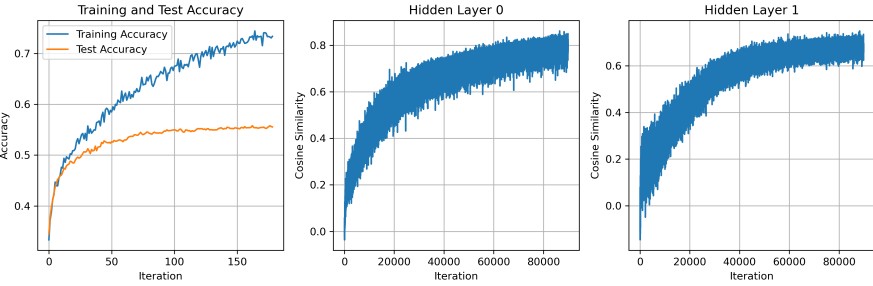

Figure 3: Cosine similarity between EBD updates and backpropagation gradients in a 3-layer MLP trained on CIFAR-10. Alignment is consistently positive and improves during training.

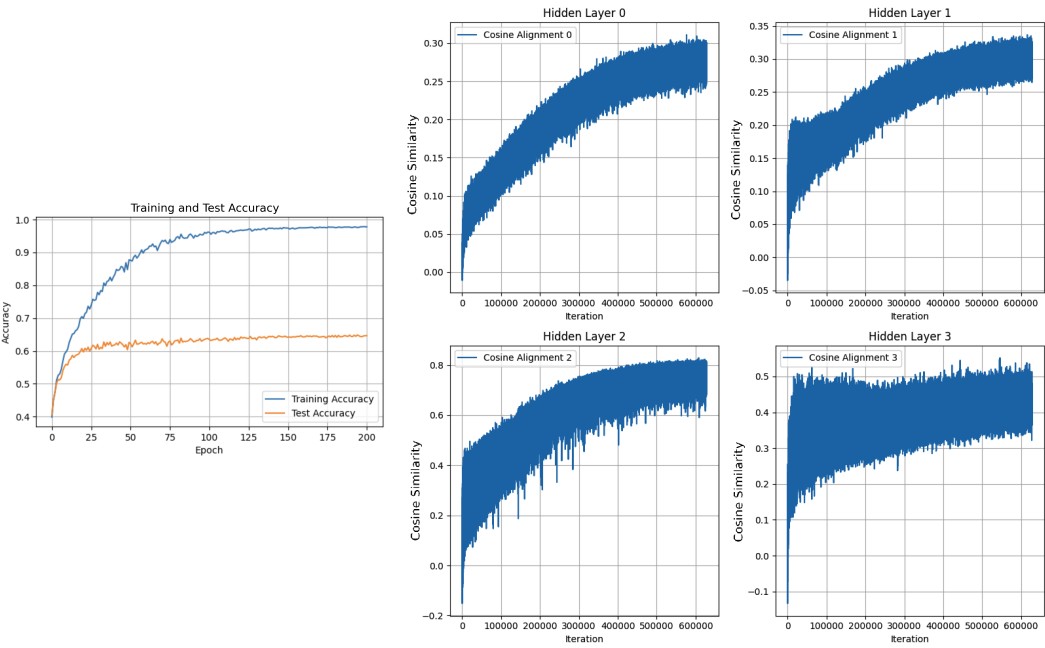

Figure 4: Cosine similarity between EBD updates and backpropagation gradients in a locally connected network on CIFAR-10. Positive alignment indicates directional consistency between EBD and BP.

These results demonstrate that EBD update directions are not arbitrary but align with the descent direction of the loss function as indicated by BP, supporting its effectiveness as a gradient-free but principled optimization strategy.

## E.2 On gradient truncation and biological plausibility

The decorrelation objective used in EBD naturally decomposes into two sets of parameter updates per layer $k$:

$$(\Delta W_1^{(k)}, \Delta b_1^{(k)}) \quad \text{and} \quad (\Delta W_2^{(k)}, \Delta b_2^{(k)}).$$

Here, $(\Delta W_1^{(k)}, \Delta b_1^{(k)})$ corresponds to updates that modify the hidden representation to reduce correlation with the output error, while $(\Delta W_2^{(k)}, \Delta b_2^{(k)})$ corresponds to updates that aims to reshape the error signal itself.

For reasons of local learning and biological plausibility, EBD retains only the $(\Delta W_1^{(k)}, \Delta b_1^{(k)})$ component and drops the error-shaping terms $(\Delta W_2^{(k)}, \Delta b_2^{(k)})$, thereby avoiding the backward propagation of gradients through the network.

To assess the impact of this truncation, we measured the cosine similarity between the full gradient (which includes both components) and the truncated update used in EBD. As shown in Figure 5, the truncated update direction remains consistently aligned with the full gradient throughout training on CIFAR-10 using a 3-layer MLP. This positive alignment suggests that the retained component is sufficient for effective learning, validating our simplification.

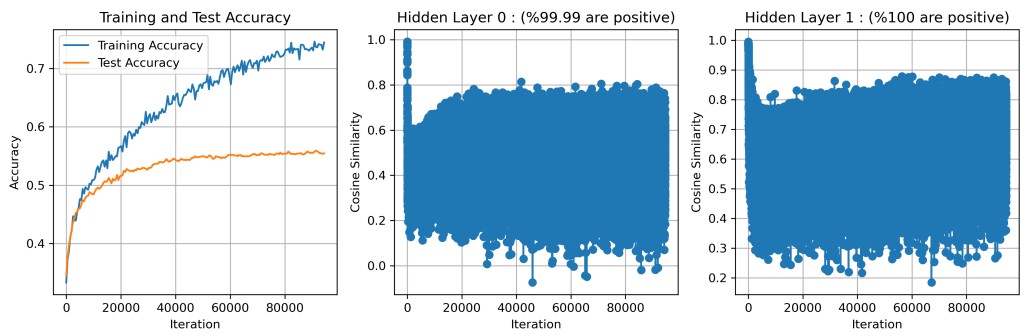

Figure 5: Cosine similarity between the full decorrelation gradient (including $(\Delta W_2^{(k)}, \Delta b_2^{(k)})$) and the truncated EBD update (only $(\Delta W_1^{(k)}, \Delta b_1^{(k)})$). Positive similarity confirms that the truncated update remains a valid descent direction.

## F    Background on online Correlative Information Maximization (CorInfoMax) based biologically plausible neural networks

Bozkurt et.al. recently proposed a framework, which we refer as CorInfoMax-EP, to address weight symmetry problem corresponding to backpropagation algorithm Bozkurt et al. [45]. In this section, we provide a brief summary of this framework.

The CorInfoMax-EP framework utilizes an online optimization setting to maximize correlative information between two consequitive layers:

$$\sum_{k=0}^{L-1} \hat{I}^{(\epsilon)}(\mathbf{h}^{(k)}, \mathbf{h}^{(k+1)})[m] - \frac{\beta}{2}\|\mathbf{y}[m] - \mathbf{h}^{(L)}[m]\|_2^2,$$

where $\hat{I}^{(\epsilon)}(\mathbf{h}^{(k)}, \mathbf{h}^{(k+1)})[m]$ is the correlative mutual information between layers $k$ and $k+1$, and the term on the right corresponds to the mean square error between the network output $\mathbf{h}^{(L)}[m]$ and the training label $\mathbf{y}[m]$. This framework utilizes two alternative but equivalent forms for the correlative mutual information

$$\hat{I}_r^{(\varepsilon_k)}(\mathbf{h}^{(k)}, \mathbf{h}^{(k+1)})[m] = \frac{1}{2}\log\det(\hat{\mathbf{R}}_{\mathbf{h}^{(k+1)}}[m] + \varepsilon_k \boldsymbol{I}) - \frac{1}{2}\log\det(\hat{\boldsymbol{R}}_{\overrightarrow{\mathbf{e}}_*^{(k+1)}}[m] + \varepsilon_k \boldsymbol{I}),$$

$$\hat{I}_l^{(\varepsilon_k)}(\mathbf{h}^{(k)}, \mathbf{h}^{(k+1)})[m] = \frac{1}{2}\log\det(\hat{\mathbf{R}}_{\mathbf{h}^{(k)}}[m] + \varepsilon_k \boldsymbol{I}) - \frac{1}{2}\log\det(\hat{\boldsymbol{R}}_{\overleftarrow{\mathbf{e}}_*^{(k)}}[m] + \varepsilon_k \boldsymbol{I}),$$

defined in terms of the correlation matrices of layer activations, i.e., $\hat{\mathbf{R}}_{\mathbf{h}^{(k)}}$ and the correlation matrices of forward and backward prediction errors ($\hat{\boldsymbol{R}}_{\overrightarrow{\mathbf{e}}_*^{(k+1)}}$ and $\hat{\boldsymbol{R}}_{\overleftarrow{\mathbf{e}}_*^{(k)}}$) between two consequitive layers.

Here, forward/backward prediction errors are defined by

$$\overrightarrow{\mathbf{e}}_*^{(k+1)}[n] = \mathbf{h}^{(k+1)}[n] - \mathbf{W}^{(f,k)}[m]\mathbf{h}^{(k)}[n], \qquad \overleftarrow{\mathbf{e}}_*^{(k)}[n] = \mathbf{h}^{(k)}[n] - \mathbf{W}^{(b,k)}[m]\mathbf{h}^{(k+1)}[n],$$

respectively. Here, $\mathbf{W}^{(f,k)}[m]$ ($\mathbf{W}^{(b,k)}[m]$) is the forward (backward) prediction matrix for layer $k$.

In order to enable online implementation, the exponentially weighted correlation matrices for hidden layer activations and prediction errors are defined as follows:

$$\hat{\mathbf{R}}_{\mathbf{h}^{(k)}}[m] = \frac{1-\lambda}{1-\lambda^m} \sum_{i=1}^{m} \lambda^{m-i} \mathbf{h}^{(k)}[m]\mathbf{h}^{(k)}[m]^T,$$

$$\hat{\mathbf{R}}_{\overrightarrow{\mathbf{e}}^{(k)}}[m] = \frac{1-\lambda}{1-\lambda^m} \sum_{i=1}^{m} \lambda^{m-i} \overrightarrow{\mathbf{e}}^{(k)}[m]\overrightarrow{\mathbf{e}}^{(k)}[m]^T,$$

$$\hat{\mathbf{R}}_{\overleftarrow{\mathbf{e}}^{(k)}}[m] = \frac{1-\lambda}{1-\lambda^m} \sum_{i=1}^{m} \lambda^{m-i} \overleftarrow{\mathbf{e}}^{(k)}[m]\overleftarrow{\mathbf{e}}^{(k)}[m]^T.$$

Through the trace approximation of $\log\det(\cdot)$ function, we obtain:

$$\log\det\left(\hat{\boldsymbol{R}}_{\overrightarrow{\mathbf{e}}^{(k+1)}}[m] + \varepsilon \boldsymbol{I}\right)$$

$$\approx \frac{1}{\varepsilon_k} \sum_{i=1}^{t} \lambda^{t-i}\|\mathbf{h}^{(k+1)}[i] - \mathbf{W}_{ff,*}^{(k)}[m]\mathbf{h}^{(k)}[i]\|_2^2 + \varepsilon_k\|\boldsymbol{W}_{ff,*}^{(k)}[m]\|_F^2 + N_{k+1}\log(\varepsilon_k)$$

$$\log\det\left(\hat{\boldsymbol{R}}_{\overleftarrow{\mathbf{e}}^{(k)}}[m] + \varepsilon_k \boldsymbol{I}\right)$$

$$\approx \frac{1}{\varepsilon_k} \sum_{i=1}^{t} \lambda^{t-i}\|\mathbf{h}^{(k)}[i] - \mathbf{W}_{fb,*}^{(k)}[m]\mathbf{h}^{(k+1)}[i]\|_2^2 + \varepsilon_k\|\boldsymbol{W}_{fb,*}^{(k)}[m]\|_F^2 + N_k\log(\varepsilon_k),$$

### F.1    The derivation of the CorInfoMax network

Based on the definitions above, the following layerwise objectives can be defined:

$$\hat{J}_k(\mathbf{h}^{(k)})[m] = \hat{I}_r^{(\epsilon_{k-1})}(\mathbf{h}^{(k-1)}, \mathbf{h}^{(k)})[m] + \hat{I}_l^{(\varepsilon_k)}(\mathbf{h}^{(k)}, \mathbf{h}^{(k+1)})[m], \text{ for } k = 1, \ldots, L-1,$$

i.e., correlative information maximization objectives for the hidden layers, and the mixture of correlation maximization and MSE objectives for the final layer

$$\hat{J}_L(\mathbf{h}^{(L)})[m] = \hat{I}_r^{(\epsilon_{L-1})}(\mathbf{h}^{(L-1)}, \mathbf{h}^{(L)})[m] - \frac{\beta}{2}\|\mathbf{h}^{(L)}[m] - \mathbf{y}[m]\|_2^2.$$

The gradient of the hidden layer objective functions with respect to the corresponding layer activations can be written as:

$$\nabla_{\mathbf{h}^{(k)}}\hat{J}_k(\mathbf{h}^{(k)})[m] = 2\gamma \boldsymbol{B}_{\mathbf{h}^{(k)}}[m]\mathbf{h}^{(k)}[m] - \frac{1}{\epsilon_{k-1}}\overrightarrow{\mathbf{e}}^{(k)}[m] - \frac{1}{\varepsilon_k}\overleftarrow{\mathbf{e}}^{(k)}[m], \tag{53}$$

where $\gamma = \frac{1-\lambda}{\lambda}$, and $\boldsymbol{B}_{\mathbf{h}^{(k)}}[m] = (\hat{\mathbf{R}}_{\mathbf{h}^{(k)}}[m] + \epsilon_{k-1}\boldsymbol{I})^{-1}$, i.e., the inverse of the layer correlation matrix.

For the output layer, we can write the gradient as

$$\nabla_{\mathbf{h}^{(L)}}\hat{J}_L(\mathbf{h}^{(L)})[m] = \gamma \boldsymbol{B}_{\mathbf{h}^{(L)}}[m]\mathbf{h}^{(L)}[m] - \frac{1}{\epsilon_{L-1}}\overrightarrow{\mathbf{e}}^{(L)}[m] - \beta(\mathbf{h}^{(L)}[m] - \boldsymbol{y}[m]).$$

The gradient ascent updates corresponding to these expressions can be organized to obtain CorInfo-Max network dynamics:

$$\tau_{\mathbf{u}}\frac{d\mathbf{u}^{(k)}[m;s]}{ds} = -g_{lk}\mathbf{u}^{(k)}[m;s] + \frac{1}{\varepsilon_k}\boldsymbol{M}^{(k)}[m]\mathbf{h}^{(k)}[m;s] - \frac{1}{\epsilon_{k-1}}\overrightarrow{\mathbf{e}}_u^{(k)}[m;s] - \frac{1}{\epsilon_k}\overleftarrow{\mathbf{e}}_u^{(k)}[m;s],$$

$$\overrightarrow{\mathbf{e}}_u^{(k)}[m;s] = \mathbf{u}^{(k)}[m;s] - \boldsymbol{W}_{ff}^{(k-1)}[t]\mathbf{h}^{(k-1)}[m;s],$$

$$\overleftarrow{\mathbf{e}}_u^{(k)}[m;s] = \mathbf{u}^{(k)}[m;s] - \boldsymbol{W}_{fb}^{(k)}[m]\mathbf{h}^{(k+1)}[m;s],$$

$$\mathbf{h}^{(k)}[m;s] = \sigma_+(\mathbf{u}^{(k)}[m;s]),$$

where $m$ is the sample index, $s$ is the time index for the network dynamics, $\tau_{\mathbf{u}}$ is the update time constant, $\boldsymbol{M}^{(k)}[t] = \varepsilon_k(2\gamma \boldsymbol{B}_{\mathbf{h}^{(k)}}[t] + g_{lk}\boldsymbol{I})$, and $\sigma_+ = \min(1, \max(u, 0))$ represents the elementwise clipped-ReLU function, which is the projection operation corresponding to the combination of the nonnegativity constraint $\mathbf{h}^{(k)} \geq 0$ and the boundedness constraint $\|\mathbf{h}^{(k)}\|_\infty \leq 1$ on the activations of the network.

Note that Bozkurt et al. [45] takes one more step to organize the network dynamics into a form that fits into the form of a network with three compartment (soma, basal dendrite and appical dendrite compartments) neuron model.

## F.2 CorInfoMax-EP learning dynamics

The CorInfoMax-EP framework in Bozkurt et al. [45] employs equilibrium propagation (EP) to update feedforward and feedback weights of the CorInfoMax network.

### F.2.1 Feedforward and feedback weights

In the CorInfoMax objective, feedforward and feedback weights correspond to forward and backward predictors corresponding to the regularized least squares objectives

$$C_{ff}(\boldsymbol{W}_{ff}^{(k)}[m]) = \varepsilon_k\|\boldsymbol{W}_{ff}^{(k)}[m]\|_F^2 + \|\overrightarrow{\mathbf{e}}^{(k+1)}[m]\|_2^2,$$

and

$$C_{fb}(\boldsymbol{W}_{fb}^{(k)}[m]) = \varepsilon_k\|\boldsymbol{W}_{ff}^{(k)}[m]\|_F^2 + \|\overleftarrow{\mathbf{e}}^{(k)}[m]\|_2^2,$$

respectively. The derivatives of these functions with respect to forward and backward synaptic weights can be written as

$$\frac{\partial C_{ff}(\boldsymbol{W}_{ff}^{(k)}[m])}{\partial \boldsymbol{W}_{ff}^{(k)}[m]} = 2\varepsilon_k\boldsymbol{W}_{ff}^{(k)}[m] - 2\overrightarrow{\mathbf{e}}^{(k+1)}[m]\mathbf{h}^{(k)}[m]^T,$$

and

$$\frac{\partial C_{fb}(\boldsymbol{W}_{fb}^{(k)}[m])}{\partial \boldsymbol{W}_{fb}^{(k)}[m]} = 2\varepsilon_k \boldsymbol{W}_{fb}^{(k)}[m] - 2\overleftarrow{\mathbf{e}}^{(k)}[m]\mathbf{h}^{(k+1)}[m]^T.$$

The EP based updates of the feedforward and feedback weights are obtained by evaluating these gradients in two different phases: the nudge phase ($\beta = \beta' > 0$), and the free phase ($\beta = 0$):

$$\delta\boldsymbol{W}_{ff}^{(k)}[m] \propto \frac{1}{\beta'}\left(\left.(\overrightarrow{\mathbf{e}}^{(k+1)}[m]\mathbf{h}^{(k)}[m]^T)\right|_{\beta=\beta'} - \left.(\overrightarrow{\mathbf{e}}^{(k+1)}[m]\mathbf{h}^{(k)}[m]^T)\right|_{\beta=0}\right),$$

$$\delta\boldsymbol{W}_{fb}^{(k)}[m] \propto \frac{1}{\beta'}\left(\left.(\overleftarrow{\mathbf{e}}^{(k)}[m]\mathbf{h}^{(k+1)}[m]^T)\right|_{\beta=\beta'} - \left.(\overleftarrow{\mathbf{e}}^{(k)}[m]\mathbf{h}^{(k+1)}[m]^T)\right|_{\beta=0}\right).$$

### F.2.2   Lateral weights

The lateral weight updates derived from the weight correlation matrices of the layer activations, using the matrix inversion lemma [43]:

$$\boldsymbol{B}^{(k)}[m+1] = \lambda_{\mathbf{r}}^{-1}(\boldsymbol{B}^{(k)}[m] - \gamma\mathbf{z}^{(k)}[m]\mathbf{z}^{(k)}[m]^T), \text{ where } \mathbf{z}^{(k)}[m] = \boldsymbol{B}^{(k)}[m]\mathbf{h}^{(k)}[m]\big|_{\beta=\beta'}.$$

### F.3   CorInfoMax-EP

Although the CorInfoMax-EP algorithm derivation above is based on single input sample based updates, it can be extendable to batch updates. Assuming a batch size of $B$, and we define the following matrices:

$$\mathbf{H}^{(k)}[m] = \begin{bmatrix} \mathbf{h}^{(k)}[mB+1] & \mathbf{h}^{(k)}[mB+2] & \dots & \mathbf{h}^{(k)}[(m+1)B] \end{bmatrix},$$

as the activation matrix for the layer-$k$,

$$\overleftarrow{\mathbf{E}}^{(k)}[m] = \begin{bmatrix} \overleftarrow{\mathbf{e}}^{(k)}[mB+1] & \overleftarrow{\mathbf{e}}^{(k)}[mB+2] & \dots & \overleftarrow{\mathbf{e}}^{(k)}[(m+1)B] \end{bmatrix}, \tag{54}$$

as the backward prediction matrix for the layer-$k$,

$$\overrightarrow{\mathbf{E}}^{(k)}[m] = \begin{bmatrix} \overrightarrow{\mathbf{e}}^{(k)}[mB+1] & \overrightarrow{\mathbf{e}}^{(k)}[mB+2] & \dots & \overrightarrow{\mathbf{e}}^{(k)}[(m+1)B] \end{bmatrix}, \tag{55}$$

as the forward prediction matrix for the layer-$k$,

$$\mathbf{Z}^{(k)}[m] = \begin{bmatrix} \mathbf{z}^{(k)}[mB+1] & \mathbf{z}^{(k)}[mB+2] & \dots & \mathbf{z}^{(k)}[(m+1)B] \end{bmatrix}, \tag{56}$$

as the lateral weights' output matrix for the layer-$k$, and

$$\mathbf{E} = \begin{bmatrix} \boldsymbol{\epsilon}[mB+1] & \boldsymbol{\epsilon}[mB+2] & \dots & \boldsymbol{\epsilon}[(m+1)B] \end{bmatrix},$$

as the output error matrix.

In terms of these definitions, Algorithm 2 lays out the details of the CorInfoMax-EP algorithm:

**Algorithm 2** CorInfoMax Equilibrium Propagation (CorInfoMax-EP) Update for Layer $k$

---

**Require:** Learning rate parameters $\lambda_E$, $\mu^{(f,k)}[m]$, $\mu^{(b,k)}[m]$
**Require:** Previous synaptic weights $\mathbf{W}^{(f,k)}[m-1]$ (forward), $\mathbf{W}^{(b,k)}[m-1]$ (backward), $\mathbf{B}^{(k)}$ (lateral)
**Require:** Batch size $B$
**Require:** Layer activations $\mathbf{H}^{(k)}[m]$, preactivations $\mathbf{U}^{(k)}[m]$, output errors $\mathbf{E}^{(k)}[m]$, lateral weight outputs $\mathbf{Z}^{(k)}[m]$, forward prediction errors $\overset{\rightarrow}{\mathbf{E}}^{(k)}[m]$ and backward prediction errors $\overset{\leftarrow}{\mathbf{E}}^{(k)}[m]$ computed by CorInfoMax network dynamics described in Bozkurt et al. [45]
**Ensure:** Updated weights $\mathbf{W}^{(f,k)}[m]$, $\mathbf{W}^{(b,k)}[m]$, $\mathbf{B}^{(k)}[m]$

1: $\gamma_E \leftarrow \frac{1-\lambda_E}{\lambda_E}$

**Update forward weights for layer $k$:**

2: $\Delta \mathbf{W}_{\text{EP}}^{(f,k)}[m] \leftarrow -\frac{\mu^{(d_f,k)}[m]}{B\beta'} \left( \left. (\overset{\rightarrow}{\mathbf{E}}^{(k+1)}[m]\mathbf{H}^{(k)}[m]^T) \right|_{\beta=\beta'} - \left. (\overset{\rightarrow}{\mathbf{E}}^{(k+1)}[m]\mathbf{H}^{(k)}[m]^T) \right|_{\beta=0} \right)$

3: $\mathbf{W}^{(f,k)}[m] \leftarrow \mathbf{W}^{(f,k)}[m-1] + \Delta \mathbf{W}_{\text{EP}}^{(f,k)}[m]$

**Update backward weights for layer $k$:**

4: $\Delta \mathbf{W}_{\text{EP}}^{(b,k)}[m] \leftarrow -\frac{\mu^{(d_b,k)}[m]}{B\beta'} \left( \left. (\overset{\leftarrow}{\mathbf{E}}^{(k)}[m]\mathbf{H}^{(k)}[m]^T) \right|_{\beta=\beta'} - \left. (\overset{\leftarrow}{\mathbf{E}}^{(k)}[m]\mathbf{H}^{(k)}[m]^T) \right|_{\beta=0} \right)$

5: $\mathbf{W}^{(b,k)}[m] \leftarrow \mathbf{W}^{(b,k)}[m-1] + \Delta \mathbf{W}_{\text{EP}}^{(b,k)}[m]$

**Update Lateral weights for layer $k$:**

6: $\Delta \mathbf{B}_{\text{E}}^{(k)}[m] \leftarrow -\frac{\gamma_E}{B} \mathbf{Z}^{(k)}[m]\mathbf{Z}^{(k)}[m]^T$

7: $\mathbf{B}^{(k)}[m] \leftarrow \frac{1}{\lambda_E}\mathbf{B}^{(k)}[m] + \Delta \mathbf{B}_{\text{E}}^{(k)}[m]$

---

# G Implementation complexity of the EBD approach

In this section, we analyze the computational and memory complexity trade-offs of the proposed Error Broadcast and Decorrelation (EBD) approach.

## G.1 Complexity analysis: Error Propagation vs. Error Broadcast

Considering the standard MLP implementation outlined in Section 2, we compare the memory and computational requirements of the error backpropagation and error broadcast approaches as follows:

### G.1.1 Delivering output error information to layers

- **Memory Requirements:** For the standard backpropagation algorithm, the error is propagated through the transposed forward filters $\mathbf{W}^{(k)}$. Consequently, no additional memory is required for the weights used in error propagation. In contrast, the error broadcast approach uses error projection matrices $\hat{\mathbf{R}}_{\mathbf{g\epsilon}}^{(k)}$, which require memory storage of $\mathcal{O}(N^{(k)}N^{(L)})$ for each layer. Thus, the total additional storage requirement for broadcast weights is given by $\mathcal{O}\left((\sum_{l=1}^{L} N^{(l)})N^{(L)}\right)$.

- **Computational Requirements:** In the standard backpropagation algorithm, propagating the error to layer $k$ (from layer $k+1$) requires $\mathcal{O}(BN^{(k)}N^{(k+1)})$ multiplications per batch, where $B$ is the batch size. On the other hand, the broadcast algorithm projects the output error to layer $k$, requiring $\mathcal{O}(BN^{(k)}N^{(L)})$ multiplications. Additionally, the projection matrix $\hat{\mathbf{R}}_{\mathbf{g\epsilon}}^{(k)}$ is updated at the end of each batch using the Hebbian rule. Therefore, the overall computational complexity for the EBD approach is:

$$\mathcal{O}\left((B+1)N^{(k)}N^{(L)}\right).$$

When the number of output elements $N^{(L)}$ is significantly smaller than the hidden dimensions $N^{(k)}$, the computational cost of the broadcast algorithm is lower. Furthermore, the error projection

operations can be implemented in parallel for all layers, whereas backpropagation must be executed sequentially.

### G.1.2 Additional cost of Entropy Regularization

- **Memory requirements:** As described in Section 2.4.1, layer entropies are based on the covariance matrix $\mathbf{R_h}^{(k)}$, which requires $\mathcal{O}\big((N^{(k)})^2\big)$ additional memory storage. In a computationally optimized implementation, storing the inverse covariance matrix $\mathbf{B_h}^{(k)} = \mathbf{R_h}^{(k)-1}$ may be preferred, though it still requires the same memory allocation.

- **Computational requirements:** The main computational load is due to the computation of the gradient of the layer entropy function. The expression for the gradient is obtained in Appendix C.3 as

$$\nabla_{\mathbf{W}^{(k)}} J^{(k)}[m] = 2\frac{1-\lambda_E}{B} \left[ (\mathbf{R}_h^{(k)}[m] + \epsilon\,\mathbf{I})^{-1}\,\mathbf{H}^{(k)}[m] \odot f'(\mathbf{W}^{(k)}\,\mathbf{H}^{(k-1)}[m] + \mathbf{b}^{(k)}\,\mathbf{1}^T) \right] \big(\mathbf{H}^{(k-1)}[m]\big)^T,$$

where for each batch, we update the layer correlation matrix matrix $\mathbf{R}_h^{(k)}[m]$. We can divide the computational requirement into following pieces:

- *Correlation Matrix Recursion*
  Recall we have

$$\mathbf{R}_h^{(k)}[m] = \lambda_E\,\mathbf{R}_h^{(k)}[m-1] + \frac{1-\lambda_E}{B}\,\mathbf{H}^{(k)}[m]\big(\mathbf{H}^{(k)}[m]\big)^T.$$

  In order to form $\mathbf{R}_h^{(k)}[m]$ explicitly at iteration $m$, we must compute $\mathbf{H}^{(k)}[m]\big(\mathbf{H}^{(k)}[m]\big)^T$ and then add the result to $\lambda_E\,\mathbf{R}_h^{(k)}[m-1]$.

  * The matrix–matrix product $\mathbf{H}^{(k)}[m] \in \mathbb{R}^{N^{(k)}\times B}$ times its transpose in $\mathbb{R}^{B\times N^{(k)}}$ yields an $N^{(k)} \times N^{(k)}$ matrix, costing $\mathcal{O}\big(N^{(k)\,2}\,B\big)$.
  * Adding $\lambda_E\,\mathbf{R}_h^{(k)}[m-1]$ to that product is another $\mathcal{O}\big((N^{(k)})^2\big)$ operation, though usually smaller in comparison to the product above if $B$ is moderate.

  Hence the complexity of *forming the new correlation matrix* $\mathbf{R}_h^{(k)}[m]$ at each iteration is

$$\mathcal{O}\big(N^{(k)\,2}\,B\big).$$

- *Naive Matrix Inversion and Gradient Computation*
  Once $\mathbf{R}_h^{(k)}[m]$ is formed, we need to invert $\mathbf{R}_h^{(k)}[m] + \epsilon\,\mathbf{I}$ to evaluate $J^{(k)}[m]$ and its gradient. Naive inversion of an $N^{(k)} \times N^{(k)}$ matrix is $\mathcal{O}\big((N^{(k)})^3\big)$. After this inversion, we multiply $(\mathbf{R}_h^{(k)}[m] + \epsilon\,\mathbf{I})^{-1}$ by $\mathbf{H}^{(k)}[m] \in \mathbb{R}^{N^{(k)}\times B}$, which costs $\mathcal{O}\big((N^{(k)})^2\,B\big)$. Next, we do the elementwise multiplication ($\odot$) with $f'(\mathbf{W}^{(k)}\,\mathbf{H}^{(k-1)}[m] + \mathbf{b}^{(k)}\mathbf{1}^T)$, costing $\mathcal{O}(N^{(k)}\,B)$. Finally, we multiply by $\big(\mathbf{H}^{(k-1)}[m]\big)^T \in \mathbb{R}^{B\times N^{(k-1)}}$, which costs $\mathcal{O}\big(N^{(k)}\,B\,N^{(k-1)}\big)$.

Summing all these terms, the dominant operations in *naive* update and inversion are:

1. Forming $\mathbf{R}_h^{(k)}[m]$: $\mathcal{O}\big(N^{(k)\,2}\,B\big)$.
2. Inverting $(\mathbf{R}_h^{(k)}[m] + \epsilon\,\mathbf{I})$: $\mathcal{O}\big((N^{(k)})^3\big)$.
3. Multiplying inverse by $\mathbf{H}^{(k)}[m]$: $\mathcal{O}\big((N^{(k)})^2\,B\big)$.
4. Final multiplication by $(\mathbf{H}^{(k-1)}[m])^T$: $\mathcal{O}\big(N^{(k)}\,B\,N^{(k-1)}\big)$.

Thus, overall cost per batch is

$$\mathcal{O}\Big(N^{(k)\,2}\,B + (N^{(k)})^3 + (N^{(k)})^2\,B + N^{(k)}\,B\,N^{(k-1)}\Big),$$

which could be simplified to

$$\mathcal{O}\Big((N^{(k)})^3 + (N^{(k)})^2\,B + N^{(k)}\,B\,N^{(k-1)}\Big).$$

If $N^{(k)}$ is large, the cubic term $(N^{(k)})^3$ associated with the matrix inversion typically dominates.

An alternative is to update the inverse of the correlation matrix *incrementally* using the fact that

$$\mathbf{R}_h^{(k)}[m] \;=\; \lambda_E \, \mathbf{R}_h^{(k)}[m-1] \;+\; \frac{1-\lambda_E}{B} \, \mathbf{H}^{(k)}[m]\big(\mathbf{H}^{(k)}[m]\big)^T,$$

so the new correlation matrix differs from the previous one by a low-rank term of rank at most $\min(N^{(k)}, B)$. Neglecting the $\epsilon$ term, the recursion for the inverse can be obtained using matrix inversion lemma [43] as

$$\mathbf{B}_{\mathbf{h}}^{(k)}[m] = \lambda_E^{-1}\left(\mathbf{B}_{\mathbf{h}}^{(k)}[m-1] - \mathbf{V}[m]\left(\frac{\lambda_E B}{1-\lambda_E}\mathbf{I} + \mathbf{H}^{(k)}[m]^T\mathbf{V}[m]\right)^{-1}\mathbf{V}[m]^T\right), \quad (57)$$

where $\mathbf{V}[m] = \mathbf{B}_{\mathbf{h}}^{(k)}[m-1]\mathbf{H}^{(k)}[m]$.

The corresponding computational cost is:

- Two matrix–matrix multiplications of shape $(N^{(k)} \times N^{(k)})$ with $(N^{(k)} \times B)$, costing $\mathcal{O}((N^{(k)})^2 B)$.
- Inverting the $B \times B$ matrix, costing $\mathcal{O}(B^3)$ if $B$ is not too large.

Thus, the update of the inverse alone costs

$$\mathcal{O}\big((N^{(k)})^2 B + B^3\big),$$

instead of $\mathcal{O}\big((N^{(k)})^3\big)$. If $B \ll N^{(k)}$, this can be a large savings compared to the naive cubic cost. Once this updated inverse is in hand, the subsequent multiplications to form the gradient (e.g. $(\mathbf{R}_h^{(k)}[m] + \epsilon \mathbf{I})^{-1}\mathbf{H}^{(k)}[m]$, etc.) still take $\mathcal{O}((N^{(k)})^2 B + N^{(k)} B N^{(k-1)})$. Overall, for each new time step $m$, the dominant costs become

$$\mathcal{O}\Big((N^{(k)})^2 B + B^3 \;+\; (N^{(k)})^2 B \;+\; N^{(k)} B N^{(k-1)}\Big),$$

which is usually simplified to

$$\mathcal{O}\Big((N^{(k)})^2 B + B^3 + N^{(k)} B N^{(k-1)}\Big).$$

Hence, using the Woodbury identity is beneficial whenever $B$ is much smaller than $N^{(k)}$, because $N^{(k)\,2} B + B^3 \ll (N^{(k)})^3$.

### G.1.3 Additional cost of Power-normalization

- **Memory requirements:** The power normalization described in Section 2.4.1, involves a power estimate parameter per hidden unit, so it will require additional $N^{(k)}$ storage elements for the layer-$k$.

- **Computational requirements:** The gradient for the power normalization regularization function derived in Appendix C.4 takes the form:

$$\nabla_{\mathbf{W}^{(k)}} J_P^{(k)}[m] = \frac{4}{B}\mathbf{D}[m](\mathbf{H}^{(k)}[m] \odot \mathbf{F}_d^{(k)})\mathbf{H}^{(k-1)}[m]^T,$$

Based on this expression, the required number of operations per batch for layer-$k$ is $\mathcal{O}\big(BN^{(k)}N^{(k-1)}\big)$.

Therefore, we can consider the impact of the power-normalization on memory and computational requirements as negligible.

In Section I.7, we provide empirical runtime results for the EBD algorithm, relative to the backpropagation algorithm. These experimental results show a 7 to 8 time increase in the runtimes of the MLP model with the EBD algorithm (employing entropy regularization) relative to the BP algorithm. The runtime increase is less for CNN and LC models.

Finally, we note that the implementation complexity analysis provided above is for the MLP based EBD approach. For the biologically more realistic CorInfoMax-EBD networks, entropy maximization is implemented through lateral weights (see Appendices F, F.2.2 and H), whose update requires $\mathcal{O}\big((N^{(k)})^2\big)$ multiplications per sample.

# H  On the biologically plausible nature of Entropy and Power-normalization updates

As discussed in Section 2.4.1, the layer-entropy and power-normalization objectives are introduced to avert potential collapse of network coefficients in the EBD algorithm. A natural question arises regarding the biological plausibility of the EBD framework when these regularizations are incorporated. We address this question by examining two specific cases:

1. **MLP Implementation with Entropy and Power-normalization regularizations (Section 2)**
2. **CorInfoMax-EBD implementation (Section 3.2)**

## H.1  MLP implementation with Entropy and Power-normalization regularizations

In Section 2, we presented an MLP-based EBD framework that uses batch-SGD to optimize the feedforward weights with EBD, along with entropy and power-normalization losses. As outlined in Sections 2.3 and 3.1, the gradient-based updates of the EBD loss naturally reduce to a three-factor update rule, which is considered biologically plausible. We now examine whether adding the layer-entropy objective in Eq. (11) and the power-normalization objective in Eq. (9) preserves this biological realism.

### H.1.1  Power normalization-based SGD updates

Appendix C.4 derives the gradient expression for the power-normalization loss:

$$\nabla_{\mathbf{W}^{(k)}} J_P^{(k)}[m] = \frac{4}{B}\, \mathbf{D}[m]\big(\mathbf{H}^{(k)}[m] \odot \mathbf{F}_d^{(k)}[m]\big)\mathbf{H}^{(k-1)}[m]^T.$$

Focusing on an individual element of this matrix gives

$$[\nabla_{\mathbf{W}^{(k)}} J_P^{(k)}[m]]_{ij} = \frac{4}{B} d_i[m] \sum_{n=mB+1}^{mB+B} h_i^{(k)}[n]\, f_i^{(k)'}(u_i^{(k)}[n])\, h_j^{(k-1)}[n]. \tag{58}$$

This update depends only on the activations of the neurons connected by the synapse $W_{ij}$, thus satisfying a local learning rule. However, the summation over the batch index in Eq. (58) might be considered biologically implausible unless the batch size $B = 1$. In practice, one can interpret the summation for $B > 1$ as an integral of local updates over the time window corresponding to the batch, which may still be reasonably viewed as local integration in a biological setting.

### H.1.2  Layer entropy regularization-based SGD updates

Appendix C.3 derives the gradient of the layer-entropy objective:

$$\nabla_{\mathbf{W}^{(k)}} J_E^{(k)}[m] = \frac{2(1-\lambda_E)}{B}\Big[\big((\mathbf{R}_{\mathbf{h}}^{(k)}[m] + \varepsilon^{(k)}\mathbf{I})^{-1}\mathbf{H}^{(k)}[m]\big) \odot \mathbf{F}_d^{(k)}[m]\Big]\mathbf{H}^{(k-1)}[m]^T. \tag{59}$$

Examining an individual element of this gradient shows

$$[\nabla_{\mathbf{W}^{(k)}} J_E^{(k)}[m]]_{ij} = \frac{2(1-\lambda_E)}{B} \sum_{n=mB+1}^{mB+B} v_i^{(k)}[n]\, f_i^{(k)'}(u_i^{(k)}[n])\, h_j^{(k-1)}[n],$$

where $\mathbf{v}^{(k)}[n] = (\mathbf{R}_{\mathbf{h}}^{(k)}[m] + \varepsilon^{(k)}\mathbf{I})^{-1}\mathbf{h}^{(k)}[n]$. Because $v_i^{(k)}[n]$ depends on all neurons' activations in the layer, the layer-entropy update for feedforward weights is not strictly local and hence violates the criteria for strict biological plausibility.

Nevertheless, this limitation is circumvented by the CorInfoMax-EBD approach, wherein the lateral (recurrent) weights, rather than the feedforward weights, implement the layer-entropy maximization objective. We discuss this in the next section.

## H.2  CorInfoMax-EBD implementation

Section 3.2 introduces a more biologically realistic network by combining the CorInfoMax framework—known to yield recurrent networks closely reflecting biological dynamics—with the proposed EBD approach to enable a three-factor update rule in supervised learning.

### H.2.1 Power-normalization-based SGD updates

In CorInfoMax-EBD, we adopt the same power-normalization gradient in Eq. (58) for updating feedforward weights. Therefore, by setting the batch size to $B = 1$, these updates remain local and thus biologically plausible.

### H.2.2 Layer entropy maximization

As summarized in Appendix F, CorInfoMax networks inherently include layer-entropy maximization via the correlative-information objective. Crucially, this entropy maximization is implemented through *lateral weights* of the RNN structure rather than by modifying feedforward weights. Specifically, from the gradient of the correlative-information objective (see Eq. (53)):

$$\nabla_{\mathbf{h}^{(k)}} \hat{J}_k(\mathbf{h}^{(k)})[m] = 2\gamma \, \boldsymbol{B}_{\mathbf{h}^{(k)}}[m] \, \mathbf{h}^{(k)}[m] \; - \; \frac{1}{\epsilon_{k-1}} \overrightarrow{\mathbf{e}}^{(k)}[m] \; - \; \frac{1}{\varepsilon_k} \overleftarrow{\mathbf{e}}^{(k)}[m], \qquad (60)$$

the first term, $2\gamma \, \boldsymbol{B}_{\mathbf{h}^{(k)}}[m] \, \mathbf{h}^{(k)}[m]$, corresponds to the layer-entropy maximization. Here, the lateral weight matrix $\boldsymbol{B}_{\mathbf{h}^{(k)}}[m]$ approximates the inverse of the layer correlation matrix. As described in Appendix F.2.2 (and in [45]), the lateral weights can be updated by an anti-Hebbian rule:

$$\boldsymbol{B}^{(k)}[m+1] = \lambda_{\mathbf{r}}^{-1} \Big( \boldsymbol{B}^{(k)}[m] \; - \; \gamma \, \mathbf{z}^{(k)}[m] \mathbf{z}^{(k)}[m]^T \Big),$$

where

$$\mathbf{z}^{(k)}[m] = \boldsymbol{B}^{(k)}[m] \, \mathbf{h}^{(k)}[m] \big|_{\beta = \beta'}.$$

Once again, this update is strictly local if $B = 1$, while for $B > 1$, the rank-$B$ extension may break strict locality. We demonstrate in Section 4 that CorInfoMax-EBD with $B = 1$ yields comparable or superior performance to CorInfoMax-EP with larger batch sizes.

### H.3 Summary and conclusions

In summary, the CorInfoMax-EBD implementation described in Section 3.2 offers a more biologically plausible approach to supervised learning compared to the MLP-based EBD approach in Section 2 due to several factors:

- Using **lateral weights** to impose layer-entropy maximization in a biologically realistic manner;
- Employing **feedforward/feedback weights** for forward and backward predictive coding;
- Adopting **neuron models** with distinct compartments (soma, basal dendrites, and apical dendrites);
- Incorporating **EBD updates**, which naturally embody a three-factor learning rule; and
- Leveraging **power-normalization updates**, which satisfy local-learning constraints when $B = 1$.

These features stem from the CorInfoMax-EP framework [45], enhanced by our proposed EBD-based regularizations. This architecture reconciles the benefits of layer-entropy and power-normalization objectives with the demands of biological plausibility.

# I  Supplementary on numerical experiments

The models were trained on an NVIDIA Tesla V100 GPU, using the hyperparameters detailed in the sections below. Each experiment was conducted five times under identical settings, and the reported results reflect the average performance. We used the standard train/test splits for the datasets, with MNIST comprising 60,000 training examples and CIFAR-10 comprising 50,000, while both datasets included 10,000 test examples. The MNIST dataset [56] is made available under the Creative Commons Attribution-Share Alike 3.0 license. The CIFAR-10 dataset [45], originating from the University of Toronto, is publicly available for academic research purposes. Both datasets were accessed via standard deep learning library functionalities.

Rather than utilizing automatic differentiation tools, we manually implemented the gradient calculations for the EBD algorithm, utilizing batched operations to ensure computational efficiency. As a side note, the $(1 - \lambda)$ factors present in the derived update expressions are absorbed into the learning rate constants and thus eliminated. In our experiments, we trained the MLP models for **120** epochs and the CNN and LC models for **100** epochs on MNIST and **200** epochs on CIFAR-10. In addition, we trained the CNN model for the CIFAR-100 dataset for **300** epochs, and the CorInfoMax-EBD (3-layer, batch size = 20) model for **60** epochs.

## I.1  Architectures

The architectural details of MLP, CNN and LC networks for the MNIST and CIFAR-10 datasets are shown in Tables 4 and 5, respectively, while the CNN model used in the CIFAR-100 experiments is detailed in Table 7. The structure of the MNIST and CIFAR-10 models are the same as in the reference [9], while the CIFAR-100 model closely matches [6], differing only in the MaxPool shape. In all architectures, we used ReLU as the nonlinear functions except the last layer. Furthermore, the architectural details of the biologically more realistic CorInfoMax network for MNIST and CIFAR-10 datasets are shown in Table 6. These techniques are the same as examples in Appendix J.5 of [45].

Table 4: MNIST architectures. **FC:** fully connected; **Conv:** convolutional; **LC:** locally connected. FC layers are reported by hidden size. Conv/LC layers are reported as (channels, kernel size, stride, padding). Pooling layers use stride 1; we report the kernel size.

| MLP | | Convolutional | | Locally connected | |
|---|---|---|---|---|---|
| FC1 | 1024 | Conv1 | 64, $3 \times 3$, 1, 1 | LC1 | 32, $3 \times 3$, 1, 1 |
| FC2 | 512 | AvgPool | $2 \times 2$ | AvgPool | $2 \times 2$ |
| | | Conv2 | 32, $3 \times 3$, 1, 1 | LC2 | 32, $3 \times 3$, 1, 1 |
| | | AvgPool | $2 \times 2$ | AvgPool | $2 \times 2$ |
| | | FC1 | 1024 | FC1 | 1024 |

Table 5: CIFAR-10 architectures. Conventions are the same as in Table 4.

| MLP | | Convolutional | | Locally connected | |
|---|---|---|---|---|---|
| FC1 | 1024 | Conv1 | 128, $5 \times 5$, 1, 2 | LC1 | 64, $5 \times 5$, 1, 2 |
| FC2 | 512 | AvgPool | $2 \times 2$ | AvgPool | $2 \times 2$ |
| FC3 | 512 | Conv2 | 64, $5 \times 5$, 1, 2 | LC2 | 32, $5 \times 5$, 1, 2 |
| | | AvgPool | $2 \times 2$ | AvgPool | $2 \times 2$ |
| | | Conv3 | 64, $2 \times 2$, 2, 0 | LC3 | 32, $2 \times 2$, 2, 0 |
| | | FC1 | 1024 | FC1 | 512 |

Table 6: CorInfoMax architectures. Conventions are the same as in Table 4.

| MNIST | | CIFAR-10 | |
|---|---|---|---|
| FC1 | 500 | FC1 | 1000 |
| FC2 | 500 | FC2 | 500 |

Table 7: CIFAR-100 architectures. Conventions are the same as in Table 4.

| Convolutional | |
|---|---|
| Conv1 | 96, $5 \times 5$, 1, 2 |
| MaxPool | $2 \times 2$ |
| Conv2 | 128, $5 \times 5$, 1, 2 |
| MaxPool | $2 \times 2$ |
| Conv3 | 256, $5 \times 5$, 1, 2 |
| MaxPool | $2 \times 2$ |
| Dropout | $p$ |
| FC1 | 2048 |
| Dropout | $p$ |
| FC2 | 2048 |
| Softmax | 100 |

## I.2   CorInfoMax-EBD

In this section, we offer additional details regarding the numerical experiments conducted with the CorInfoMax Error Broadcast and Decorrelation (CorInfoMax-EBD) algorithm. Appendix I.2.1 elaborates on the general implementation details. Appendix I.2.2 presents the fundamental learning steps of the algorithm, which are based on the EBD method. Appendices I.2.3 and I.2.4 discuss the initialization of the algorithm's variables and describe the hyperparameters. Finally, Appendix I.2.5 ( 3-Layer and batch size=20, 3-Layer batch size=1) and Appendix I.2.6 (10-Layer batch size=1) detail the specific hyperparameter configurations used in our numerical experiments for the MNIST and CIFAR-10 datasets. In Appendix I.10 we present the accuracy and loss learning curves for the CorInfoMax-EBD, shown in Figures 6.(g)-(h) and Figures 7.(g)-(h), respectively.

### I.2.1   Implementation details

We implemented the CorInfoMax-EBD algorithm based on the repository available at GitHub [2]. This repository from Bozkurt et al. [45], used as a basis for our CorInfoMax-EBD implementation, did not specify an explicit license in its public repository at the time of access. Our use and modification are for academic research purposes, building upon the published scientific work presented in [45]. The following modifications were made to the original code:

- **Reduction to a single phase:** We simplified the algorithm by reducing it to a single phase. Specifically, we removed the nudge phase, during which the label is coupled to the network dynamics. In this modified version, the network operates solely in the free phase, where the label is decoupled from the network. This change aligns with the removal of time-contrastive updates from the CorInfoMax-EP algorithm.

- **Algorithmic updates:** We incorporated the updates outlined in Algorithm 3.

- **Hyperparameters:** We maintained the same hyperparameters for the neural dynamics as in the original code. Additionally, new hyperparameters specific to the learning dynamics were introduced, which are detailed in Appendix I.2.4.

In the CorInfoMax-EBD implementation the following loss and regularization functions are used

- EBD loss: $J^{(k)}$,

- Power normalization loss: $J_P^{(k)}$,

- $\ell_2$ weight regularization (weight decay): $J_{\ell_2}^{(k)}$,

- Activation sparsity regularization: $J_{\ell_1}^{(k)} = \|\mathbf{H}^{(k)}\|_1$.

---

[2] https://github.com/BariscanBozkurt/Supervised-CorInfoMax

### I.2.2 Algorithm

The CorInfoMax-EBD algorithm follows the same neural dynamics framework detailed in [45] for computing neuron activations. Consequently, we only outline the steps specific to the learning process, which distinguishes it from the original CorInfoMax-EP algorithm described in [45]. The full iterative process for updating weights in the CorInfoMax-EBD algorithm is provided in Algorithm 3.

---

**Algorithm 3** CorInfoMax Error Broadcast and Decorrelation (CorInfoMax-EBD) Update for Layer $k$

---

**Require:** Learning rate parameters $\lambda_d, \lambda_E$, $\mu^{(d,k)}[m]$, $\mu^{(f,k)}[m]$, $\mu^{(b,k)}[m]$

**Require:** Previous synaptic weights $\mathbf{W}^{(f,k)}[m-1]$ (forward), $\mathbf{W}^{(b,k)}[m-1]$ (backward), $\mathbf{B}^{(k)}[m-1]$ (lateral)

**Require:** Previous error projection weights $\mathbf{R}_{g(\mathbf{h}^{(k)})\boldsymbol{\epsilon}}[m-1]$

**Require:** Batch size $B$

**Require:** Layer activations $\mathbf{H}^{(k)}[m]$ in Eq. (13), the derivative of activations $\mathbf{F}_d^{(k)}$ in Eq. (29), in Eq. (5), prediction errors $\overleftarrow{\mathbf{E}}$ and $\overrightarrow{\mathbf{E}}^{(k)}$ in Eq. (54)-Eq. (55), lateral weight outputs $\mathbf{Z}^{(k)}$ in Eq. (56) computed by CorInfoMax network dynamics described in Bozkurt et al. [45] (and Appendix F)

**Require:** The nonlinear function of layer activations $\mathbf{G}^{(k)}$ in Eq. (4) and the derivative of the nonlinear function of layer activations $\mathbf{G}_d^{(k)}$ in Eq. (28)

**Ensure:** Updated weights $\mathbf{W}^{(f,k)}[m]$, $\mathbf{W}^{(b,k)}[m]$, $\mathbf{B}^{(k)}[m]$

---

**Error projection weight update for layer $k$:**

1: $\hat{\mathbf{R}}_{\mathbf{g}\boldsymbol{\epsilon}}^{(k)}[m] \leftarrow \lambda_d\,\hat{\mathbf{R}}_{\mathbf{g}\boldsymbol{\epsilon}}^{(k)}[m-1] + \dfrac{1-\lambda_d}{B}\,\mathbf{G}^{(k)}[m]\,\mathbf{E}^{(k)}[m]^T$

**Project errors to layer $k$:**

2: $\mathbf{Q}^{(k)}[m] \leftarrow \hat{\mathbf{R}}_{\mathbf{g}\boldsymbol{\epsilon}}^{(k)}[m]\,\mathbf{E}^{(k)}[m]$

**Find the gradient of the nonlinear function of activations for layer $k$:**

3: $\mathbf{\Phi}^{(k)}[m] = \mathbf{F}_d^{(k)}[m] \odot \mathbf{Q}^{(k)}[m] \odot \mathbf{G}_d^{(k)}[m]$

**Update forward weights for layer $k$:**

4: $\Delta\mathbf{W}_{\mathrm{EBD}}^{(f,k)}[m] \leftarrow -\dfrac{\mu^{(d_f,k)}[m]}{B}\,\mathbf{\Phi}^{(k)}[m]\mathbf{H}^{(k-1)}[m]^{\top}$

5: $\Delta\mathbf{W}_{\mathrm{Pred}}^{(f,k)}[m] \leftarrow \dfrac{\mu^{(f,k)}[m]}{B}\,\overrightarrow{\mathbf{E}}^{(k)}[m]\left(\mathbf{H}^{(k-1)}[m]\right)^{\top}$

6: $\mathbf{W}^{(f,k)}[m] \leftarrow \mathbf{W}^{(f,k)}[m-1] + \Delta\mathbf{W}_{\mathrm{EBD}}^{(f,k)}[m] + \Delta\mathbf{W}_{\mathrm{Pred}}^{(f,k)}[m]$

**Update backward weights for layer $k$:**

7: $\Delta\mathbf{W}_{\mathrm{EBD}}^{(b,k)}[m] \leftarrow -\dfrac{\mu^{(d_b,k)}[m]}{B}\,\mathbf{\Phi}^{(k)}[m]\,\mathbf{H}^{(k+1)}[m]^{\top}$

8: $\Delta\mathbf{W}_{\mathrm{Pred}}^{(b,k)}[m] \leftarrow \dfrac{\mu^{(b,k)}[m]}{B}\,\overleftarrow{\mathbf{E}}^{(k)}[m]\,\mathbf{H}^{(k+1)}[m]^{\top}$

9: $\mathbf{W}^{(b,k)}[m] \leftarrow \mathbf{W}^{(b,k)}[m-1] + \Delta\mathbf{W}_{\mathrm{EBD}}^{(b,k)}[m] + \Delta\mathbf{W}_{\mathrm{Pred}}^{(b,k)}[m]$

**Update Lateral weights for layer $k$:**

10: $\Delta\mathbf{B}_{\mathrm{EBD}}^{(k)}[m] \leftarrow -\dfrac{\mu^{(d_l,k)}[m]}{B}\,\mathbf{\Phi}^{(k)}[m]\,\mathbf{H}^{(k)}[m]^{\top}$

11: $\Delta\mathbf{B}_{\mathrm{E}}^{(k)}[m] \leftarrow -\dfrac{\gamma_E}{B}\,\mathbf{Z}^{(k)}[m]\mathbf{Z}^{(k)}[m]^T$

12: $\mathbf{B}^{(k)}[m] \leftarrow \dfrac{1}{\lambda_E}\mathbf{B}^{(k)}[m] + \Delta\mathbf{B}_{\mathrm{E}}^{(k)}[m] + \Delta\mathbf{B}_{\mathrm{EBD}}^{(k)}[m]$

---

### I.2.3 Initialization of algorithm variables

We initialize the variables $\mathbf{W}^{(f,k)}$, $\mathbf{W}^{(b,k)}$, and $\mathbf{R}_{\mathbf{h}^{(k)}\epsilon}$ using PyTorch's Xavier uniform initialization with its default parameters for the MNIST dataset. For the CIFAR-10 dataset is initialized with gain 0.25. For the lateral weights $\mathbf{B}^{(k)}$, we first generate a random matrix $\mathbf{J}^{(k)}$ of the same dimensions, also using the Xavier uniform distribution, with gain= 1 for the MNIST dataset and with gain= 0.5 for the CIFAR-10 dataset. We then compute $\mathbf{B}^{(k)}[0] = \mathbf{J}^{(k)}\mathbf{J}^{(k)^T}$, ensuring that $\mathbf{B}^{(k)}[0]$ is a positive definite symmetric matrix.

### I.2.4 Description of hyperparameters

Table 8 presents a description of the hyperparameters used in the CorInfoMax-EBD implementation.

Table 8: Detailed explanation of hyperparameter notations for the CorInfoMax-EBD algorithm

| Hyperparameter | Description |
|---|---|
| $\alpha[m]$ | Learning rate dynamic scaling factor |
| $\alpha_2[m]$ | Learning rate dynamic scaling factor 2 |
| $\mu^{(d_f,k)}$ | Learning rate for decorrelation loss (forward weights) |
| $\mu^{(d_b,k)}$ | Learning rate for decorrelation loss (backward weights) |
| $\mu^{(d_l,k)}$ | Learning rate for decorrelation loss (lateral weights) |
| $\mu^{(f,k)}$ | Learning rate for forward prediction |
| $\mu^{(b,k)}$ | Learning rate for backward prediction |
| $\mu^{(p,k)}$ | Learning rate for power normalization loss |
| $p^{(k)}$ | Target power level |
| $\mu_{f,\ell_1}^{(k)}$ | Learning rate for activation sparsity (forward weights) |
| $\mu_{b,\ell_1}^{(k)}$ | Learning rate for activation sparsity (backward weights) |
| $\mu_{f,w-\ell_2}^{(k)}$ | Forward weight $\ell_2$-regularization coefficent |
| $\mu_{b,w-\ell_2}^{(k)}$ | Backward weight $\ell_2$-regularization coefficent |
| $\lambda_E$ | Layer correlation matrix update forgetting factor |
| $\lambda_d$ | Error-layer activation cross-correlation forgetting factor |
| $m^{(d)}$ | Momentum factor for decorrelation forward weight gradient |
| $B$ | Batch size |

### I.2.5 Hyperparameters for 3-Layer MNIST and CIFAR-10 Models

Table 9 and 10 summarizes the hyperparameters used in the 3-layer CorInfoMax-EBD experiments for the MNIST and CIFAR-10 datasets with a batch size of 20 and 1 respectively. The iteration index is denoted by $m$ in all expressions.

Table 9: 3-Layer CorInfoMax-EBD hyperparameters for MNIST and CIFAR-10 datasets ($B = 20$).

| Hyperparameter | MNIST | CIFAR-10 |
|---|---|---|
| $\alpha[m]$ | $\dfrac{1}{3 \times 10^{-3} \times \lfloor \frac{m}{10} \rfloor + 1}$ | $\dfrac{1}{3 \times 10^{-3} \times \lfloor \frac{m}{10} \rfloor + 1}$ |
| $\alpha_2[m]$ | $\dfrac{1}{3 \times \lfloor \frac{m}{10} \rfloor + 1}$ | $\dfrac{1}{3 \times \lfloor \frac{m}{10} \rfloor + 1}$ |
| $\mu^{(d_f,k)}[m]$ | $[96, 60, 1e5]\alpha[m]$ | $[80, 50, 1e5]\alpha[m]$ for epoch= 0 
 $[320, 400, 1e5]\alpha[m]$ for epoch> 0 |
| $\mu^{(d_b,k)}[m]$ | $[96, 60, 1e5]\alpha[m]$ | $[0, 0, 0]\alpha[m]$ |
| $\mu^{(d_l,k)}[m]$ | $[0.25, 0.25, 0.25]\alpha[m]$ for epoch= 0 
 $[0.5, 0.5, 0.5]\alpha[m]$ for epoch> 0 | $[0.5, 0.5, 0.5]\alpha[m]$ for epoch= 0 
 $[2.0, 2.0, 2.0]\alpha[m]$ for epoch> 0 |
| $\mu^{(f,k)}[m]$ | $[0.11 \times 10^{-18}, 0.06 \times 10^{-18}, 0.035 \times 10^{-18}]\alpha[m]$ | $[0.11 \times 10^{-18}, 0.06 \times 10^{-18}, 0.035 \times 10^{-18}]\alpha[m]$ |
| $\mu^{(b,k)}[m]$ | $[1.125 \times 10^{-18}, 0.375 \times 10^{-18}]\alpha[m]$ | $[1.125 \times 10^{-18}, 0.375 \times 10^{-18}]\alpha[m]$ |
| $\mu^{(p,k)}[m]$ | $[4.4 \times 10^{-3}, 6 \times 10^{-3}, 3.5 \times 10^{-12}]\alpha_2[m]$ | $[4.4 \times 10^{-3}, 6 \times 10^{-3}, 3.5 \times 10^{-12}]\alpha_2[m]$ |
| $p^{(k)}$ | $[2.5, 2.5, 0.1]$ | $[2.5, 2.5, 0.1]$ |
| $\mu_{f,\ell_1}^{(k)}[m]$ | $[0.008, 0.135, 0]\alpha_2[m]$ | $[0.008, 0.135, 0]\alpha_2[m]$ |
| $\mu_{b,\ell_1}^{(k)}[m]$ | $[0, 0.35, 0.05]\alpha_2[m]$ | $[0, 0.35, 0.05]\alpha_2[m]$ |
| $\mu_{f,w-\ell_2}^{(k)}[m]$ | $\dfrac{8 \times 10^{-2}}{10^{-2} \times \lfloor \frac{m}{10} \rfloor + 1}$ | $\dfrac{8 \times 10^{-2}}{10^{-2} \times \lfloor \frac{m}{10} \rfloor + 1}$ |
| $\mu_{b,w-\ell_2}^{(k)}[m]$ | $\dfrac{8 \times 10^{-2}}{10^{-2} \times \lfloor \frac{m}{10} \rfloor + 1}$ | $\dfrac{8 \times 10^{-2}}{10^{-2} \times \lfloor \frac{m}{10} \rfloor + 1}$ |
| $\lambda_E$ | 0.999999 | 0.999999 |
| $\lambda_d$ | 0.99999 | 0.99999 |
| $m^{(d)}[m]$ | $0.99\dfrac{1}{\lfloor \frac{m}{10} \rfloor + 1} + 0.999\left(1 - \dfrac{1}{\lfloor \frac{m}{10} \rfloor + 1}\right)$ | $0.99\dfrac{1}{\lfloor \frac{m}{10} \rfloor + 1} + 0.999\left(1 - \dfrac{1}{\lfloor \frac{m}{10} \rfloor + 1}\right)$ |
| $B$ | 20 | 20 |

Table 10: 3-Layer CorInfoMax-EBD hyperparameters for MNIST and CIFAR-10 datasets ($B = 1$).

| Hyperparameter | MNIST | CIFAR-10 |
|---|---|---|
| $\alpha[m]$ | $\dfrac{1}{3 \times 10^{-3} \times \lfloor \frac{m}{10} \rfloor + 1}$ | $\dfrac{1}{3 \times 10^{-3} \times \lfloor \frac{m}{10} \rfloor + 1}$ |
| $\alpha_2[m]$ | $\dfrac{1}{3 \times \lfloor \frac{m}{10} \rfloor + 1}$ | $\dfrac{1}{3 \times \lfloor \frac{m}{10} \rfloor + 1}$ |
| $\mu^{(d_f,k)}[m]$ | $[4.8, 3.0, 5 \times 10^3]\alpha[m]$ | $[4, 2.5, 5 \times 10^3]\alpha[m]$ for epoch$= 0$ $[16, 20, 5 \times 10^3]\alpha[m]$ for epoch$> 0$ |
| $\mu^{(d_b,k)}[m]$ | $[4.8, 3.0, 5 \times 10^3]\alpha[m]$ | $[0, 0, 0]\alpha[m]$ |
| $\mu^{(d_l,k)}[m]$ | $[0.0125, 0.0125, 0.0125]\alpha[m]$ for epoch$= 0$ $[0.025, 0.025, 0.025]\alpha[m]$ for epoch$> 0$ | $[0.025, 0.025, 0.025]\alpha[m]$ for epoch$= 0$ $[0.1, 0.1, 0.1]\alpha[m]$ for epoch$> 0$ |
| $\mu^{(f,k)}[m]$ | $[0.11 \times 10^{-18}, 0.06 \times 10^{-18}, 0.035 \times 10^{-18}]\alpha[m]$ | $[0.11 \times 10^{-18}, 0.06 \times 10^{-18}, 0.035 \times 10^{-18}]\alpha[m]$ |
| $\mu^{(b,k)}[m]$ | $[1.125 \times 10^{-18}, 0.375 \times 10^{-18}]\alpha[m]$ | $[1.125 \times 10^{-18}, 0.375 \times 10^{-18}]\alpha[m]$ |
| $\mu^{(p,k)}[m]$ | $[2.2 \times 10^{-4}, 3 \times 10^{-4}, 3.5 \times 10^{-12}]\alpha_2[m]$ | $[2.2 \times 10^{-4}, 3 \times 10^{-4}, 3.5 \times 10^{-12}]\alpha_2[m]$ |
| $p^{(k)}$ | $[2.5, 2.5, 0.1]$ | $[0.125, 0.125, 0.005]$ |
| $\mu^{(k)}_{f,\ell_1}[m]$ | $[0.0004, 0.00675, 0]\alpha_2[m]$ | $[0.0004, 0.000675, 0]\alpha_2[m]$ |
| $\mu^{(k)}_{b,\ell_1}[m]$ | $[0, 0.0175, 0.0025]\alpha_2[m]$ | $[0, 0.0175, 0.0025]\alpha_2[m]$ |
| $\mu^{(k)}_{f,w-\ell_2}[m]$ | $\dfrac{8 \times 10^{-2}}{10^{-2} \times \lfloor \frac{m}{10} \rfloor + 1}$ | $\dfrac{8 \times 10^{-2}}{10^{-2} \times \lfloor \frac{m}{10} \rfloor + 1}$ |
| $\mu^{(k)}_{b,w-\ell_2}[m]$ | $\dfrac{8 \times 10^{-2}}{10^{-2} \times \lfloor \frac{m}{10} \rfloor + 1}$ | $\dfrac{8 \times 10^{-2}}{10^{-2} \times \lfloor \frac{m}{10} \rfloor + 1}$ |
| $\lambda_E$ | $0.99999995$ | $0.99999995$ |
| $\lambda_d$ | $0.99999$ | $0.99999$ |
| $m^{(d)}[m]$ | $0.99\dfrac{1}{\lfloor \frac{m}{10} \rfloor + 1} + 0.999\left(1 - \dfrac{1}{\lfloor \frac{m}{10} \rfloor + 1}\right)$ | $0.99\dfrac{1}{\lfloor \frac{m}{10} \rfloor + 1} + 0.999\left(1 - \dfrac{1}{\lfloor \frac{m}{10} \rfloor + 1}\right)$ |
| $B$ | $1$ | $1$ |

### I.2.6 Hyperparameters for 10-layer CorInfoMax-EBD on MNIST and CIFAR-10 datasets for batch size 1

Table 11 list the hyperparameters used in the 10-Layer CorInfoMax-EBD numerical experiments for the MNIST and CIFAR-10 datasets with a batch size of 1. In these experiments, a weight thresholding scheme is applied to the network weights for every 5000 samples, where the weights with 0.00003 scale (relative to the peak) are set to zero.

Table 11: 10-Layer CorInfoMax-EBD hyperparameters: MNIST and CIFAR-10 datasets ($B = 1$).

| Hyperparameter | Value |
|---|---|
| $\alpha[m]$ | $\frac{1}{3\times10^{-3}\times\lfloor \frac{m}{10} \rfloor+1}$ |
| $\alpha_2[m]$ | $\frac{1}{3\times\lfloor \frac{m}{10} \rfloor+1}$ |
| $\mu^{(d_f,k)}[m]$ | $[\ 3.5 \quad 3.5 \quad \ldots \quad 3.5 \quad 6e4\ ]\,\alpha[m]$ |
| $\mu^{(d_l,k)}[m]$ | $0.03\alpha[m] \cdot \mathbf{1}_{1\times10}$ |
| $\mu^{(f,k)}[m]$ | $[\ 0.1, 0.1, 0.1, 0.1, 0.1, 0.1, 0.1, 0.11, 0.06, 0.035\ ] \cdot 1e(-18) \cdot \alpha[m]$ |
| $\mu^{(b,k)}[m]$ | $[\ 1.1, 0.4, 0.4, 0.4, 0.4, 0.4, 0.4, 0.4, 0.4, 0.4\ ] \cdot 1e(-18) \cdot \alpha[m]$ |
| $\mu^{(p,k)}[m]$ | $[\ 2, 2, 2, 2, 2, 2, 2, 2, 5, 1e-7\ ] \cdot 1e(-3) \cdot \alpha_2[m]$ |
| $p^{(k)}$ | $[\ 1, 1, 1, 1, 1, 1, 1, 1, 2, 0.1\ ]$ |
| $\mu^{(k)}_{f,\ell_1}[m]$ | $[\ 0.16, 0.16, 0.16, 0.16, 0.16, 0.16, 0.16, 0.16, 0.16, 0.0\ ]\,\alpha_2[m]$ |
| $\mu^{(k)}_{f,w-\ell_2}[m]$ | $\mathbf{0}_{1\times10}$ |
| $\mu^{(k)}_{l,w-\ell_2}[m]$ | $5e-4 \cdot \alpha_2[m]\mathbf{1}_{1\times10}$ |
| $\lambda_E$ | $0.99999995$ |
| $\lambda_d$ | $0.999999\gamma + (1-\gamma)0.99999999$ with $\gamma = \frac{1}{\frac{m}{5}+1}$ |
| $m^{(d)}[m]$ | $0.99\frac{1}{\lfloor \frac{m}{10} \rfloor+1} + 0.999(1 - \frac{1}{\lfloor \frac{m}{10} \rfloor+1})$ |
| $B$ | $1$ |

### I.3 Multi-Layer Perceptron

In this section, we provide additional details about the numerical experiments conducted to train Multi-layer Perceptrons (MLPs) using the EBD algorithm (MLP-EBD). Appendix I.3.1 outlines the implementation details of these experiments, while Appendix I.3.2 discusses the initialization of algorithm variables. Information about hyperparameters and their values for the MNIST and CIFAR-10 datasets can be found in Appendices I.3.3-I.3.4. In Appendix I.10 we present the accuracy and loss learning curves for the MLP architecture, shown in Figures 6.(a)-(b) and Figures 7.(a)-(b), respectively.

#### I.3.1 Implementation details

For the MLP experiments using the proposed EBD approach, we adopted the same network architecture as described in [9], detailed in Tables 4 and 5.

In the MLP-EBD implementation, the following loss and regularization functions were employed:

- EBD loss: $J^{(k)}$,

- Power normalization loss: $J_P^{(k)}$,

- Entropy objective: $J_E^{(k)}$,

- $\ell_2$ weight regularization (weight decay): $J_{\ell_2}^{(k)}$,

- Activation sparsity regularization: $J_{\ell_1}^{(k)} = \|\mathbf{H}^{(k)}\|_1$.

Additionally, we imposed a weight-sparsity constraint by setting $WS$ percent of the weights to zero during the initialization phase and maintaining these weights at zero throughout training.

#### I.3.2 Initialization of algorithm variables

We use the Pytorch framework's Xavier uniform initialization with gain value $10^{-2}$ on the $\mathbf{R}_{\mathbf{h}^{(k)}\boldsymbol{\epsilon}}$ variables, and Kaiming uniform distribution with gain $0.75$ for synaptic weights $\mathbf{W}^{(k)}$.

#### I.3.3 Description of hyperparameters

Table 12 provides the description of the hyperparameters for the MLP-EBD implementation.

Table 12: Description of the hyperparameter notations for MLP-EBD.

| Hyperparameter | Description |
|---|---|
| $\alpha[m]$ | Learning rate dynamic scaling factor |
| $\alpha_2[m]$ | Learning rate dynamic scaling factor 2 |
| $\mu^{(d,b,k)}$ | Learning rate for (backward projection) decorrelation loss |
| $\mu^{(d,f,k)}$ | Learning rate for (forward projection) decorrelation loss |
| $\mu^{(E,k)}$ | Learning rate for entropy objective |
| $\mu^{(p,k)}$ | Learning rate for power normalization loss |
| $p^{(k)}$ | Target power level |
| $\mu_{\ell_1}^{(k)}$ | Learning rate for activation sparsity |
| $\mu_{w-\ell_2}^{(k)}$ | Weight $\ell_2$-regularization coefficent |
| $\lambda_E$ | Layer autocorrelation matrix update forgetting factor |
| $\lambda_d$ | Error-layer activation cross-correlation forgetting factor |
| $m^{(d)}$ | Momentum factor for decorrelation gradient |
| $B$ | Batch size |
| $WS$ | Weight Sparsity |

### I.3.4 Hyperparameters for MLP-EBD on MNIST and CIFAR-10 Datasets

Table 13 summarizes the hyperparameters used in the MLP-EBD experiments for the MNIST and CIFAR-10 datasets. The iteration index is denoted by $m$ in all expressions.

Table 13: MLP-EBD hyperparameters for MNIST and CIFAR-10 datasets.

| Hyperparameter | MNIST | CIFAR-10 |
|---|---|---|
| $\alpha[m]$ | $\dfrac{1}{1.5 \times \lfloor \frac{m}{10} \rfloor + 1}$ | $\dfrac{1}{1.5 \times \lfloor \frac{m}{10} \rfloor + 1}$ |
| $\alpha_2[m]$ | $\dfrac{\lfloor \frac{m}{10} \rfloor}{3 \times 10^4} + 1$ | $\dfrac{\lfloor \frac{m}{10} \rfloor}{3 \times 10^4} + 1$ |
| $\mu^{(d,b,k)}[m]$ | $18000\,\alpha[m]\alpha_2[m]$ for $k = 0, 1$ 
 $20000\,\alpha[m]\alpha_2[m]$ for $k = 2$ | $[4000,\, 2000,\, 2000,\, 3500]\,\alpha[m]\alpha_2[m]$ |
| $\mu^{(d,f,k)}[m]$ | $0.005\,\alpha[m]\alpha_2[m]$ for $k = 0, 1$ | $0.005\,\alpha[m]\alpha_2[m]$ for $k = 0, 1$ |
| $\mu^{(E,k)}[m]$ | $[2.5 \times 10^{-4},\, 1.5 \times 10^{-3},\, 0]\,\alpha[m]$ | $[2.5 \times 10^{-4},\, 1.5 \times 10^{-3},\, 1.5 \times 10^{-3},\, 0]\,\alpha[m]$ |
| $\mu^{(p,k)}[m]$ | $[4 \times 10^{-3},\, 6 \times 10^{-3},\, 1 \times 10^{-10}]\,\alpha[m]$ | $[4 \times 10^{-3},\, 6 \times 10^{-3},\, 6 \times 10^{-3},\, 1 \times 10^{-10}]\,\alpha[m]$ |
| $p^{(k)}[m]$ | $[0.25,\, 0.25,\, 0.1]\,\alpha[m]$ | $[0.25,\, 0.25,\, 0.25,\, 0.1]\,\alpha[m]$ |
| $\mu^{(k)}_{\ell_1}$ | $[0.8,\, 0.3,\, 0]\,\alpha[m]$ | $[0.8,\, 0.3,\, 0.3,\, 0]\,\alpha[m]$ |
| $\mu^{(k)}_{w-\ell_2}$ | $1.6 \times 10^{-4}\,\alpha[m]$ for all layers | $1.6 \times 10^{-4}\,\alpha[m]$ for all layers |
| $\lambda_E$ | 0.99999 | 0.99999 |
| $\lambda_d$ | 0.999999 | 0.999999 |
| $m^{(d)}$ | 0.9999 for all layers | 0.9999 for all layers |
| $B$ | 20 | 20 |
| $WS$ | 55 | 40 |

## I.4 Convolutional Neural Network

In this section, we offer additional details regarding the numerical experiments for training Convolutional Neural Neural Networks (CNNs) using EBD algorithm (CNN-EBD). Section I.4.1 provides information about implemetation details. Appendices I.4.2 and I.4.3 discuss the initialization of the algorithm's variables and describe the hyperparameters. Finally, Appendix I.4.4 detail the specific hyperparameter configurations used in our numerical experiments for the MNIST, CIFAR-10 and CIFAR-100 datasets. In Appendix I.10 we present the accuracy and loss learning curves for the CNN, shown in Figures 6.(c)-(d) and Figures 7.(c)-(d), respectively.

### I.4.1 Implementation details

The architectures we utilized for the CNN networks can be found in tables 4 and 5 respectively for the MNIST and CIFAR10 datasets. In the training, we used the Adam optimizer with hyperparameters $\beta_1 = 0.9$, $\beta_2 = 0.999$, and $\epsilon = 10^{-8}$ [57]. Also, the model biases are not utilized. In the CNN-EBD implementation the following loss and regularization functions as detailed in section D.1 are used:

- EBD loss: $J^{(k)}$,
- Entropy objective: $J_E^{(k)}$,
- Activation sparsity regularization: $J_{\ell_1}^{(k)}$.

Specifically for the CIFAR-100 experiments, we applied both training and test time augmentation of data to improve model generalization and handle increased difficulty in the task. At training time, each image was randomly translated into 2-pixels with reflection padding and a deterministic alternating horizontal flip that ensures that every image is flipped every other epoch, reducing redundancy compared to standard random flipping. During evaluation, we used test-time augmentation combining horizontal flipping and multi-crop averaging over six translated views (the original, two one-pixel translations, and their mirrored counterparts).

### I.4.2 Initialization of algorithm variables

We use the Kaiming normal initialization for the weights, with a common standard deviation scaling parameter $\sigma_\mathbf{W}$, on both the linear and convolutional layers. Furthermore, the estimated cross-correlation variable $\mathbf{R}_{\mathbf{h}^{(k)}\epsilon}$ (linear layers) and $\mathbf{R}_{\mathbf{g}^{(k)}(\mathbf{H}^{(k,p)})\epsilon}$ (convolutional layers) are initialized with zero mean normal distributions with standard deviations $\sigma_{\mathbf{R}_{lin}}$ and $\sigma_{\mathbf{R}_{conv}}$ respectively.

### I.4.3 Description of hyperparameters

Table 14 describes the notation for the hyperparameters used to train CNNs using the Error Broadcast and Decorrelation (EBD) approach.

Table 14: Description of the hyperparameter notations for CNN-EBD.

| Hyperparameter | Description |
|---|---|
| $\alpha_{\exp}$ | Exponential learning rate decay parameter. |
| $\alpha[i]$ | Learning rate dynamic scaling factor where $i$ is the epoch index |
| $\mu^{(d,b,k)}$ | Learning rate for (backward projection) decorrelation loss |
| $\mu^{(E,k)}$ | Learning rate for entropy objective |
| $\mu_{\ell_1}^{(k)}$ | Learning rate for activation sparsity |
| $\sigma_{\mathbf{W}}$ | Standard deviation of the weight initialization. |
| $\sigma_{\mathbf{R}_{lin}}$ | Std. dev. of $\mathbf{R}_{\mathbf{h}^{(k)}\boldsymbol{\epsilon}}$ initialization in linear layers |
| $\sigma_{\mathbf{R}_{conv}}$ | Std. dev. of $\mathbf{R}_{\mathbf{g}^{(k)}(\mathbf{H}^{(k,p)})\boldsymbol{\epsilon}}$ initialization in convolutional layers |
| $\sigma_{\mathbf{R}_{local}}$ | Gain parameter for $\mathbf{R}_{\mathbf{g}^{(k)}(\mathbf{H}^{(k,p)})\boldsymbol{\epsilon}}$ initialization in locally connected layers |
| $\lambda_E$ | Layer autocorrelation matrix update forgetting factor |
| $\lambda_d$ | Error-layer activation cross-correlation forgetting factor |
| $\lambda_R$ | Convergence parameter for $\lambda$ as in Equations (61), (62) |
| $\epsilon_L$ | Entropy objective epsilon parameter for linear layers |
| $\epsilon$ | Entropy objective epsilon parameter for conv. or locally con. layers |
| $\beta$ | Adam Optimizer weight decay parameter |
| $p$ | Dropout probability |
| $B$ | Batch size |

We also introduce a convergence parameter $\lambda_R$ which increases the estimation parameter for the decorrelation loss $\lambda_d$, together with the estimation parameter for the layer entropy objective $\lambda_E$, to converge to 1 as the training proceeds with the following Equations (61), (612) where $i$ is the epoch index:

$$\lambda_d^{(i+1)} = \lambda_d^{(i)} + \lambda_R \cdot \left(1 - \lambda_d^{(i)}\right), i \geq 0. \tag{61}$$

$$\lambda_E^{(i+1)} = \lambda_E^{(i)} + \lambda_R \cdot \left(1 - \lambda_E^{(i)}\right), i \geq 0. \tag{62}$$

### I.4.4 Hyperparameters for MNIST, CIFAR-10 and CIFAR-100 datasets

Table 15, lists the hyperparameters as defined in Table 14, used in the CNN-EBD training experiments.

Table 15: Hyperparameters for CNN-EBD for the MNIST, CIFAR-10 and CIFAR-100 datasets, where $i$ denotes the epoch index.

| Hyperparameter | MNIST | CIFAR-10 | CIFAR-100 |
|---|---|---|---|
| $\alpha_{\exp}$ | 0.97 | 0.97 | 0.97 |
| $\alpha[i]$ | $10^{-4} \cdot \alpha_{\exp}^{-i}$ | $10^{-4} \cdot \alpha_{\exp}^{-i}$ | $10^{-4} \cdot \alpha_{\exp}^{-i}$ |
| $\mu^{(d,b,k)[i]}$ | $0.1\alpha[i]$ for $k=0,1,2,3$ $10\alpha[i]$ for $k=4$ | $0.1\alpha[i]$ for $k=0,1,2,3$ $10\alpha[i]$ for $k=4$ | $0.1\alpha[i]$ for $k=0,1,2,3,4$ $10\alpha[i]$ for $k=5$ |
| $\mu^{(E,k)}[i]$ | $\begin{bmatrix} 1 & 1 & 1 & 10 & 0 \end{bmatrix}10^{-7}\alpha[i]$ | $\begin{bmatrix} 1 & 1 & 1 & 1 & 1 \end{bmatrix}10^{-6}\alpha[i]$ | $\begin{bmatrix} 0 & 0 & 0 & 5 & 5 & 5 \end{bmatrix}10^{-7}\alpha[i]$ |
| $\mu_{\ell_1}^{(k)}[i]$ | $\begin{bmatrix} 1 & 1 & 1 & 10 & 0 \end{bmatrix}10^{-11}\alpha[i]$ | $\begin{bmatrix} 1 & 1 & 1 & 10^2 & 0 \end{bmatrix}10^{-10}\alpha[i]$ | $\begin{bmatrix} 0 & 0 & 0 & 1 & 1 & 0 \end{bmatrix}10^{-7}\alpha[i]$ |
| $\sigma_{\mathbf{W}}$ | $\sqrt{\frac{1}{6}}$ | $\sqrt{\frac{1}{6}}$ | $\sqrt{\frac{1}{6}}$ |
| $\sigma_{\mathbf{R}_{lin}}$ | $1e-2$ | $1e-2$ | $1e-2$ |
| $\sigma_{\mathbf{R}_{conv}}$ | $1e-2$ | $1e-2$ | $1e-2$ |
| $\lambda_d$ | 0.99999 | 0.99999 | 0.99999 |
| $\lambda_E$ | 0.99999 | 0.99999 | 0.99999 |
| $\lambda_R$ | $2e-2$ | $2e-2$ | $2e-2$ |
| $\beta$ | $1e-8$ | $1e-5$ | $1e-5$ |
| $\epsilon_L$ | $1e-8$ | $1e-8$ | $1e-8$ |
| $\epsilon$ | $1e-5$ | $1e-5$ | $1e-5$ |
| $B$ | 16 | 16 | 16 |
| $p$ | N/A | N/A | 0.075 |

## I.5  Locally Connected Network

In this section, we offer additional details regarding the numerical experiments for the training of Locally Connected Networks (LCs) using EBD algorithm (LC-EBD). Appendix I.5.1 provides information about implemetation details. Appendices I.5.2 and I.5.3 discuss the initialization of the algorithm's variables and describe the hyperparameters. Finally, Appendix I.5.4 detail the specific hyperparameter configurations used in our numerical experiments for the MNIST and CIFAR-10 datasets. In Appendix I.10 we present the accuracy and loss learning curves for the LCs, shown in Figures 6.(e)-(f) and Figures 7.(e)-(f), respectively.

### I.5.1  Implementation details

The training procedure mirrors the CNN approach described in Section I.4.1 for CNNs. In the LC-EBD implementation, the loss and regularization functions detailed in section D.2 are used:

- EBD loss: $J^{(k)}$,

- Entropy objective: $J_E^{(k)}$,

- Activation sparsity regularization: $J_{\ell_1}^{(k)}$.

### I.5.2  Initialization of algorithm variables

We use the Kaiming uniform initialization for the weights, with a common standard deviation scaling parameter $\sigma_{\mathbf{W}}$, on both the linear and locally connected layers. The estimated cross-correlation variable $\mathbf{R}_{\mathbf{h}^{(k)}\boldsymbol{\epsilon}}$ (linear layers) is initialized with a normal distribution with zero mean and standard deviation $\sigma_{\mathbf{R}_{lin}}$. Also, the parameter $\mathbf{R}_{\mathbf{g}^{(k)}(\mathbf{H}^{(k,p)})\boldsymbol{\epsilon}}$ (locally connected layers) is initialized with Pytorch framework's Xavier uniform initialization with the gain parameter equal to $\sigma_{\mathbf{R}_{local}}$.

### I.5.3  Description of hyperparameters

Table 14 in the CNN section, again describes the notation for the hyperparameters used to train LCs using the Error Broadcast and Decorrelation (EBD) approach. The convergence parameter $\lambda_R$ introduced in equations Eq. (61) and Eq. (61) is used as well.

### I.5.4  Hyperparameters for MNIST and CIFAR-10 datasets

Table 16, lists the hyperparameters as defined in Table 14, used in the LC-EBD training experiments.

Table 16: Hyperparameters for LC-EBD for both the MNIST and CIFAR-10 datasets, where $i$ denotes the epoch index.

| Hyperparameter | MNIST | CIFAR-10 |
|---|---|---|
| $\alpha_{\exp}$ | 0.96 | 0.98 |
| $\alpha[i]$ | $10^{-4}\cdot\alpha_{\exp}^{-i}$ | $10^{-4}\cdot\alpha_{\exp}^{-i}$ |
| $\mu^{(d,b,k)}[i]$ | $0.1\alpha[i]$ for $k=0,1,2,3$ 
 $10\alpha[i]$ for $k=4$ | $0.5\alpha[i]$ for $k=0,1,2,3$ 
 $5\alpha[i]$ for $k=4$ |
| $\mu^{(E,k)}[i]$ | $\begin{bmatrix} 1 & 1 & 1 & 10^2 & 0 \end{bmatrix}10^{-9}\alpha[i]$ | $\begin{bmatrix} 1 & 1 & 1 & 10 & 10^3 \end{bmatrix}10^{-11}\alpha[i]$ |
| $\mu_{\ell_1}^{(k)}[i]$ | $\begin{bmatrix} 1 & 1 & 1 & 10 & 0 \end{bmatrix}10^{-11}\alpha[i]$ | $\begin{bmatrix} 1 & 1 & 1 & 10 & 0 \end{bmatrix}10^{-13}\alpha[i]$ |
| $\sigma_{\mathbf{W}}$ | $\sqrt{\frac{1}{6}}$ | $\sqrt{\frac{1}{6}}$ |
| $\sigma_{\mathbf{R}_{lin}}$ | 1 | $1e-3$ |
| $\sigma_{\mathbf{R}_{local}}$ | 1 | $1e-1$ |
| $\lambda_d$ | 0.99999 | 0.99999 |
| $\lambda_E$ | 0.99999 | 0.99999 |
| $\lambda_R$ | $3e-2$ | $3e-2$ |
| $\beta$ | $1e-8$ | $1e-6$ |
| $\epsilon_L$ | $1e-8$ | $1e-8$ |
| $\epsilon$ | $1e-5$ | $1e-5$ |
| $B$ | 16 | 16 |
| $p$ | N/A | N/A |

### I.6 Implementation details for Direct Feedback Alignment (DFA) and backpropagation training

This section presents further details on the numerical experiments comparing Direct Feedback Alignment (DFA) and Backpropagation (BP) methods, conducted under the same training conditions and number of epochs as those used for our proposed EBD algorithm. The results of these experiments are provided in Table 1 . We also include the DFA+E method, which extends DFA by incorporating correlative entropy regularization similar to the EBD. Note that, when the update on the $\mathbf{R}_{\mathbf{h}^{(k)}\epsilon}$ is fixed to its initialization, the EBD algorithm reduces to standard DFA.

For BP-based models trained on MNIST, CIFAR-10 and CIFAR-100, we used the Adam optimizer with hyperparameters $\beta_1 = 0.9$, $\beta_2 = 0.999$, and $\epsilon = 10^{-8}$ [57]. For DFA and DFA+E models, we again used the Adam optimizer for CNN and LC models, while MLP models were trained using SGD with momentum.

In Tables 17 and 18, we detail the hyperparameters for models trained with BP, DFA, and DFA+E update rules on MNIST and CIFAR-10 respectively. In Table 19, we give the hyperparameters for the CNN model (detailed in Table 7) trained with with BP and DFA. Some of the learning rate and the learning rate decay values or methodologies are linked to the tables corresponding to the hyperparameter details of its EBD counterpart, where the same method is also utilized for its DFA or DFA+E counterpart. Unlinked values denote a constant value applied to each layer, or the step decay multiplier applied per epoch. Additionally, sparsity inducing losses are not utilized for BP, DFA and DFA+E models.

Table 17: Hyperparameter details for models trained on the MNIST dataset, including learning rate, L2 regularization coefficient, learning rate decay, and number of epochs for MLP, CNN, and LC models using BP, DFA, and DFA+E methods.

| Model | Method | Learning Rate ($\mu^{(d,b,k)}$) | L2 Reg. Coef. | LR Decay ($\alpha_{\text{exp}}$) | Epochs |
|---|---|---|---|---|---|
| MLP | BP | $5e-5$ | $1e-5$ | 0.96 | 120 |
| | DFA | Table-13 | Table-13 | Table-13 | 120 |
| | DFA+E | Table-13 | Table-13 | Table-13 | 120 |
| CNN | BP | $5e-5$ | $1e-8$ | 0.97 | 100 |
| | DFA | Table-15 | $1e-8$ | 0.97 | 100 |
| | DFA+E | Table-15 | $1e-8$ | 0.97 | 100 |
| LC | BP | $5e-5$ | $1e-8$ | 0.96 | 100 |
| | DFA | Table-16 | $1e-8$ | 0.96 | 100 |
| | DFA+E | Table-16 | $1e-8$ | 0.96 | 100 |

Table 18: Hyperparameter details for models trained on the CIFAR-10 dataset, including learning rate, L2 regularization coefficient, learning rate decay, and number of epochs for MLP, CNN, and LC models using BP, DFA, and DFA+E methods.

| Model | Method | Learning Rate ($\mu^{(d,b,k)}$) | L2 Reg. Coef. | LR Decay ($\alpha_{\text{exp}}$) | Epochs |
|---|---|---|---|---|---|
| MLP | BP | $5e-5$ | $1e-5$ | 0.85 | 120 |
| | DFA | Table-13 | 0 | Table-13 | 120 |
| | DFA+E | Table-13 | 0 | Table-13 | 120 |
| CNN | BP | $5e-5$ | $1e-5$ | 0.92 | 200 |
| | DFA | Table-15 | $1e-5$ | 0.97 | 200 |
| | DFA+E | Table-15 | $1e-5$ | 0.97 | 200 |
| LC | BP | $1e-4$ | $1e-6$ | 0.90 | 200 |
| | DFA | Table-16 | $1e-6$ | 0.96 | 200 |
| | DFA+E | Table-16 | $1e-6$ | 0.96 | 200 |

Table 19: Hyperparameter details for the CNN model trained on the CIFAR-100 dataset, including learning rate, L2 regularization coefficient, learning rate decay, dropout probability and number of epochs using BP and DFA methods.

| Model | Method | Learning Rate ($\mu^{(d,b,k)}$) | L2 Reg. Coef. | LR Decay ($\alpha_{\text{exp}}$) | Dropout ($p$) | Epochs |
|---|---|---|---|---|---|---|
| CNN | BP | $5e-5$ | $1e-5$ | 0.97 | 0.5 | 200 |
| | DFA | $[0.1\ 0.1\ 0.1\ 1\ 1\ 50] \cdot 0.25\alpha[i]$ | 0 | 0.95 | 0.075 | 300 |

## I.7 Runtime comparisons for the update rules

In this section, we present the relative average runtimes from the simulations, normalized to BP for the MNIST and CIFAR-10 models in Tables 20 and 21 respectively, for the models that we implemented and demonstrated their performance in Table 1.

The results show that entropy regularization in both EBD and DFA+E more than doubles the average runtime. However, these runtimes could be significantly improved by optimizing the implementation of the entropy gradient terms, specifically by avoiding repeated matrix inverse calculations. A more efficient approach would involve directly updating the inverses of the correlation matrices instead of recalculating both the matrices and their inverses at each step. This strategy aligns with the CorInfoMax-(EP/EBD) network structure. Nonetheless, we chose not to pursue this optimization, as CorInfoMax networks already employ it effectively.

The efficiency of the DFA, DFA+E, and EBD methods can be further enhanced through low-level optimizations and improved implementations.

Table 20: Average Runtimes in MNIST (relative to BP)

| Model | DFA | DFA+E | BP | EBD |
|-------|-----|-------|-----|-----|
| MLP | 3.40 | 7.68 | 1.0 | 8.06 |
| CNN | 1.68 | 2.95 | 1.0 | 3.85 |
| LC | 1.61 | 3.57 | 1.0 | 3.54 |

Table 21: Average Runtimes in CIFAR-10 (relative to BP)

| Model | DFA | DFA+E | BP | EBD |
|-------|-----|-------|-----|-----|
| MLP | 2.85 | 6.94 | 1.0 | 7.61 |
| CNN | 2.10 | 3.24 | 1.0 | 4.11 |
| LC | 1.35 | 2.01 | 1.0 | 2.41 |

## I.8 Reproducibility

To facilitate the reproducibility of our results, we have included the following:

  i. Detailed information on the derivation of the weight and bias updates of the Error Broadcast and Decorrelation (EBD) Algorithm for various networks in Appendix C for MLPs, D.1 for CNNs, D.2 for LCs,

 ii. Full list of hyperparameters used in the experiments in Appendix I.2.5, I.3.4, I.4.4, I.5.4,

 iii. Algorithm descriptions for CorInfoMax Error Broadcast and Decorrelation (CorInfoMax-EBD) Algorithm in pseudo-code format in Appendix I.2.2,

 iv. Python scripts, Jupyter notebooks, and bash scripts for replicating the individual experiments and reported results are included in the supplementary zip file.

## I.9 Computational resources

All experiments were conducted within a High-Performance Computing (HPC) facility. Each experimental run utilized a single NVIDIA Tesla V100 GPU equipped with 32GB of HBM2 memory. To provide context on execution times for our proposed CorInfoMax-EBD models:

  • Training the 3-Layer CorInfoMax-EBD model (as described in Appendix I.2.5) for 30 epochs required approximately 22 hours.

  • Training the 10-Layer CorInfoMax-EBD model (as described in Appendix I.2.6) for 100 epochs took approximately 75 hours. Execution times for other models (MLP, CNN, LC) and baseline methods were generally shorter; relative runtime comparisons are provided in Appendix I.7.

### I.10 Accuracy and loss curves

Figures 6 and 7 present the training/test accuracy and MSE loss curves over epochs for the CIFAR-10 and MNIST datasets. Solid lines represent test curves; dashed lines denote training curves.

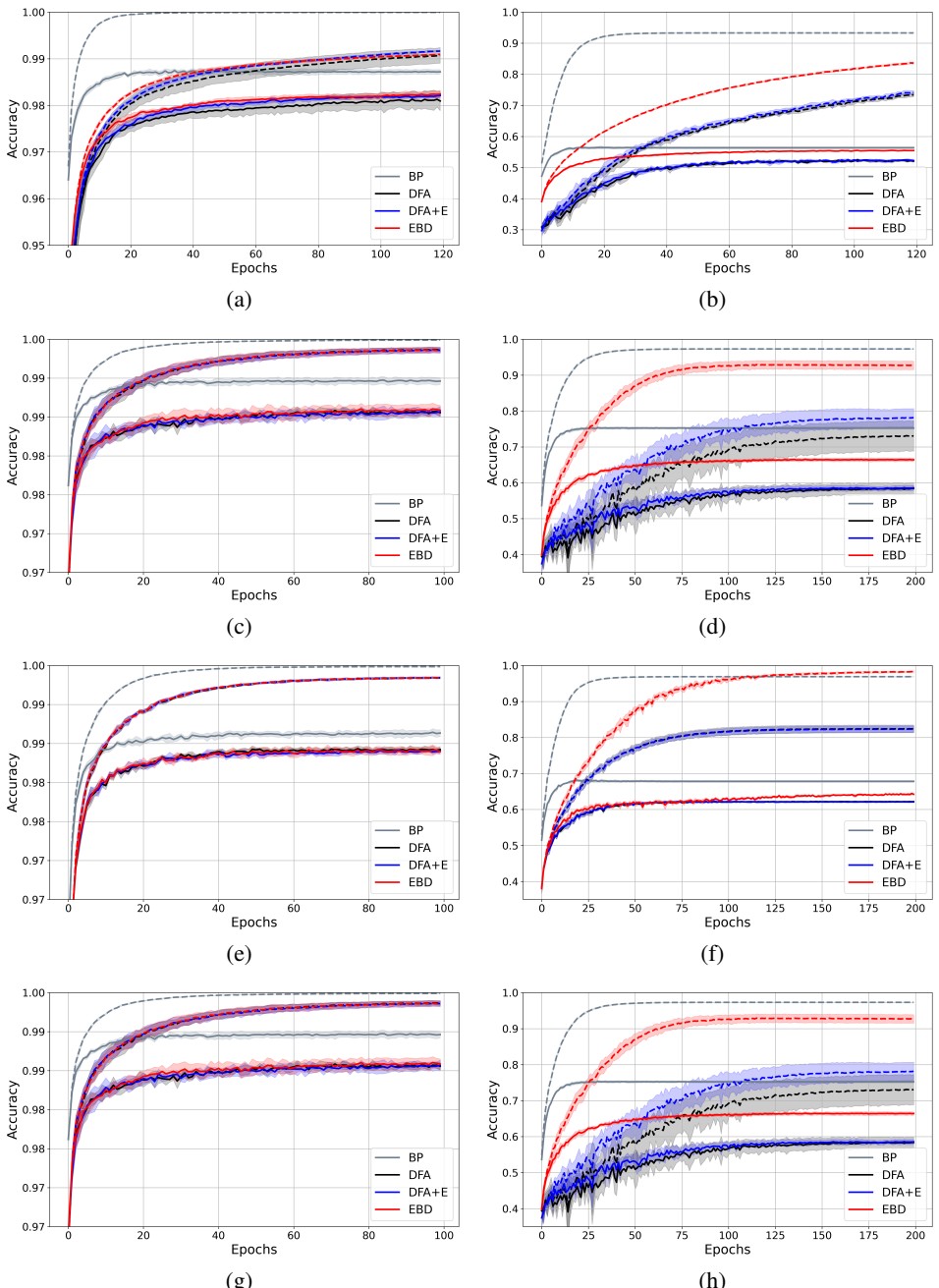

Figure 6: Train and test accuracies plotted as a function of algorithm epochs for various update rules (averaged over $n = 5$ runs associated with the corresponding $\pm$ std envelopes) for the (a) MLP on MNIST (b) MLP on CIFAR-10 (c) CNN on MNIST (d) CNN on CIFAR-10 (e) LC on MNIST (f) LC on CIFAR-10 (g) 10-Layer CorInfoMax-EBD with batchsize=1 on MNIST (h) 10-Layer CorInfoMax-EBD with batchsize=1 on CIFAR-10.

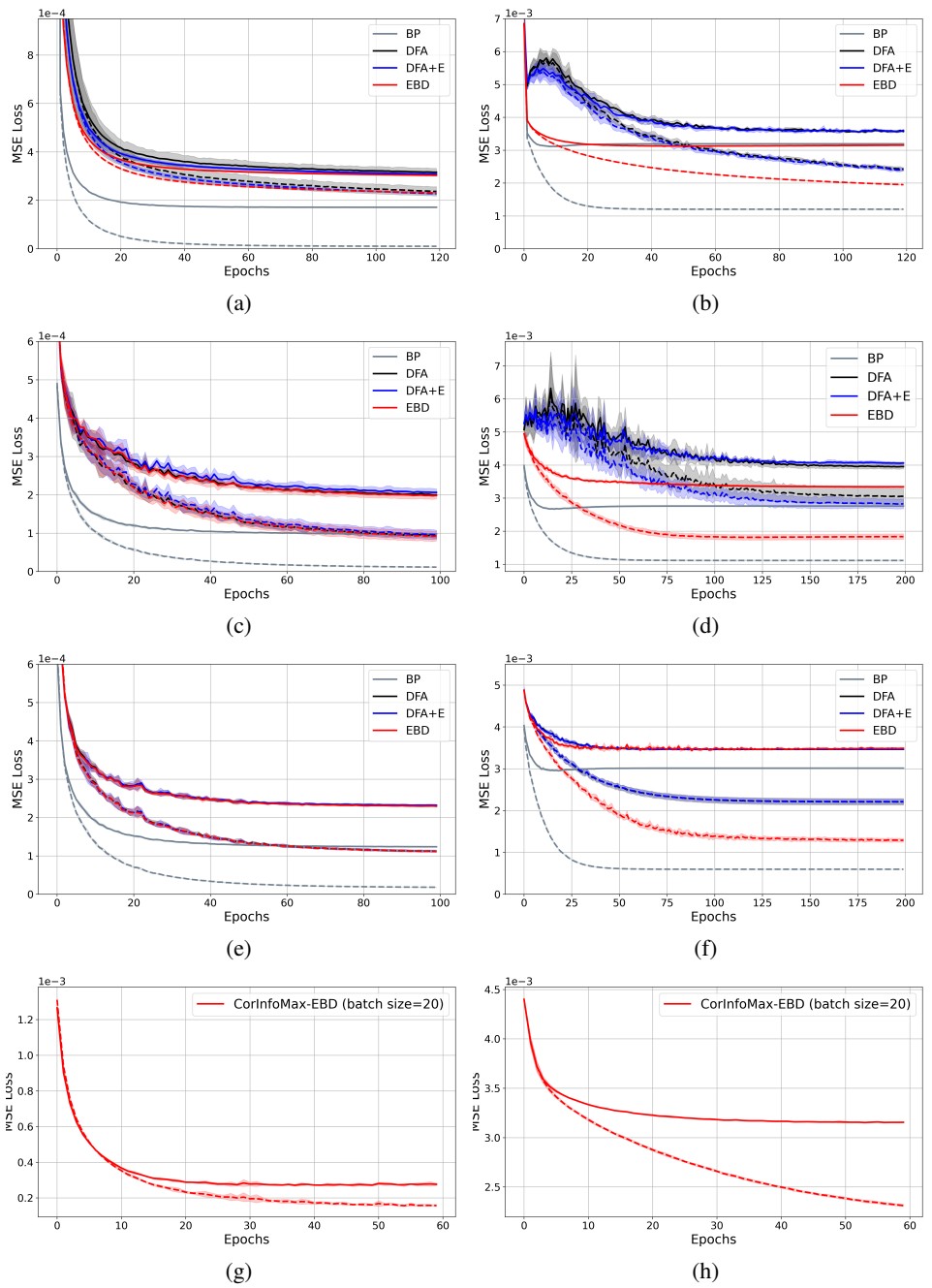

Figure 7: Train and test mean squared errors (MSE) plotted as a function of algorithm epochs for various update rules (averaged over $n = 5$ runs associated with the corresponding $\pm$ std envelopes) for the (a) MLP on MNIST (b) MLP on CIFAR-10 (c) CNN on MNIST (d) CNN on CIFAR-10 (e) LC on MNIST (f) LC on CIFAR-10 (g) 3-Layer CorInfoMax-EBD with batchsize=20 on MNIST (h) 3-Layer CorInfoMax-EBD with batchsize=20 on CIFAR-10.

Figure 8 shows the training/test accuracy curves over epochs in the 10-Layer CorInfoMax-EBD numerical experiments for the CIFAR-10 and MNIST datasets. Furthermore, Figure 9 presents the training/test accuracy curves over epochs for the CNN model trained with the CIFAR-100 dataset. Solid lines represent test curves; dashed lines denote training curves.

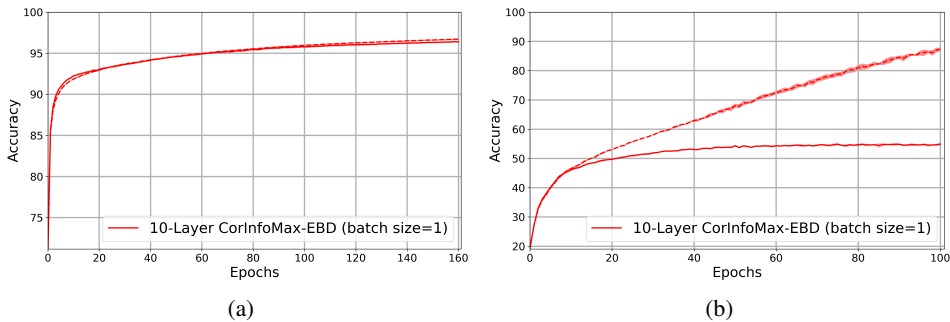

(a)                                                        (b)

Figure 8: Train and test accuracies plotted as a function of algorithm epochs (averaged over $n = 5$ runs associated with the corresponding $\pm$ std envelopes) for training with (a) 10-Layer CorInfoMax-EBD with batchsize=1 on MNIST (b) 10-Layer CorInfoMax-EBD with batchsize=1 on CIFAR-10.

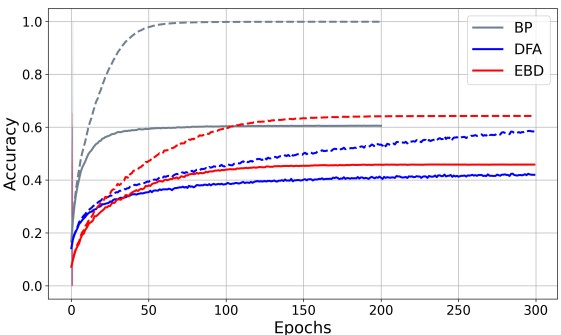

Figure 9: Train and test accuracies plotted as a function of algorithm epochs for various update rules (averaged over $n = 5$ runs associated with the corresponding $\pm$ std envelopes) for training with the CNN network on CIFAR-100.

## J   Calculation of the correlation between layer activations and output error

Figure 1c illustrates the decrease in the average absolute correlation between hidden activations and output error during backpropagation, using a Multi-layer Perceptron (MLP) model with the architecture outlined in Table 5, on the CIFAR-10 dataset. Details for the MSE based training and the Cross-Entropy based training are explained in Appendices J.1 and J.2 respectively.

### J.1   Correlation in the mean squared error (MSE) criterion-based training

The MLP models are trained using the Stochastic Gradient Descent (SGD) optimizer with a small learning rate of $10^{-4}$ and the MSE criterion. In both plots, the initial value represents the correlation before training begins. The reduction in correlation observed during training provides insight into the core principle of the EBD algorithm.

To compute these correlations, we apply a batched version of Welford's algorithm [58], which efficiently calculates the Pearson correlation coefficient between hidden activations and errors in a memory-efficient way by using streaming statistics.

Welford's algorithm works by accumulating the necessary statistics (e.g., sums and sums of squares) across batches of data and finalizing the correlation computation only after all data has been processed, avoiding the need to store all hidden activations simultaneously.

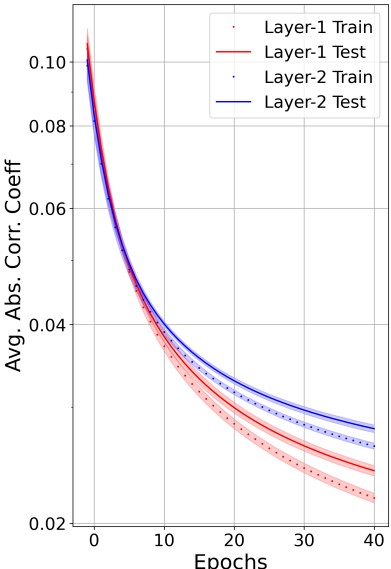

Figure 10: The evolution of the average absolute correlation between layer activations and the error signal during backpropagation training of an MLP with two hidden layers (using the MSE criterion) on the MNIST dataset, showing the correlation decrease over epochs, on both the training and test sets.

Given the hidden activations $\mathbf{h}^{(k)} \in \mathbb{R}^{b \times N^{(k)}}$, where $b$ is the batch size and $N^{(k)}$ is the number of hidden units, and the errors $\boldsymbol{\epsilon} \in \mathbb{R}^{b \times k}$, where $k$ is the number of output dimensions (e.g., classes); the goal is to compute the Pearson correlation coefficient between activations $h_i$ for each hidden unit $i$ and the corresponding error values across all samples as:

$$\rho^{(k)} = \frac{\mathrm{Cov}(\mathbf{h}^{(k)}, \boldsymbol{\epsilon})}{\sqrt{\sigma^2_{\mathbf{h}^{(k)}}} \sqrt{\sigma^2_{\boldsymbol{\epsilon}}} + \epsilon}$$

where $\epsilon$ is a small constant for numerical stability. Finally, we compute the average of the absolute values of the correlation coefficients for each hidden layer $k$ to generate the corresponding plots.

Figure 10 shows the correlation throughout training on the MNIST dataset, employing the MLP model detailed in Table 4, where we observe the correlation decay behavior similar to Figure 1c.

To verify that the correlation–decay phenomenon observed for fully-connected networks extends to convolutional architectures, we trained a compact five-layer CNN consisting of three $\mathrm{Conv}+\mathrm{ReLU}+\mathrm{MaxPool}$ blocks (with 32, 64, and 128 $3\times3$ filters, respectively) followed by a fully–connected layer with 512 hidden units and an output layer with 10 logits. The network was optimized for 30 epochs on $\mathrm{CIFAR}-10$ using mini-batch SGD ($\eta = 10^{-3}$, momentum 0.9) and the mean-squared error criterion on one-hot labels. After every forward pass we streamed the Pearson correlation between each hidden activation vector and the output error using the batched Welford estimator. Figure 11 plots the epoch-wise evolution of the average absolute correlation coefficient for both training and test data. As with the MLP in Figure 1c, all layers exhibit a pronounced monotonic decline, confirming that back-propagation progressively enforces the stochastic orthogonality of hidden activations and output errors in convolutional networks as well, as expected.

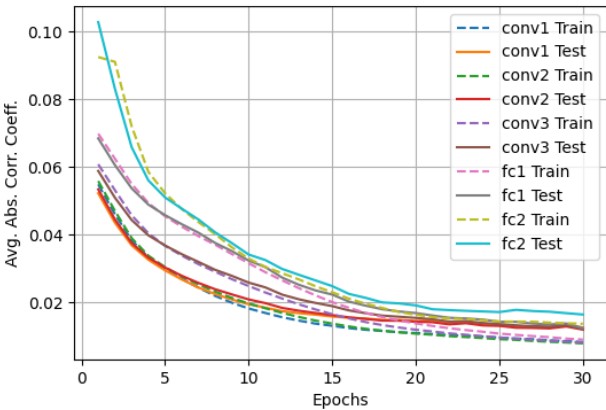

Figure 11: The evolution of the average absolute correlation between layer activations and the error signal during back-propagation training of a CNN on CIFAR–10. The CNN contains three convolutional layers and two fully–connected layers and is trained with the MSE criterion. The correlation decreases over epochs on both the training and test sets, mirroring the behavior observed for the MLP architecture and supporting the generality of the correlation decay across architectures.

## J.2 Correlation in the cross-entropy criterion-based training

Although the stochastic orthogonality property is specifically associated with the MSE loss, we also explored the dynamics of cross-correlation between layer activations and output errors when cross-entropy is used as the training criterion.

With the same experimental setup as described in Appendix J.1, but replacing the MSE loss with cross-entropy, we obtained the correlation evolution curves shown in Figure 12a for CIFAR-10 and in Figure 12b for MNIST dataset. Notably, the correlation between layer activations and output errors still decreases over epochs, despite the change in the loss function.

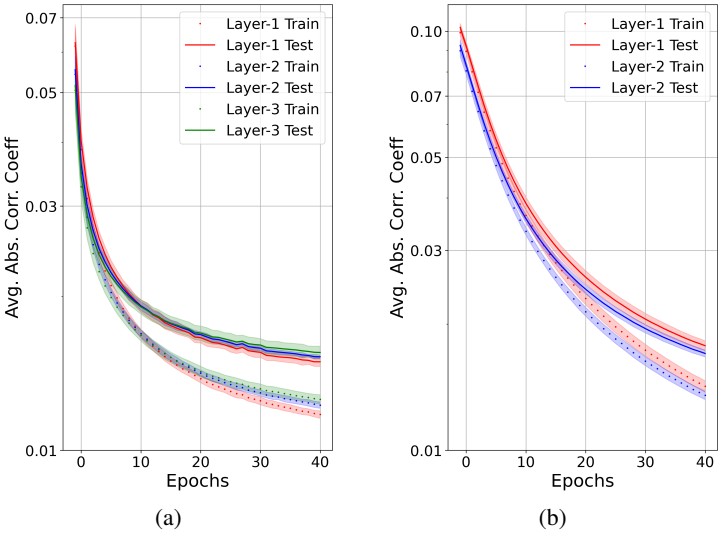

(a)                                    (b)

Figure 12: Evolution of the average absolute correlation between layer activations and output errors during backpropagation training of an MLP with three hidden layers, trained using cross-entropy loss. (a) CIFAR-10 dataset and (b) MNIST dataset. Despite the use of cross-entropy, the correlation decreases similarly to the MSE criterion.

# NeurIPS Paper Checklist

