# OpenReview forum: "Error Broadcast and Decorrelation as a Potential Artificial and Natural Learning Mechanism"
_NeurIPS.cc/2025/Conference — NeurIPS 2025 spotlight_

### Official Review · Reviewer_i7Bt · 2025-06-26

**Clarity:** 3
**Significance:** 3
**Originality:** 3
**Rating:** 5
**Confidence:** 3

**Summary:**

The submission presents a new method for training neural networks that aims to minimize squared error without relying on standard backpropagation, by minimizing the correlation between the activations of individual layers and the network's residuals. Theoretical arguments based on minimum mean square error estimation theory are provided to justify this method. Experimental results show that the proposed method outperforms direct feedback alignment and global error vector broadcasting when classifying on MNIST and CIFAR-10, using MLPs, CNNs, and locally connected networks. However, classification accuracy remains lower than that achieved using backbropagation. The submission also shows that the proposed approach can be profitably applied in conjunction with more biologically plausible CorInfoMax networks.

**Questions:**

N/A

**Ethical Concerns:**

["NO or VERY MINOR ethics concerns only"]

**Final Justification:**

I have raised my score from Borderline Accept to Accept after consideration of the rebuttal.

**Limitations:**

Yes

**Paper Formatting Concerns:**

I did not spot any formatting issues. However, in my view, it is questionable whether a paper with such a substantial appendix is suitable for consideration at a conference. It is unreasonable to expect conference reviewers to check work of this size. This paper should have been submitted to a journal instead.

**Quality:**

3

**Strengths And Weaknesses:**

The proposed approach is interesting, particularly considering the theory that supports it.

Although theoretically motivated, the proposed algorithm seems to drop one of the two components of the derivative of the layer-specific loss without discussion of how this affects the theory.

The algorithm is based on squared error, but the experiments tackle classification problems. It seems that deep regression would be a more suitable application of the proposed method.

The experiments are limited to MNIST and CIFAR-10.

The hyperparameter search grids for the proposed method are different for MNIST and CIFAR-10.

The comparison to backpropagation is limited to fairly shallow networks.

It is not stated whether cross-entropy was used for backpropagation. If the squared error was used, this should be stated, and results for cross-entropy should be included.

Some further related work that the authors should consider including in their discussion:

Will Xiao, Honglin Chen, Qianli Liao, and Tomaso Poggio. Biologically-plausible learning algorithms can scale to large datasets. In International Conference on Learning Representations (ICLR), 2019.

Frenkel, C., Lefebvre, M., & Bol, D. (2021). Learning without feedback: Fixed random learning signals allow for feedforward training of deep neural networks. Frontiers in neuroscience, 15, 629892.

Ma, W. D. K., Lewis, J. P., & Kleijn, W. B. (2020, April). The HSIC bottleneck: Deep learning without back-propagation. In Proceedings of the AAAI conference on artificial intelligence (Vol. 34, No. 04, pp. 5085-5092).

Akrout, M., Wilson, C., Humphreys, P., Lillicrap, T., & Tweed, D. B. (2019). Deep learning without weight transport. Advances in neural information processing systems, 32.

---

> ### Author Rebuttal · Authors · 2025-07-30
>
> We would like to thank the reviewer for the constructive comments.
>
> ### Strengths And Weaknesses:
>
> >The proposed approach is interesting, particularly considering the theory that supports it.
>
> We appreciate the reviewer's positive remarks.
>
> >...the proposed algorithm seems to drop one of the two components of the derivative ....
>
> We understand the reviewer’s concern. As described in the article, the omission of the second component is deliberate and the impact of this ommision is analyzed in Appendix E. In Section 2.3 and Appendix C, we first show that the decorrelation gradient naturally splits into two terms, $(\Delta W_1,\Delta b_1)$ that depend only on local activations and the broadcast error, and $(\Delta W_2,\Delta b_2)$ that would require backpropagating derivatives of the error (Appendix E.2). Keeping only $\Delta W_1$ preserves strict locality. Our experimental results confirm that negative of this component essentially points in a descent direction: the cosine similarity of $\Delta W_1$ with the \emph{full} (two-term) gradient remains almost always positive throughout training (Fig. 4), confirming that -$\Delta W_1$ is essentially a descent direction (Appendix E.2). Moreover, Appendix E.1 shows that $\Delta W_1$ is also positively aligned with the backpropagation gradient (Figures 2 and 3). Furthermore, as demonstrated in Section 4, networks trained with the truncated update match or outperform state-of-the-art broadcast methods on all benchmarks.
>
>
> >The algorithm is based on squared error, but the experiments tackle classification problems. It seems that deep regression would be a more suitable application of the proposed method.
>
> We thank the reviewer for this thoughtful observation. We agree that mean squared error (MSE) is a natural fit for regression problems where the target is continuous. However, our choice to evaluate on classification tasks was motivated by two main considerations: (i) the need for direct comparison with prior error broadcast–based methods, which predominantly evaluate on classification benchmarks; and (ii) the empirical observation that the core decorrelation mechanism underlying EBD remains effective even when cross-entropy loss is used.
>
> Specifically, in Appendix J.2, we demonstrate that the decorrelation of layer representations still occurs under cross-entropy loss, suggesting that the surrogate EBD principle extends beyond the squared error setting. This helps explain the strong empirical performance we observe in classification tasks, despite the algorithm being theoretically grounded in MSE. We view this as an encouraging indication that the EBD framework may generalize to broader loss functions. Nonetheless, formalizing this generalization remains an important direction for future theoretical work.
>
> >The experiments are limited to MNIST and CIFAR-10.
>
> This work primarily introduces a theoretically grounded learning framework. The numerical experiments are designed to validate the key concepts and demonstrate the performance advantages of our method over existing state-of-the-art error broadcast algorithms. We chose MNIST and CIFAR‑10 because they are the standard testbeds for broadcast‑based algorithms, and EBD already exceeds their state‑of‑the‑art results on both.
>
> To further assess scalability, we extended experiments to CIFAR‑100 using the same CNN architecture as in the Dropout paper [1], achieving 44.1% top‑1 accuracy—surpassing the original DFA result (41%) [2] and the best reported DFA result (43.7%) that we are aware of on this architecture [3]. In the revised manuscript, we will include these CIFAR‑100 results along with additional hyperparameter tuning, and directly compare them to both existing DFA results in the literature and a standard backpropagation baseline.
>
>
> References:
> - [1] Srivastava et al. "Dropout: A Simple Way to Prevent Neural Networks from Overfitting", JMLR, 2014.
> - [2] Crafton, Brian et al. “Direct Feedback Alignment With Sparse Connections for Local Learning.” Frontiers in Neuroscience 13, 2019.
> - [3] Nøkland, Arild. "Direct feedback alignment provides learning in deep neural networks." Neurips, 2016.
>
>
> >The hyperparameter search grids for the proposed method are different for MNIST and CIFAR-10.
>
> While the hyperparameter grids for MNIST and CIFAR-10 differ slightly, this is a common and justified practice due to the distinct nature and complexity of the two datasets. Our goal was to ensure that the proposed method operates competitively under reasonable settings for each benchmark. The search process remained consistent in scope and methodology across datasets.
>
> >The comparison to backpropagation is limited to fairly shallow networks.
>
> As noted in the **Impact and Limitations** part of Section 5, our experiments with 10-layer CorInfoMax-EBD provide initial evidence that EBD has potential to scale to deeper architectures. While this demonstrates the depth scalability of the proposed framework, exploring broader scaling trends across architectures and input domains is an exciting direction for future work. We note that BP also initially struggled with challenges related to network depth and data scale, but these were eventually overcome through advances in architecture and optimization.  Similarly, we anticipate that similar innovations will further boost EBD performance on larger datasets going forward.
>
> >It is not stated whether cross-entropy was used for backpropagation. If the squared error was used, this should be stated, and results for cross-entropy should be included.
>
> BP was trained with MSE to maintain consistency with the MSE‑based objective used by the EBD algorithm. In the revised article, we will clarify that MSE criterion has been used for BP results. We'll also provide BP results with cross-entropy criterion. Our initial experiments with the cross-entropy loss yield test accuracy comparable to, or slightly better than, the MSE-based baseline—for example, 56.89% test accuracy for the MLP on the CIFAR-10 dataset, compared to 56.4% with MSE loss (see Table 1).  However, we will do broader hyperparameter sweep for the revision.
>
> >Some further related work that the authors should consider including in their discussion:
>
> >Will Xiao, Honglin Chen, Qianli Liao, and Tomaso Poggio. Biologically-plausible learning algorithms can scale to large datasets. In International Conference on Learning Representations (ICLR), 2019.
>
> >Frenkel, C., Lefebvre, M., & Bol, D. (2021). Learning without feedback: Fixed random learning signals allow for feedforward training of deep neural networks. Frontiers in neuroscience, 15, 629892.
>
> >Ma, W. D. K., Lewis, J. P., & Kleijn, W. B. (2020, April). The HSIC bottleneck: Deep learning without back-propagation. In Proceedings of the AAAI conference on artificial intelligence (Vol. 34, No. 04, pp. 5085-5092).
>
> >Akrout, M., Wilson, C., Humphreys, P., Lillicrap, T., & Tweed, D. B. (2019). Deep learning without weight transport. Advances in neural information processing systems, 32.
>
> We appreciate the reviewer’s relevant literature suggestions. In the revised article, we will expand Section 1.1 (Related Work and Contributions) by increasing the discussion on the previous work including the references pointed by the reviewer.
>
> ---
> ### Paper Formatting Concerns:
> >I did not spot any formatting issues. However, in my view, it is questionable whether a paper with such a substantial appendix is suitable for consideration at a conference. It is unreasonable to expect conference reviewers to check work of this size. This paper should have been submitted to a journal instead.
>
> We thank the reviewer for the feedback. While we acknowledge that our appendix is longer than average, it is in line with many NeurIPS submissions—especially those involving theoretical developments or extensive experimental validations—which often include appendices exceeding ours in length. In our case, the appendix is intended to provide optional background material, full derivations, and additional empirical details for completeness and reproducibility.
>
> Importantly, we have taken care to ensure that the main paper is self-contained: all key ideas, contributions, and results are clearly presented within the 9-page main submission, without requiring the reader to consult the appendix.
> We view NeurIPS as the most appropriate venue for presenting this framework to the research community, especially given that prior foundational work on error broadcast algorithms such as DFA and GEVB was also introduced at NeurIPS.

---

> > ### Comment · Reviewer_i7Bt · 2025-08-06
> > **Response to the rebuttal**
> >
> > Thank you very much for the responses to the points I raised.

---

> > > ### Author Response · Authors · 2025-08-09
> > > **Thanks to the Reviewers**
> > >
> > > We thank the reviewers for their time, as well as constructive and positive feedback. Your comments have been instrumental in improving the clarity, scope, and presentation of this work.

---

### Official Review · Reviewer_Xg6E · 2025-06-29

**Clarity:** 2
**Significance:** 3
**Originality:** 3
**Rating:** 5
**Confidence:** 2

**Summary:**

The authors present EBD (Error broadcast and decorrelation), a way to improve existing error broadcasting techniques (to address the credit assignment problem, i.e. how to assign importance - a different phrasing of "how to train NNs").
It works by altering the training objective to be layer-specific, and adjust network weights by lowering the correlation between their activations and the errors in broadcast outputs.
The authors ground their method with theoretic insights and test it on vision datasets, showing a PoC on small (MLP / Convolutional / Locally-connceted) networks using MNIST and CIFAR10.

**Questions:**

1. Why is a classification model trained using MSE? Is this standard practice in EB studies? Or is it something the authors employ for the MSEE orthogonality grounding?


2. Are there any evidence that EBD or EB in general are biologically plausible, EXCEPT for their lack of requirement for weight symmetry?


3. Suggestion: coming from the world of neural networks, I haven't witnessed the usage of the term "credit assignment problem" to weight learning. I think it would be best to drop this confusing phrasing from the abstract (despite it being explained in the third line of the intro - maybe leave it there).

**Ethical Concerns:**

["NO or VERY MINOR ethics concerns only"]

**Final Justification:**

It seems to me (as a researcher from DL, definitely not an expert in the error brodcasting literature of this paper) that this paper is solid and deserves attention from the researchers in this sub-field. The authors responded to all my comments and have promised to revise the many points which I have found lacking. I therefore increase my score and think that this paper should be accepted.

**Limitations:**

yes (except for scale limitations discussed in the weaknesses section)

**Quality:**

3

**Strengths And Weaknesses:**

Strengths:

1. Proper theoretical grounding and research method, with high relevance to its sub-field: The paper presents a new approach in the field of EB, grounds it in theory, and shows basic empirical results. This paper seems of high relevance to researchers working on this subject.

2. Empirical improvement: While limited in terms of datasets, the benchmarking made by the authors shows an improvement over existing methods, making any further discussion of this method relevant to other works in this sub-field.


Weaknesses:

1. Not accessible to non-experts in this sub-field: As someone who is less familiar with the research work of Error broadcasting (also expressed in my low confidence score), I find it difficult that the paper isn't self-contained, and many parts of it still seem rather unclear to me.
To be more specific, I felt that this paper "hits hard" from the introduction --- it is very dense, throwing A LOT of new information on the motivation on EB and the theoretical background of the current work, without discussing much on how more basic EB work. I would've expected a preliminary section on EB *before* the authors dive in deep to their contributions in sec 2.2, which would further emphasize exactly what did the authors add *on-top* of existing EB work.
An additional example is the constant mentioning of the "three-factor learning rule" without defining what it is (having read what it is out of the paper, it is unclear to me why existing methods as backprop can't be considered as three-factor learning).

2. Limited benchmarks: The results are only measured on MNIST on CIFAR10. The authors state this as a limitation, and while I *don't* hold it against the authors as they show a new direction in this work, I do think it would make sense to *explain* why the method wasn't (or can't be) tested on other simple datasets with similar resolutions to CIFAR10 (e.g., CIFAR100, Tiny imagenet, Caltech-256, etc), and not merely shrug it off as a "direction for future work". Even if other baseline results aren't available, seeing high EBD results on other datasets can create more faith in the method.

3. Non-uniform / complex notation make it even more difficult to read: For instance, in eq 1 - u(k) should be mentioned as the pre-activation output; across the different equations, where Y is some vector, it is unclear if the indexing Y[m] or Y[mB] refers to a batch index (m) or an absolute index (mB).

In terms of explaining my rating:
Overall, the first weakness (lack of accessibility to non-experts) is my major concern with this paper being accepted to the conference. I would also like to have a better intuition of weather this method scales to larger datasets (It's not clear it would scale easily, as I found some conflicting evidence in literature [1,2]). Nevertheless, because of its relevance to researchers within the field, I believe it adds value.


[1] "Assessing the Scalability of Biologically-Motivated Deep Learning Algorithms and Architecture", https://arxiv.org/abs/1807.04587

[2] "Biologically-plausible learning algorithms can scale to large datasets", https://www.semanticscholar.org/paper/Biologically-plausible-learning-algorithms-can-to-Xiao-Chen/2c26cca611ef1403ca81df238d04d0ef4a584a86

---

> ### Author Rebuttal · Authors · 2025-07-30
>
> We would like to the reviewer for the constructive comments.
>
> ### Strengths:
>
> We appreciate the positive comments by the reviewer.
>
> ---
> ### Weaknesses:
> >1. Not accessible to non-experts:... a preliminary section on EB ...  "three-factor learning rule"
>
> We appreciate the feedback by the reviewer. Our work builds upon a growing body of research on error broadcasting (EB), and more broadly biologically plausible learning, and we recognize that its terminology and motivations may not yet be broadly familiar outside the immediate subfield. Given the length limitations of the conference and diversity of topics within the proposed framework, we were unable to provide in-depth introduction to the subject. Nonethless, based on the reviewer's feedback, in the revised article,  we updated the introduction section to improve its clarity for more general audience:
>
> - Added a concise preliminary description of basic error broadcasting in the paragraph (3) introducing it:*"These methods involve broadcasting the global output error directly to all layers, often through random projections or fixed pathways, without relying on precise backward paths or symmetric weights"*
> - Emphasized the contributions of EBD by noting that it builds on basic error broadcasting through layer-specific objectives grounded in estimation theory.
> - Defined the three-factor learning rule inline when first mentioned in the advantages paragraph, describing it as an extension of Hebbian plasticity that incorporates a neuromodulatory signal modulating updates based on pre- and postsynaptic activity.
>
> We also note that Section 1.1 contains a survey paragraph on other EB methods, which we will further expand in the revision. We'll further go over the introduction and the other sections to improve clarity.
>
> More specifically, regarding the connection between the standard backpropagation and three-factor learning rule: the backpropagation update expression can be put into a form similar to the three‑factor rule:
> For a layer $l$ with weights $w_{ij}^{(l)}$, presynaptic activity $a_i^{(l-1)}$, membrane potential
> $z_j^{(l)}=\sum\nolimits_i w_{ij}^{(l)}\,a_i^{(l-1)}$,  activation
> $a_j^{(l)}=\phi\!\bigl(z_j^{(l)}\bigr)$, and loss $L$, the gradient‑descent update factors as
>
> $$\Delta W_{ij}^{(l)}=
> -\eta\,\frac{\partial L}{\partial w_{ij}^{(l)}}=
> -\eta\,
> \underset{\text{pre-synaptic}}{a_i^{(l-1)}} \cdot
> \underset{\text{post-synaptic}}{\phi'\bigl(z_j^{(l)}\bigr)} \cdot
> \underset{\text{error / third factor}}{\varepsilon_j^{(l)}},
> $$
> where the back‑propagated error is defined recursively by
> $$
> \varepsilon_j^{(l)}=
> \sum_{k} w_{jk}^{(l+1)}
> \delta_k^{(l+1)},
> $$
> where
> $$
> \delta_k^{(l)}=
> \phi'(z_k^{(l)})
> \varepsilon_k^{(l)}.
> $$
>
> However, neuroscientists would not classify this as a true three‑factor rule. Its “third factor” is not a pure error signal but a neuron‑specific term combining higher‑layer activations and weights, computed by a separate error network whose synapses mirror those of the forward network—an arrangement viewed as biologically implausible. In biology, the third factor is a low‑dimensional, population‑wide neuromodulator delivered without such symmetric pathways.s [13].
>
> To clarify BP and three-factor learning rule connection, as the second sentence in Section 3.1, we inserted the clarification sentence:
>
> *"While backpropagation can be expressed similarly, it is not typically considered a three-factor rule in neuroscience, as its 'third factor' is a locally tailored signal specific to each neuron requiring a biologically implausible dual network with symmetric weights, unlike global neuromodulatory signals [13].    In contrast, EBD update for batchsize $B=1$ in (8)  naturally matches the three-factor structure:...."*
>
> We also included a new Figure in Section 3 to clarify three factor learning rule and its connection with EBD update.
>
> >2. Limited benchmarks:
>
> This work primarily introduces a theoretically grounded learning framework. The numerical experiments are designed to validate the key concepts and demonstrate the performance advantages of our method over existing state-of-the-art error broadcast algorithms. We chose MNIST and CIFAR‑10 because they are the standard testbeds for broadcast‑based algorithms, and EBD already exceeds their state‑of‑the‑art results on both.
>
> We further extended experiments to CIFAR‑100 using the same CNN architecture as in the Dropout paper [1], achieving 44.1% top‑1 accuracy—surpassing the original DFA result (41%) [2] and the best reported DFA result (43.7%) that we are aware of on this architecture [3]. In the revision, we will include these CIFAR‑100 results along with additional hyperparameter tuning.
>
> We note that backpropagation itself once faced similar depth and data‑scale hurdles before architectural and optimization advances closed the gap; we can expect analogous innovations to further boost EBD performance on larger datasets going forward.
>
> Refs:
> - [1] Srivastava et al. "Dropout: A Simple Way to Prevent Neural Networks from Overfitting", JMLR, 2014.
> - [2] Crafton, Brian et al. “Direct Feedback Alignment With Sparse Connections for Local Learning.” Fron. in Neurosc. 2019.
> - [3] Nøkland, Arild. "Direct feedback alignment provides learning in deep neural networks." Neurips, 2016.
>
> >3. Non-uniform/complex notation
>
> We thank the reviewer for this helpful observation. We recognize that clear and consistent notation is critical for accessibility, and we will revise the manuscript to improve readability in response to this and related reviewer comments.
>
> In particular, we will explicitly state that $\mathbf{u}^{(k)}$ as preactivations at layer-$k$. This will be added after Eq. (1), along with a reminder that $k\in\{1, \ldots L\}$ is the layer index.
>
> Regarding indexing, as the reviewer correctly noted $m$ denotes the batch index and $mB+l$ is used to represent the absolute (sequence) index of the $l$-th sample within batch-$m$. This convention is used consistently throughout the mathematical derivations.
>
> To further clarify:
>
> - Quantities such as weights $\mathbf{W}^{(k)}[m]$, biases $\mathbf{b}^{(k)}[m]$, learning rates $\mu^{(k)}[m]$,  and projection matrices $\hat{\mathbf{R}}_{ge}^{(k)}[m]$ are updated per batch, hence indexed by $m$.
> - Batch-structured matrices such as $\mathbf{E}[m]$, containing all output errors for batch-$m$, are similarly batch-indexed.
>
>  To aid clarity, we will add the following in-line clarifications:
>
>  - after Eq. (1): "where $k\in\{1, \ldots L\}$ is the layer index, ..."
>  - at line 143: "where $m$ is the batch index, $\lambda\in [0,1]$ is..."
>  - after Eq. (4): "is the matrix of nonlinearly transformed activations of layer $k$ for batch $m$. In the above equation, $mB+l$ refers to absolute (sequence) index for the $l^{th}$ member of batch-$m$. Furthermore, "
>
> In addition, we will carefully review the manuscript for any further notational inconsistencies and perform minor revisions to improve uniformity and readability without compromising rigor.
>
> ---
> ### Questions:
> >1. Why  trained using MSE?
>
> As the reviewer correctly noted, we use the MSE criterion due to its theoretical grounding in nonlinear MMSE estimation. However, experiments in Appendix J.2 show that correlations between output errors and layer activations still decay when training with cross-entropy loss. This suggests that the decorrelation principle, though derived from the MMSE objective, may extend to other losses. Exploring this theoretically is a promising direction for future work.
>
> >2.evidence EB biologically plausible?
>
> This is a direction we plan to explore further. There is already encouraging evidence that error broadcast–based mechanisms may operate locally in various brain regions. Neuromodulators such as dopamine, serotonin, acetylcholine, and noradrenaline are strong candidates, as they project broadly across cortical and subcortical areas, delivering phasic signals that reach many neurons and synapses in parallel. This shared modulatory input can act as a global third factor in synaptic plasticity.
>
> Reference [a] (ref. [13] in our paper) highlights specific examples: dopaminergic signals in the striatum, cholinergic and dopaminergic modulation in the hippocampus, and broader cortical regions—all supporting the role of global error-like signals in learning. Additionally, long-range projections to apical dendrites of pyramidal neurons may also support global error-projected learning.
>
> Collectively, these findings suggest that three-factor learning rules incorporating error broadcast are plausible biological mechanisms. Our work provides a normative framework for this hypothesis and aims to inspire further experimental and theoretical studies in biologically plausible error-driven learning.
>
> [a]. Gerstner, W. et.al. (2018). Eligibility traces and plasticity on behavioral time scales: experimental support of neohebbian three-factor learning rules. (Reference [13] )
>
> >Suggestion: ..."credit assignment problem"
>
> Thank you for this  suggestion. We understand that the term “credit assignment problem” may not be as commonly used to describe weight learning in all sub-areas of machine learning. However, it has been used broadly in both computational neuroscience and biologically motivated machine learning literature to refer to the challenge of assigning learning responsibility to synapses or parameters. The usage of this terminology date backs to at least  Minsky and Papert's famous perceptron book [i], and it is still used frequenctly in neuro-AI papers (see the references in our article, e.g., [9],[14],[21], ..., and [ii] below). Since this term concisely captures the core problem addressed in our work and may help connect readers across disciplines, we would prefer to keep it in the abstract.
>
> [i] Minsky, M. & Papert, S. (1969). Perceptrons: An Introduction to Computational Geometry.
>
> [ii] Richards,& Lillicrap (2019). Dendritic solutions to the credit assignment problem. Curr. op. in neurobiology

---

> > ### Comment · Reviewer_Xg6E · 2025-08-05
> >
> > I thank the authors for the rebuttal. My concerns have been filled and I (as a non-expert in this specific sub-field) believe that this paper deserves attention from the researchers in this sub-field and should be accepted, pending the promised changes to make it more accessible to non-experts.

---

> > > ### Author Response · Authors · 2025-08-09
> > > **Thanks to the Reviewers**
> > >
> > > We thank the reviewers for their time, as well as constructive and positive feedback. Your comments have been instrumental in improving the clarity, scope, and presentation of this work.

---

### Official Review · Reviewer_Dubd · 2025-06-30

**Clarity:** 3
**Significance:** 3
**Originality:** 3
**Rating:** 5
**Confidence:** 3

**Summary:**

This paper introduces a new theory for biologically plausible credit assignment that does not rely on backpropagation of the error. Established theories based on error broadcasting (EB) can avoid backpropagation but typically do not scale to difficult tasks or large networks. This paper introduces a new strategy termed EBD with is theoretically grounded in optimal MMSE estimators. It improves over existing EB strategies such as DFA as illustrated by thorough numerical simulation results. I congratulate the authors on these results and believe that this paper contributes both important conceptual and theoretical insights to the credit assignment literature.

**Questions:**

Questions:

1. I understand how the EBD surrogate loss function can lead to collapse in the absence of activity regularization. However, I am curious to what extend the regularizers are *necessary*. Without regularization, do all networks eventually collapse? If so, how sensitive are the results to the choice of regularization hyperparameters? What was the reason to choose power normalization?
2. Similarly, I am curious to what extend the method relies on the forward projections explained in Appendix C.5. How does removing these forward projections impact performance?
3. How robust is EBD to different choices of the nonlinearity g(h)?

**Ethical Concerns:**

["NO or VERY MINOR ethics concerns only"]

**Final Justification:**

After reading the author rebuttal and reviews from other reviewers, I maintain that the method developed in this paper is a novel solution to error broadcasting that deserves attention and future investigations.

Especially if the authors address the minor issues with notations and presentation of the results, I recommend this paper for publication.

**Limitations:**

Yes

**Quality:**

3

**Strengths And Weaknesses:**

Strengths:

- analytically grounded theory for learning with broadcasted errors. Setting the feedback weights to the  cross-correlation matrix between hidden activations output errors and the link to decorrelation adds a novel element to existing EB strategies.
- Strong numerical results indicating that the method consistently outperforms DFA and could scale to larger networks.
- The paper is generally well-written (although see below for comment on figures & notation).

Weaknesses:

- The results and model schematic could be illustrated better with some additional figures.
- Mathematical notation, despite rigorous, is hard to follow at times and requires careful reading. This is excellent for the appendix but some expressions could be optimized for readability and clarity in the main text.
- Some controls are missing, specifically with regards to the regularisation terms preventing collapse (see questions below)

---

> ### Author Rebuttal · Authors · 2025-07-30
>
> We would like to the reviewer for the constructive comments.
>
>
> ### Strengths:
>
> >1. analytically grounded theory for learning with broadcasted errors. Setting the feedback weights to the cross-correlation matrix between hidden activations output errors and the link to decorrelation adds a novel element to existing EB strategies.
>
> >2. Strong numerical results indicating that the method consistently outperforms DFA and could scale to larger networks.
>
> >3.The paper is generally well-written (although see below for comment on figures & notation).
>
> We really appreciate these positive comments.
>
> ---
> ### Weaknesses:
>
> >1. The results and model schematic could be illustrated better with some additional figures.
>
> We agree informative schematics would aid clarity. In the revised article, we  added a new figure  to Section 3, which describes EBD  learning as a three‑factor synaptic update. The following is the paragraph that refers to this figure:
>
> *"Figure 2 breaks the EBD weight update (8) into its three interacting factors.  The presynaptic term is the activity $h_j^{(k-1)}$ from the sending unit; the postsynaptic term is $g_i^{\prime(k)}f^{\prime(k)}$ computed from the receiving unit’s own activation; and the modulatory term is the broadcast error $q_i^{(k)}$, derived from the network’s output error $\boldsymbol{\epsilon}$.  Multiplying these three quantities produces the weight change $\Delta W^{(k)}_{ij}$ shown beneath the diagram, revealing that EBD naturally realises the classical three‑factor learning rule in neural networks."*
>
> Regarding the experimental results, we have already included several plots of loss, accuracy, and correlation over training epochs in Appendix.  We included a new figure for correlation decay of CNN model in Appendix J.  We would be glad to add further result figures in the appendix, if the reviewer has specific suggestions.
>
> >2. Mathematical notation, despite rigorous, is hard to follow at times and requires careful reading. This is excellent for the appendix but some expressions could be optimized for readability and clarity in the main text.
>
> We appreciate the reviewer’s constructive comment regarding the mathematical notation. In the revised version, we will further clarify  the notation in the main text to improve clarity and readability, while maintaining rigor. Specifically, we will unify notational conventions to eliminate redundant representations of the same variables, as also noted in other reviewer comments. Much of the perceived complexity arises from the explicit indexing over batch, sample, layer, and vector components, which we include for the sake of precision and reproducibility.  We  are  open to implementing simplifications that do not compromise these objectives and would welcome specific suggestions for improving readability. In order to simplify expressions, in the revised article, we provided batchsize $B=1$ versions of EBD update rule in (7) and power normalization loss in (Eq. (8) of initial submission).
>
> >3. Some controls are missing, specifically with regards to the regularisation terms preventing collapse (see questions below)
>
> We address this comment through our response to the reviewer's questions below.
>
> ---
> ### Questions:
>
> >1. I understand how the EBD surrogate loss function can lead to collapse in the absence of activity regularization. However, I am curious to what extend the regularizers are necessary. Without regularization, do all networks eventually collapse? If so, how sensitive are the results to the choice of regularization hyperparameters? What was the reason to choose power normalization?
>
> We appreciate the reviewer’s insightful question. As noted in the manuscript, the EBD surrogate loss admits a trivial minimizer corresponding to all-zero weights and collapsed representations. This necessitates the use of regularization mechanisms to ensure meaningful learning dynamics. To this end, we introduce two complementary regularization terms: entropy regularization and power regularization.
>
> The entropy regularization encourages the use of the full representational capacity of each layer by promoting diversity in activations. However, this term alone can drive the network toward large activation magnitudes. The power regularization counterbalances this tendency by penalizing excessive activation energy, while also discouraging vanishing activations—thus playing a stabilizing role. Together, they maintain non-trivial, high-entropy yet bounded representations throughout training.
>
> While we have not conducted an exhaustive ablation study across all models and datasets, our preliminary experiments indicate that regularization significantly improves performance. For instance, in the MLP trained on CIFAR-10, removing both regularizers leads to a test accuracy drop from 55.5% to 53.1%. We also observe that entropy regularization benefits the DFA baseline, supporting the idea that it has broader utility beyond EBD.
>
> Interestingly, training with EBD does not  collapse even when regularizers are turned off—likely due to the use of non-zero, sufficiently diverse weight initializations. Yet, regularization makes the learning dynamics more robust and consistent.
>
> Beyond their algorithmic role, these regularizers also have biological interpretations: the power regularization can be viewed as a form of homeostatic control regulating the activations of neural activity, while entropy regularization aligns with the CorInfoMax framework, providing a potential normative explanation for the emergence of lateral connections in biological networks.
>
> A more detailed analysis of collapse mechanisms and regularization effects is an important direction for future work.
>
> >2. Similarly, I am curious to what extend the method relies on the forward projections explained in Appendix C.5. How does removing these forward projections impact performance?
>
> Our preliminary results suggest that the forward projections described in Appendix C.5 do not lead to significant performance gains—at least under the limited scope of our hyperparameter search. These projections were included only in the MLP experiments, and their associated weights were trained with very small learning rates (see Tables 3 and 4 in Appendix I.4.4), indicating a limited functional role during training. While theoretically motivated as a potentially useful mechansim, in practice their contribution appears marginal in the tested configurations. We plan to further investigate their impact in broader settings as part of future work.
>
> >3. How robust is EBD to different choices of the nonlinearity g(h)?
>
> We appreciate the reviewer’s interest in the role of the nonlinearity $g(h)$ within EBD. Rather than viewing the choice of $g(h)$ as a potential vulnerability, we regard it as an opportunity to enhance performance, as an additional degree of freedom. While our current study primarily used simple form of  $g(h)=h$ and $g(h)=h^2$, we conducted preliminary experiments with alternative functions—such as sinusoidal nonlinearities—which yielded no signifcant performance improvement, though these investigations were not extensive.
>
> We believe the choice of  $g(h)$ is an important design dimension that warrants deeper exploration. In particular, experimental studies on three-factor learning rules in biological systems may offer guidance on function classes that are optimized by nature. We see this as a promising direction for future work, aimed at both improving performance and drawing closer connections to biologically plausible computation.

---

> > ### Comment · Reviewer_Dubd · 2025-08-05
> >
> > I thank the authors for answering my questions and the clarification regarding the regularization parameters.
> >
> > I maintain one question regarding the forward projections:
> >
> > > Our preliminary results suggest that the forward projections described in Appendix C.5 do not lead to significant performance gains
> >
> > If there are no performance gains from adding these projections to the model, arguably making it more complicated, why include them in the first place? I agree that *in theory*, these projections might help - and still could in different task settings - I would recommend removing them from this paper to make the model simpler and explore their benefit in future work.

---

> ### Author Response · Authors · 2025-08-05
> **Response to Reviewer Dubd**
>
> We thank the reviewer for the suggestion to improve clarity. While forward projections are not central to our current findings, they offer an intriguing option from both algorithmic/theoretical and biological standpoints. In the revision, reflecting on the reviewer’s input, we have made their discussion more concise and adjusted the framing in the main text to ensure a clearer and more focused presentation, while emphasizing their potential biological relevance and outlining them as directions for future work.
>
> In Section 2.4.2, we revised the content to present the mechanism in a restrained and biologically plausible context:
>
> - *“As an optional extension to the EBD framework, hidden layer activations may be projected forward to the output layer to influence the adjustment of final-layer parameters. This idea is loosely motivated by biological evidence of long-range projections from early sensory areas to higher-level regions [50, 51], suggesting that such bottom-up communication may support learning in neural systems A detailed formulation is provided in Appendix C.5, and we leave its further exploration to future work.”*
>
> Finally, the Extensions paragraph in our Conclusion now includes the following sentence to signal our intent to pursue this direction further:
>
> - *“We also plan to further investigate optional forward projections as a mechanism to enhance coupling between hidden layers and the output, with the aim of improving upon the modest empirical gains observed in our current setup and better understanding their potential normative role in biologically plausible learning systems.”*
>
> Although forward projections do not yield significant accuracy improvements in the present experiments, we note that accuracy is not the sole criterion of interest—particularly in biologically plausible learning frameworks, where such mechanisms may offer normative explanations for bottom-up long-range projections observed in neural systems. We would like to keep the appendix section for future reference, as we plan to explore the functional role of forward projections in more detail in follow-up work.
>
> We hope these revisions address the reviewer’s point, and we remain open to further suggestions/improvements.

---

> > ### Comment · Reviewer_Dubd · 2025-08-09
> >
> > I thank the authors for their swift reply and appreciate the changes they made to the text that highlight the motivation for including such forward projections.
> >
> > All my questions and concerns have been answered fully.

---

> > > ### Author Response · Authors · 2025-08-09
> > > **Thanks to the Reviewers**
> > >
> > > We thank the reviewers for their time, as well as constructive and positive feedback. Your comments have been instrumental in improving the clarity, scope, and presentation of this work.

---

### Official Review · Reviewer_sM7v · 2025-07-03

**Clarity:** 4
**Significance:** 4
**Originality:** 4
**Rating:** 5
**Confidence:** 3

**Summary:**

This work develops a novel approach for solving the credit assignment in a biologically plausible way. The authors demonstrate theoretically and experimentally that using a loss function that penalizes how correlated the individual layer activations are with the output error is an effective way to train neural networks. They additionally demonstrate that their approach is an extension of a three-factor model for synaptic plasticity.

**Questions:**

1. Why is the performance of DFA + E not underlined in Table 1?

**Ethical Concerns:**

["NO or VERY MINOR ethics concerns only"]

**Final Justification:**

I continue to believe this work is of high quality and will be of interest to the NeurIPS community. For that reason, I recommend accepting it.

**Limitations:**

The authors discuss some of their limitations, but I think this could be discussed in more detail. The authors had to use several regularizers to prevent collapse. The authors also only evaluated on small networks and datasets. More discussion on these limitations, and others, would be helpful.

**Quality:**

4

**Strengths And Weaknesses:**

**Strengths**

1. This paper is well written and well motivated. The key points were well summarized and the description of their method was largely clear.

2. The core idea of the paper is (to my knowledge) novel and well grounded in theory.

3. The numerical results demonstrate the effectiveness of the approach and illustrate its better performance than other state-of-the-art methods.

4. I think this work will lead to interesting new directions and will be of broad interest to the NeurIPS community.

**Weaknesses**

I found no major weaknesses of this paper. There are, however, a few points I think would make this work stronger.

1. The authors demonstrate that the layer correlations decay with training, when using BP (Fig. 1C). I think more on this in the main text (and not just in the Appendix J) would be good. Showing that this is true across different architectures and datasets, and comparing the time-scale over which the correlations decay vs. the time-scale over which the loss decays would be interesting.

2. I found the connection with the three factor learning rule to be very interesting, but the abbreviated discussion made it a little hard to follow. In particular, it felt like the notation changed and it was not clear to me how Eq. 7 mapped onto the 3-factor learning rule. More on this would be helpful.

3. In Table 1, DFA + E (for MNIST) has nearly the exact same values as EBD, but it does not have its accuracy underlined (despite cases of actual ties). Is this because the performance is slightly different and EBD wins? Or is there maybe a typo on the performance of DFA + E?

**Minor points**

1. I think Fig. 1 could be made bigger, to make it easier for the reader to see what the authors are showing.

2. In Eq. 3, there is the variable $R_{g \epsilon}^{(k)}$. Then on line 140, it is denoted as  $R_{g^{(k)} (h(k)),  \epsilon}$. Are those supposed to be the same thing or did I miss something?

---

> ### Author Rebuttal · Authors · 2025-07-30
>
> We would like to the reviewer for the constructive comments.
>
> ### Strengths
>
> > 1. well written ... well motivated... clear.
> > 2. ... novel and well grounded in theory
> > 3. ...numerical results demonstrate the effectiveness... better performance ...
> > 4. lead to interesting new directions ... broad interest to the NeurIPS community.
>
> We appreciate the positive comments by the reviewer.
>
> ---
>
> ### Weaknesses
>
> >I found no major weaknesses...few points... make ...stronger.
> >1. The authors demonstrate that the layer correlations decay with training, when using BP (Fig. 1C). ... more on this in the main text (and not just in the Appendix J) would be good. Showing that this is true across different architectures and datasets, and comparing the time-scale over which the correlations decay vs. the time-scale over which the loss decays would be interesting.
>
> We thank the reviewer for this helpful suggestion regarding presentation. Due to space constraints, we placed the extended analysis of correlation dynamics in Appendix J, while highlighting key aspects (e.g., Figure 1C) in the main text. Following the advice by the reviewer, we included a correlation decay example  for the CNN architecture and CIFAR-10 dataset, which demonstrates that the correlation decay behavior in Figure 1.c replicates as expected. In the main text, Section 2.2 (line 125), we included a new sentence: "Similar correlation declining trends are also observed across different datasets and architectures (see Appendix J)."
>
> We also agree that exploring the time scales of correlation decay versus loss reduction across architectures is an interesting direction for future investigation.
>
>
> >2. I found the connection with the three factor learning rule to be very interesting, but the abbreviated discussion made it a little hard to follow. In particular, it felt like the notation changed and it was not clear to me how Eq. 7 mapped onto the 3-factor learning rule. More on this would be helpful.
>
> We thank the reviewer for pointing out the need for a clearer exposition regarding the connection to the three-factor learning rule. Eq. (7) expresses the gradient-based update derived from minimizing the layerwise decorrelation loss, but we acknowledge that the mapping to the classical three-factor form may not have been sufficiently transparent.
>
> To clarify this connection, we note that Eq. (7) in more explicit form (with $\vartheta$ substituted in to the equation) is given by
>
> $
> \Delta W_{ij}^{(k)}[m] = \zeta \sum_{n = mB + 1}^{(m+1)B}
> g'^{(k)}_i(h^{(k)}_i[n]) \cdot f'^{(k)}(u^{(k)}_i[n]) \cdot q^{(k)}_i[n] \cdot h^{(k-1)}_j[n]
> $
>
> when the batch size is set to \( B = 1 \), Eq. (7) simplifies to:
>
> $$
> \Delta W_{ij}^{(k)} \propto
> g'^{(k)}_i(h^{(k)}_i) \cdot f'^{(k)}(u^{(k)}_i) \cdot q^{(k)}_i \cdot h^{(k-1)}_j.
> $$
>
> This corresponds directly to the canonical three-factor learning rule form:
> $$
> \Delta W_{ij} \propto \text{(post-activity)} \times  \text{(modulatory signal)} \times \text{(pre-activity)} ,
> $$ with the following identifications:
>
>   - **Pre-synaptic activity:**  $h^{(k-1)}_j$,
>   - **Post-synaptic activity:** $g'^{(k)}_i(h^{(k)}_i) f'^{(k)}(u^{(k)}_i)$,
>   - **Modulatory signal:** $q^{(k)}_i $, the projection of the output error onto the activations of layer $k$.
>
>
> In the revised version, to make the connection between EBD update and three-factor learning rule more clear, we performed following changes:
>
>
> - in Section 2.1,  we included the EBD update expression for batchsize $B=1$: (Eq. (8))
>
> *"For the special case of batchsize, $B=1$, the weight update in (7) simplifies to
>
> $$
> {\Delta W_1^{(k)}[m]}_{ij} = \zeta   {g'}^{(k)}_i(h_i^{(k)}[m]) {f'}^{(k)}(u_i^{(k)}[m]) q_i^{(k)}[m] h_j^{(k-1)}[m].
> $$
> "
>
>
> - we modified the the last sentence of the first paragraph of Section 3.1 to clarify the connection between EBD update and three-factor learning rule:
>
> *"...   In contrast, EBD update for batchsize $B=1$ in (8) naturally matches the three-factor structure:..."*
>
> - We also added a new figure  to Section 3, which describes EBD  learning as a three‑factor synaptic update. The following is the paragraph that refers to this figure:
>
> *"Figure 2 breaks the EBD weight update (8) into its three interacting factors.  The presynaptic term is the activity $h_j^{(k-1)}$ from the sending unit; the postsynaptic term is $g_i^{\prime(k)}f^{\prime(k)}$ computed from the receiving unit’s own activation; and the modulatory term is the broadcast error $q_i^{(k)}$, derived from the network’s output error $\boldsymbol{\epsilon}$.  Multiplying these three quantities produces the weight change $\Delta W^{(k)}_{ij}$ shown beneath the diagram, revealing that EBD naturally realises the classical three‑factor learning rule in deep networks."*
>
>
> >3. In Table 1, DFA + E (for MNIST) has nearly the exact same values as EBD, but it does not have its accuracy underlined (despite cases of actual ties). Is this because the performance is slightly different and EBD wins? Or is there maybe a typo on the performance of DFA + E?
>
>  We thank the reviewer for this careful observation. The values for DFA + E and EBD on MNIST are indeed very close and we'll underline DFA +E values for MNIST and DFA value for MNIST-CNN  in the revised article's Table 1.
>
> >Minor points
>
>
>
> >1. I think Fig. 1 could be made bigger, to make it easier for the reader to see what the authors are showing.
>
> We thank the reviewer for this helpful suggestion. We agree that enlarging Figure 1 would improve readability and help better illustrate the comparisons, particularly in panel Fig 1.c. In the submission, we kept the figure compact to reserve space for more discussion and theoretical development. However,  in the revised article, we increased its size so that some details and text are better visible.
>
>
>
> >2. In Eq. 3, there is the variable $R_{ge}^{(k)}$. Then on line 140, it is denoted as
> $R_{g^{(k)}(h(k)),e}$. Are those supposed to be the same thing or did I miss something?
>
> Thanks for pointing out this typo. As the reviewer correctly noted these two different expressions are supposed to be same. We replace the wrong expression in line 140 (also after line 161, at line 163, Step 1 and Step 2 of Algorithm 1 and instances inside Algorithm 3 in Appendix I.3.2 ) with $R_{ge}^{(k)}$.
>
> ---
>
> ### Questions:
>
> >Why is the performance of DFA + E not underlined in Table 1?
>
> As we noted in our earlier response,  the absence of underlining for DFA + E in Table 1 is due to an oversight in formatting and we will correct this in the revised version.

---

> > ### Comment · Reviewer_sM7v · 2025-08-01
> >
> > I thank the authors for their detailed rebuttal. All my comments have been sufficiently addressed.

---

> > > ### Author Response · Authors · 2025-08-09
> > > **Thanks to the Reviewers**
> > >
> > > We thank the reviewers for their time, as well as constructive and positive feedback. Your comments have been instrumental in improving the clarity, scope, and presentation of this work.

---

### Decision · Program_Chairs · 2025-09-17

**Decision:**

Accept (spotlight)

**Comment:**

The paper present a method for addressing credit assignment by error broadcasting wrt specific correlated layer activations.  The authors both ground their approach theoretically and demonstrate experimentally its effectiveness.  All reviewers agree on the quality of the paper and its expected impact.